

# Boreal forest soil is a significant and diverse source of volatile organic compounds

Mari Mäki[1,2], Hermanni Aaltonen[3], Jussi Heinonsalo[3], Heidi Hellén[3], Jukka Pumpanen[4], and Jaana Bäck[1,2]

[1]Institute for Atmospheric and Earth System Research / Forest Sciences, Helsinki, 00560, Finland.
[2]Faculty of Agriculture and Forestry, University of Helsinki, Helsinki, 00790, Finland.
[3]Finnish Meteorological Institute, Helsinki, Helsinki, 00560, Finland.
[4]Department of Environmental and Biological Sciences, University of Eastern Finland, Kuopio, 70600, Finland.

Keywords: volatile organic compounds, boreal forest, organic soil horizon

*Correspondence to*: Mari Mäki (mari.maki@helsinki.fi)

**Abstract.** Vegetation emissions of volatile organic compounds (VOCs) are intensively studied world-wide because oxidation products of VOCs contribute to atmospheric processes, but the quantities by which different species of VOCs are produced by soil, or how effectively belowground VOCs are released into the atmosphere from soil remains largely unknown. This is the first published study that measures belowground VOC concentrations at different depths in a podzol combined with
simultaneous soil surface flux measurements in a boreal coniferous forest. More than 50 VOCs, dominated by monoterpenes and sesquiterpenes, were detected in the air space in the soil during the two measurement campaigns. Organic forest soil was a significant monoterpene source as it contained fresh isoprenoid-rich litter, and the concentrations of monoterpenes were comparable to the VOC concentrations in the air above the coniferous forest. Belowground monoterpene concentrations were largely decoupled from forest floor monoterpene fluxes; thus, it seems that production processes and storages of VOCs partly
differ from those VOCs that are simultaneously emitted from the soil surface. Relatively high isoprenoid concentrations were measured under snow cover, which indicates that snow and ice cover hinders gas diffusion and causes belowground accumulation of VOCs when the activity of vegetation is very low.

## 1 Introduction

Soil and understorey vegetation emit VOCs and these emissions are released from the diverse storages and processes (Hayward
et al., 2001; Smolander et al., 2006; Leff and Fierer, 2008; Bäck et al., 2010; Aaltonen et al., 2011; Faubert et al, 2012, and Mäki et al., 2017). These studies reported that VOCs are produced by understorey vegetation, roots, decomposition processes, soil microbes, and vegetative litter concentrated in the organic soil layer. Organic soils can be a substantial source of VOCs due to their high organic carbon content and vegetative litter that drives decomposition processes by releasing easily available carbon for microbial metabolism.



VOC exchange from the boreal forest floor varies from several per cent to tens of per cent of the boreal forest VOC exchange depending on the season (Aaltonen et al., 2013) and VOCs such as isoprene, monoterpenes, and especially sesquiterpenes have a precursor potential for secondary organic aerosol (SOA) formation. An SOA is formed in the atmosphere from condensed oxidation products of VOCs, and SOA particles contribute to cloud formation and affect the Earth's radiation

budget by scattering and absorbing solar radiation (Arneth et al., 2010, Virtanen et al., 2010, Mahowald, 2011). This outcome is opposite to the effect of greenhouse gases, which is warming the climate. Warming can change vegetation cover and almost double VOC emissions from subarctic and arctic plants (Faubert et al., 2010, Kramshøj et al., 2016). Warming can also affect VOC synthesis in soils by mediating decomposition processes, whereby microbial enzyme activity is regulated by soil temperature and water content (Davidson and Janssens, 2006).

Soil temperature and humidity influence many physical and biological processes related to VOC formation in soils (Asensio et al., 2007; Aaltonen et al., 2013). Soil water content impacts upon the transport of organic compounds (Zhong et al., 2014) and VOC emissions from vegetation (Svendsen et al., 2016), whereas gas volatilization and diffusion are mainly regulated by temperature. Soil water content also affects the decomposition of soil organic matter i.e. biological processes (Davidson and Janssens, 2006). Soil temperature and water content should be measured in parallel with belowground VOC

concentrations and soil surface flux measurements to study how effectively VOCs from soils are released into the atmosphere from very complex structured podzol soils.

The VOCs in soils have also been suggested to have an effect on biological interactions, although the quantities and functions of compounds in soils are largely unknown (Tholl et al., 2006). VOCs can promote plant growth, control the nitrogen cycle, affect microbial metabolism and transmit long-distance communication between different decomposers (Insam

and Seewald, 2010, Asensio et al., 2012, Peñuelas et al., 2014, Tahir et al., 2017). A deeper understanding on the dynamics of soil processes and the roles of different soil components to VOC formation is needed (Asensio et al., 2007, Leff and Fierer, 2008; Gray et al., 2010). The wintertime dynamics of soil VOC production is especially interesting, as activity of the vegetation during the snow cover period is low, but the concentrations in soil and inside the snowpack can be quite high (Aaltonen et al., 2012), which is probably due to microbial decomposition activity.

Studies on belowground concentrations of VOCs are scarce (Lin et al., 2007), especially those in which measurements are made in situ. Earlier studies (Smolander et al., 2006; Pihlatie et al., 2007; Pumpanen et al., 2008; Leff and Fierer, 2008; Asensio et al., 2008; Gray et al., 2010) measured greenhouse gases such as $CO_2$ and $CH_4$ belowground routinely or presented results on soil VOC content, obtained by laboratory measurements from soil cores. There is, however, no available established and well-evaluated method for measuring VOCs belowground; it is also clear that these experiments have

significant limitations compared to measuring VOC exchange in undisturbed forest soil. Laboratory experiments allow the manipulation of environmental conditions, but cause severe disturbances to natural soil processes that may include the regulation and release of VOCs from damaged roots and interfere with the balance between the roots and soil microbial components. We have developed a method to collect VOC samples in situ from collectors installed belowground and which equilibrates with VOC concentrations and fluxes of the surrounding soil.



Belowground VOC concentration measurements were conducted on organic and mineral soils in a boreal coniferous forest during two measurement campaigns from November 2008 to end of 2011 and from April to December of 2016. Belowground VOC concentrations were also compared with VOC fluxes in and from the forest floor in 2016. The overall aim of this study was to identify and quantify the VOC compounds that originate from the boreal podzolized forest soil at

different depths, in addition to studying the association of VOC concentrations with VOC emissions from the boreal soil surface. We used the following hypotheses in our study: (1) Organic soil and the top mineral soil (the A-horizon) produce a major part of the VOC emissions as organic soil contains isoprenoid-rich litter, and fine roots and root-associated microbes are concentrated in the top horizon of the mineral soil. (2) The seasonal dynamic and the production processes of belowground VOC concentrations are similar to those that contribute to VOC fluxes from the soil surface. (3) Soil temperature and water

content significantly affect VOC production belowground. (4) Belowground VOC concentrations differ between years.

## 2 Material and methods

### 2.1 Measurement site

The campaigns were performed in the southern boreal forest at the SMEAR II (Station for Measuring Ecosystem-Atmosphere Relations) station (61°51'N, 24°17'E, 180 m a.s.l) (Hari and Kulmala, 2005). The forest is a 56-yr old (in 2017) Scots pine

stand (*Pinus sylvestris* L.) with mean ~18 m stand height and a tree density ~1170 ha$^{-1}$ (Ilvesniemi et al., 2009). Below-canopy vegetation includes tree seedlings such as *Sorbus aucuparia*, *Betula pendula* and *Picea abies* (Mäki et al., 2017) and the dominating vascular plants in ground vegetation are *Vaccinium myrtillus* L., *Vaccinum vitis-idea* L., *Deschampsia flexuosa* (L.) Trin., and *Calluna vulgaris* (L.) Hull. (Ilvesniemi et al., 2009; Aaltonen et al., 2011). In addition, the soil is 33–60% covered by mosses such as *Pleurozium schreberi, Dicranum sp.*, and *Hylocomium splendens* (Aaltonen et al., 2011). The soil

above the homogeneous bedrock is Haplic podzol (FAO-UNESCO, 1990) formed in a glacial till, with a depth range of 0.5–0.7 m (Hari and Kulmala, 2005). The total C storage of the soil is 7 kg m$^{-2}$ (Ilvesniemi et al., 2009) and it has been formed during the last 7000 years. Frequent forest fires have an influence on soil C recovery and turnover time (Köster et al., 2014) and the last occasion the SMEAR II forest site was burned was in 1962 when the cutting residues were slash burned on site. The vertical thickness of the organic soil is 6.0 cm, and that of A- and B-horizon 2.0 cm and 16 cm, respectively in 2016 (Mäki

et al., 2017). The mean C content was highest in the organic soil layer (356 mg g$^{-1}$), much lower in the A-horizon (32 mg g$^{-1}$) and lowest in the B (24 mg g$^{-1}$) and in the C-horizons (5 mg g$^{-1}$), when the plots were established in 1995 (Pumpanen et al., 2008). The average N content is also highest in organic soil (13 mg g$^{-1}$) and decreases towards the deeper soil horizons (~1 mg g$^{-1}$) (Table 1). The total surface area of the roots <2mm is 3.5 m$^2$ m$^{-2}$ in the organic soil, 1.8 m$^2$ m$^{-2}$ in the A-horizon and 0.8 m$^2$ m$^{-2}$ in the B-horizon (Ilvesniemi and Liu, 2001).  A Swedish study reported that total tree fine-root biomass from organic

soil down to 30 cm in mineral soil is 227 g m$^{-2}$ in Scots pine stands (Hansson et al., 2013).



## 2.2 Permeability test for the gas collector

A permeability test was performed to monitor how fast VOCs permeate from the soil into the collector and to determine how fast VOC concentrations stabilize between the air inside and outside the collector. The effect of soil moisture was also evaluated by a permeability test. These results were used as a background to facilitate taking decisions about measurement and

stabilization times. The permeability of the gas collectors for VOCs was determined in laboratory conditions before the installation in the field. The determinations was made for both dry and wet collectors by using a gas mixture contained known concentrations of nine compounds (methanol, acetonitrile, acetaldehyde, acetone, isoprene, methyl vinyl ketone, 2-butanone, hexanal, and α-pinene) and proton transfer reaction-mass spectrometer (PTR-MS, Ionicon LTD, Austria) for on-line analysis (Fig. 1). The PTR-MS enabled fast response monitoring of the diffusion of VOC mixture. The field conditions did not allow

the implementation of the PTR-MS for continuous VOC measurements, thus only an adsorbent tube collection method was used. The break length was designed so that the stabilisation time of VOC concentration between the collector inside and outside air was clearly shorter than the 15-min break. All the VOC standard compounds permeate the collector easily and concentrations reach a constant level in order of minutes (maximum 7 minutes) also with the wetted collector (Fig. 2). α-Pinene was the heaviest compound in the VOC gas mixture, and consequently its diffusion through the wall of the collector

was the slowest of the VOCs measured. In contrast methanol peaked immediately after introducing the gas mixture into the glass bottle, but after that it stabilised quickly. It can be assumed therefore that stabilization of sesquiterpenes would take longer, since they are heavier than monoterpenes, thus the 15-min break time was chosen.

## 2.3 VOC concentration measurements in the soil profile

During the first campaign from 2008 to 2011, samples were collected from cylindrical hydrophobic gas collectors (4 cm in

diameter, 12 cm long) that had been installed into the soil pits (setup 1, Fig. 3a). The gas collectors were made of polytetrafluoroethylene (PTFE) by sintering, with pore sizes of 5–10 μm. The pores in the collectors allow the diffusion of gases to occur. The sampling system consisted of a gas permeable PTFE-tube (International Polymer Engineering, Arizona, USA) connected to stainless steel tubes from both ends with air-tight connections using Swagelok connectors (Swagelok, Straight fitting, Union 10) with the pits (setup 2, Fig. 3a). A mesh made of stainless steel was installed around the PTFE-tube

to protect the PTFE tube from physical damage from contact with the soil.

During the second campaign in 2016, samples were collected using PTFE tubes (International Polymer Engineering, Arizona, USA) that had been installed into the soil pits (setup 2, Fig. 3b), where the porosity enabled the diffusion of the VOC and air gases into the tubes. For aboveground sampling, two PTFE sampling tubes (8 mm internal diameter) were installed inside stainless steel tubes (10 mm internal diameter) and connected to the gas collector. The PTFE tubes were installed inside

the stainless steel tubing to prevent possible diffusion of VOCs through the PTFE and to protect the tubing against physical damage.



Samples were collected by circulating air in the gas collectors and through Tenax TA–Carbopack-B adsorbent tubes at flow rates that ranged 100–150 ml min$^{-1}$ using portable pumps and impermeable PTFE tubing. Air was aspirated from the collector and pumped through the adsorbent tube then returned back to the collector. The possibility of creating a flux collectors that did not originate from the actual measured horizon was minimized by using a relatively small flow rate and circulating the gaseous mixture back to the collectors. The aboveground ends of the PTFE tubes were closed between the samplings. Each sampling consisted of four 15-min pumping periods, with 15-min time intervals between them, which enabled the VOC concentrations to equilibrate fully between the collectors and into the soil around them. The permeability test revealed that all compounds in the VOC standard permeated into the collector easily and concentration in the collector reached a constant rate in the order of minutes (maximum 7 minutes) for both the dry and with the wetted collectors (Fig. 2). The total amount of VOCs in the air volume inside the collector (~0.15 L) alone would not have been sufficient for analysis as this volume of sample was below the detection limit. The sampling times were prolonged to 60 min (a total of 1 h 45 min with breaks), which resulted in a total sample volume of 6–9 L. These measurements were designed so that they would cause a minimal disturbance to the soil profile: the pits were carefully prepared and soil layers kept apart so that they could be placed back into the pit as close to the original intact soil profile as possible.

### 2.3.1 Measurement setup for VOC concentration profile of the first campaign, 2008 to 2011

The collectors were permanently installed in the three permanent soil pits at two depths, 5 and 17 cm below the soil surface (0 cm being the surface of organic layer, not mineral soil) in May 2008 (Fig. 3a). The upper collectors were installed at the humus layer-mineral soil interface, the lower collectors completely embedded in the mineral soil horizon (Fig. 3a). The measurements were performed during the snow free seasons that started on November 2008 and ended on October of 2011; a total of 18 samplings events (Table A2) that provided 104 individual samples. The long-term annual mean precipitation and the annual mean temperature at the SMEAR II station are 711 mm and 3.5°C, respectively (Pirinen et al., 2012). During the sampling period, the year 2009 was clearly drier than normal (mean annual precipitation 565.5 mm), and the years 2008 and 2011 were warmer than normal (mean annual 5.8°C for 2008 and 6.1°C for 2011).

### 2.3.2 Measurement setup for VOC concentration profile of the second campaign, 2016

The gas collectors were installed in four different soil horizons (H-horizon, A-horizon, B-horizon, and C-horizon) in five soil pits permanently in 2011 (Fig. 3b, Table 2, and Table A1). The pits were carefully excavated and the soil horizons H-, A-, B and C- were kept separate. The collectors were installed in the vertical face interfacing with the undisturbed soil in the excavated pits to minimize the excavation disturbance effect. After the installation, the soil layers were carefully placed back in the original order and compacted to the original volume of the soil. The measurements started on 21th of April 2016 and ended on 2nd of December of 2016; a total of 13 sampling events (Table A2). The mean temperature was 5.9°C. There was some snow on the ground in November and permanent snow cover in December, but no snow remained in April, when measurements commenced.



## 2.4 VOC and CO₂ flux measurements and supporting data

Soil collars for VOC and $CO_2$ flux measurements were placed next (20–50 cm) in the five VOC measurement pits (Table 2). Soil collars were placed in March, 2016, and the measurements were started in April, 2016. Isoprenoid and oxygenated VOC fluxes were measured using a dynamic enclosure chamber technique as described by Mäki et al. (2017). The headspace (height

40 cm, chamber volume 10 L) was a glass chamber placed for measurements on permanently installed soil collars (height 7 cm, diameter 21.7 cm). The fluxes of $CO_2$ were determined using a dark static chamber technique (diameter 20 cm and height 30 cm) whereby a concentration of $CO_2$ the closed chamber headspace was measured for 5 minutes using a GMP343 $CO_2$ probe (Vaisala Oyj, Vantaa, Finland) and the $CO_2$ efflux was calculated by linear fitting against time and $CO_2$ concentration in the chamber headspace (Pumpanen et al., 2015). The SMEAR II data from Avaa (https://avaa.tdata.fi/web/smart) was used

as an ancillary dataset (Hari and Kulmala, 2005). This dataset also included soil temperatures and soil water content for each soil horizon from the same measurement pits, where VOC concentrations were measured in 2016. Soil temperature was measured by thermistors (Philips KTY81-110, Philips semiconductor, Eindhoven, the Netherlands) and soil water content with TDR method (TDR 100, Campbell Scientific Inc., Logan, USA). Precipitation was measured by an FD12P weather sensor (Vaisala Oyj, Helsinki, Finland). The understory vegetation cover of the different species were visually estimated for each soil

VOC/$CO_2$-collar (Table 2).

## 2.5 Analytical methods

The adsorbent tubes were analyzed in the laboratory, using a thermodesorption instrument (Perkin-Elmer TurboMatrix 650; PerkinElmer, Waltham, MA, USA) attached to a gas-chromatograph (Perkin-Elmer Clarus 600) with a mass-selective detector (Perkin-Elmer Clarus 600T) (Aaltonen et al., 2011, Mäki et al., 2017). The sample tubes were desorbed at 300°C for 5 min,

cryofocused in a Tenax cold trap (-30°C) prior to injecting the compounds into the column by rapidly heating the cold trap (40°C min⁻¹) to 300°C. The mass detector used enabled simultaneous full scan and singular ion monitoring. Four-point calibration standards in methanol solutions were used, except for the isoprene measurements for which we used one gaseous calibration standard (National Physical Laboratory). The standards were injected into the sampling tubes and the methanol was flushed away for 10 minutes before the analysis. The analytical variability was determined using replicate standard analysis.

The detection limits varied from 0.0002 to 0.057 μg m⁻³ in concentration measurements and from 0.0005 to 1.477 μg m⁻² h⁻¹ in flux measurements (Appendix Table A3). The VOC concentrations of isoprene, monoterpenes, sesquiterpenes and different oxygenated VOCs ($C_4$-$C_{15}$ alcohols, carbonyls and acetates, methyl-2/3-furoates and α-pinene oxide) were analyzed. Calibration solutions for the sesquiterpenes, contained only longicyclene, isolongifolene, β-caryophyllene, α-humulene, α-gurjunene and β-farnesene. In 2016, other sesquiterpenes found in the samples were tentatively identified by their mass spectra

and retention indices and quantified as β-caryophyllene, isolongifolene, or longicyclene. One of the sesquiterpenes could not be tentatively identified and was therefore denoted as SQT1.

## 2.6 Calculations and statistical analyses




The VOC flux rate (µg m$^{-2}$ h$^{-1}$) calculations were performed using equations described by Mäki et al. (2017). The VOC concentrations (C, µg m$^{-3}$) for the different soil horizons were calculated with Eq. (1):

$$C = \frac{m}{\frac{V}{1000000}}\qquad(1)$$

where $m$ is the mass of sample (ng) and $V$ is sampled volume (ml), divided by 1000000 to calculate the unit conversion from ml to m$^3$. $V$ was calculated using Eq. (2):

$$V = F * t \qquad(2)$$

where t is the sampling time (min) and F is the flow rate of the sampling (F, ml min$^{-1}$).

Data analyses were performed with MATLAB software (version 2015a, MathWorks, Natick, MA, USA). The Kolmogorov-Smirnov and Shapiro-Wilkin tests were used to test the normality of the individual VOC concentrations for the different soil horizons (H-horizon n = 52, A-horizon n = 65, B-horizon n = 65, and C-horizon n = 65). The Kolmogorov-Smirnov and Shapiro-Wilkin tests were also used for the total fluxes of monoterpene, sesquiterpene, oxygenated VOC fluxes, chamber temperature (°C) and soil water content for each measurement pit (N = 6–13). The non-parametric Kruskal-Wallis test (n = 65, df = 1, significance level of p<0.100 (°), p<0.050, p<0.010, p<0.001) was used to determine whether the VOC concentrations of the soil horizons were statistically different from each other (Appendix Table A4). The non-parametric Kruskal-Wallis test (n = 13, df = 1) was used for comparing the following flux parameters between the different soil pits: CO$_2$, total monoterpene, total sesquiterpene, total oxygenated VOCs. The Kruskal-Wallis test was also used to compare chamber temperature, soil temperature and soil water content between the soil pits (Table 3). Pearson correlation coefficients were calculated for the following parameters: correlations between the monoterpene and sesquiterpene fluxes and concentrations within the different pits (n = 5–13, Appendix Table A5). The Pearson correlation coefficients (n = 34–65, Table 4) were also calculated for the following parameters: correlations between the monoterpene and sesquiterpene concentrations and between the soil temperature and water content within the different soil horizons. The monthly mean monoterpene concentrations were compared for the H-horizon and the B-horizon in summertime between years using the non-parametric Kruskal-Wallis test to determine if belowground VOC production differed strongly between years (n = 3–10) or between the different measurement pits and gas collectors (n = 17–24) used in campaigns one (2008–2011) and two (2016). The detection limit (µg m$^{-2}$ h$^{-1}$) of the VOC flux quantification (Appendix Table A3) was calculated for each VOC compound and for all 13 samplings based on the equations that can be found in the publication by Mäki et al. (2017). The detection limit (µg m$^{-3}$) of the VOC concentration quantification (Appendix Table A3) for the soil horizon measurements was calculated using the signal-to-noise ratio data obtained from the VOC quantification.





## 3 Results

Over 50 different VOCs were detected in the soil air during the two measurement campaigns with high annual variation in the belowground VOC concentrations. Belowground VOC concentrations were dominated by monoterpenes and sesquiterpenes, but the monoterpene concentrations were mainly decoupled from forest floor monoterpene fluxes.

### 3.1 Belowground VOC concentrations in the vertical soil horizons in 2016

Belowground VOC concentrations in the different soil horizons were compared (Fig. 4). Our hypothesis, that the concentrations in the topsoil were highest, was only partly confirmed. The median of the total monoterpene concentrations was highest in the organic soil (36 µg m$^{-3}$), lower in the A-horizon (10 µg m$^{-3}$, p<0.1) and significantly smaller in the lower horizons (B-horizon 4 µg m$^{-3}$, p<0.001 and C-horizon 4 µg m$^{-3}$, p<0.001) (Appendix Table A4). However, total sesquiterpene

and total oxygenated VOC (OVOC) concentration means were highest in the A-horizon (15 and 10 µg m$^{-3}$, respectively) (Appendix Table A4). The spatial and temporal variation in belowground VOC concentrations was remarkable, and statistically significant differences between soil horizons could not be observed for the total sesquiterpenes or for the total oxygenated VOCs. Sesquiterpenes and OVOCs are two very diverse groups of chemical compounds in which some compounds occurred in the highest concentrations in organic soil and other compounds in the A-horizon (Fig. 4, Appendix Table A4).

There were no differences in individual VOC concentrations between the pits when each soil horizon was tested separately (data not shown). Soil water content was high in the C-horizon, which led to the high humidity in the adsorbent tubes and consequently in some samples (pits 4 and 5) could not be analyzed using the thermal desorption-gas chromatography–mass spectrometry method. However, it can be assumed that concentrations would have been relatively low as low oxygen availability slows down microbial activity (Davidson and Janssens, 2006).

### 3.2 Belowground VOC concentrations and VOC fluxes from the soil surface in 2016

Total monoterpene concentrations in organic soil were highest in late summer and in December when the soils were under snow cover (Fig. 5). Monoterpene concentrations in mineral soil (A- and B-horizons) were generally higher in spring and summer and decreased towards autumn except in December, when concentrations suddenly increased (Fig. 5). In general, there seemed to be a clear trend that belowground concentrations were exceptionally high in December under snow cover. Total

sesquiterpene concentrations in mineral soil were clearly highest in spring, in early June, in late summer, and in October (Fig. 5). There was no clear seasonal variation in the organic soil, except in October and in December when the concentrations suddenly increased in the whole soil profile (Fig. 5).

Total monoterpene fluxes were highest in October and lowest in late summer, whereas total sesquiterpene fluxes were highest in spring, in late summer and in October. Total monoterpene flux varied between 19.7–61.9 µg m$^{-2}$ h$^{-1}$,

sesquiterpene flux between 0.7–11.2 µg m$^{-2}$ h$^{-1}$, and oxygenated VOC flux between 0.6–5.1 µg m$^{-2}$ h$^{-1}$ in 2016 (Table 3). There was no difference in VOC fluxes between measurement pits. The CO$_2$ flux was lower (0.03–0.08 mg m$^{-2}$ s$^{-1}$) when understorey



vegetation cover was high (45–63%) and $CO_2$ flux increased (0.13–0.21 mg m$^{-2}$ s$^{-1}$) when the understorey vegetation cover was low (10–35%) (Table 3).

Belowground monoterpene and sesquiterpene concentrations in vertical soil horizons were decoupled from forest floor monoterpene and sesquiterpene fluxes, when the whole data were combined contrary to our hypothesis. In the individual pits, the monoterpene flux in the individual pits correlated with the organic soil concentration in pit four ($R^2=0.78$, $p<0.05$) and with the A-horizon in pit five ($R^2=0.83$, $p<0.050$) (Appendix Table A5). The sesquiterpene flux also correlated with concentrations of the organic soil ($R^2=0.62$, $p<0.050$) and of the A-horizon ($R^2=0.72$, $p<0.010$) in pit three (Appendix Table A5). However, sesquiterpene concentrations and forest floor fluxes had similar seasonal variation (Fig. 5).

### 3.3 Soil temperature and water content impact on VOC concentrations, 2008–2011 and 2016

There was a moderate correlation between the monoterpene concentration and soil temperature in the organic soil ($R^2=0.35$ $p<0.010$) and on the B-horizon ($R^2=0.46$, $p<0.001$) and between the sesquiterpene concentration and soil temperature in the organic soil ($R^2=0.29$, $p<0.050$) from 2008 to 2011 (Table 4). Confirming our third hypothesis that soil temperature and water content can be used to explain belowground VOC synthesis. Relatively low correlation between the monoterpene concentration and soil temperature ($R^2=0.33$ $p<0.010$) and of the sesquiterpene concentration and soil water content ($R^2=0.23$ $p<0.050$) was also found in the A-horizon in 2016 (Table 4). The correlation between the monoterpene concentration and soil water content was always negative, but not significant. In general, soil water content and temperature variation remained at normal levels during the measurement years, compared to the values reported for the same site in the literature (Kolari et al., 2009).

### 3.4 Inter-annual variation of soil VOC concentrations from 2008 to 2011 and in 2016

We took measurements to establish whether the belowground VOC concentrations differ between years as podzol soil is heterogenic and measurements were conducted from the different soil pits during both measurement campaigns. Statistically higher ($p<0.050$) summertime concentrations for monoterpenes were measured in the organic soil in 2016 in comparison to the values obtained in the first campaign between 2009 and 2011 (Fig. 6). There was no significant difference in the B-horizon between the measurement campaigns. The organic soil showed seasonal variation in 2011 and 2016, when monoterpene concentrations increased together with soil temperature from May to July (Fig. 6).

Monoterpenes constituted almost 90% of the total VOC concentration, sesquiterpenes accounted for less than 10% between 2008 and 2011 (Table 5). Monoterpenes were observed in every single sampling, but sesquiterpenes were absent in the months of April, May, June, and November in 2009. Low concentrations of isoprene and oxygenated VOCs were also observed. The mean annual monoterpene concentrations in 2008‑2011 varied between 1.7 µg m$^{-3}$ and 6.3 µg m$^{-3}$ in organic soil and between 1.4 µg m$^{-3}$ and 4.4 µg m$^{-3}$ in mineral soil during the first campaign (Table 5). Similar to 2016, monoterpene concentrations in 2008‑2011 were almost consistently higher in the collectors that were located in the organic layer, but the differences between the organic and the mineral soil were small (Table 5). Monoterpene concentrations in 2009‑2011 were



generally highest in summer/early autumn in organic soil, whereas the concentrations in mineral soil tended to peak slightly later (data not shown). Monoterpene concentrations in 2016 were highest in organic soil in summer, in October and in December, whereas seasonal variation was relatively small in mineral soil. The mean sesquiterpene concentration in the organic layer in 2008-2011 was 0.3 µg m$^{-3}$ and 0.8 µg m$^{-3}$ in mineral soil, but the concentrations were basically similar in both

profiles (Table 5). The belowground concentrations of isoprene were low, only 0.03 and 0.01 µg m$^{-3}$ in organic and mineral layers in the 2008‑ 2011 period and they were also less than 0.06 µg m$^{-3}$ in 2016, except for October 2016 when isoprene concentration suddenly increased in the top of the mineral soil. Statistically significant differences between the organic and the mineral soil were not obtained for the 2008‑ 2011 period for any major compound or compound group and the spatial variation in belowground isoprenoid concentrations between the three measurement pits was substantial.

**4 Discussion**

This is the first study in which soil vertical layer VOC concentrations were analysed and compared with simultaneous chamber flux measurements in field conditions in a Boreal coniferous forest. We detected more than 50 different VOCs, mostly mono- and sesquiterpenes, and belowground concentrations of VOCs differed between soil layers during the second campaign. Sources of the forest soil VOCs probably differ depending on the compound and soil layer.

**4.1 VOC concentrations reflect the biological and physico-chemical properties of soil horizons**

Our results clearly show that monoterpene concentrations are highest in organic soil. Podzol soil surface is formed by fresh vegetative litter that contains easily decomposable glucose, starch and cellulose, and very slow-decomposable organic matter (Beyer et al., 1996; Prescott et al., 2000; 2010). Monoterpene concentrations and emissions from the organic layer are probably driven by the monoterpene rich litter, in which the decomposition processes are regulated by litter quantity and quality, climate

and soil microbial populations (Prescott, 2000). The decomposing litter has been assumed to be the main source for VOCs in the forest floor (Hayward et al., 2001; Leff and Fierer, 2008; Mäki et al., 2017). It is evident that both decomposers and the decomposing material affect the formation of VOCs, and also that VOCs released through the decomposition processes are probably very dependent on the litter type (Gray et al., 2010). Microbes are most active in organic soil, which contains easily available carbon for their metabolism (Makkonen and Helmisaari, 1998; Leff and Fierer, 2008; Pumpanen et al., 2008).

Organic carbon and nitrogen availability is typically higher in organic soil than in mineral soil (Parmelee et al., 1993; Deluca and Boisvenue, 2012).

Microbial community composition is determined by the carbon nitrogen ratio (C:N) ratio, pH and tree cover (Högberg et al., 2007), whereas high organic carbon and oxygen availability enhances the decomposition processes. A low pH favours fungi as the main decomposers over bacteria in boreal coniferous forest soils (Alexander 1977). However, sequencing

revealed that the dominating groups of bacteria in humus are Acidobacteria, Proteobacteria, and Actinobacteria (Timonen et al., 2017). The individual sources of VOCs are very difficult to determine under field measurement conditions, but laboratory experiments show the capability of soil fungi to produce and emit numerous volatile compounds (Bäck et al., 2010; Müller et



al., 2013). Roots of trees and perennial shrubs are also an important belowground VOC source (Smolander et al., 2006; Lin et al., 2007) and their uneven coverage also causes spatial variation among the concentration measurements. Detectable isoprene concentrations belowground were surprising because isoprene is known to be produced in photosynthetic tissues, and production is strongly light dependent without storages in plant cells (Monson et al., 1989, Delwiche et al., 1993, Sharkey and

Singsaas, 1995). However, laboratory measurements reported indicate that isoprene can also be produced by fungi (Bäck et al., 2010) and by needle litter during decomposition (Gray et al., 2010).

We detected higher monoterpene concentrations in organic soil compared to the B and C horizons, which could be explained by low biological activity as quantities of roots and organic carbon content for microbial metabolism decrease with depth. However, high monoterpene and sesquiterpene concentrations were also detected in the top horizon of

mineral soil, which contains the bulk of roots and most of the root-associated microbes. The VOCs in mineral soil may be related to the living roots or decaying root-litter. Monoterpene emissions from the root-soil interface can be quantitatively and qualitatively different from those emitted by dead roots (Lin et al., 2007), which can cause variation to the VOC concentrations between the soil horizons. Sesquiterpene concentrations were quite homogeneous between soil horizons, which indicates that sources for sesquiterpenes are more stable and possibly also relatively independent of environmental factors. Sesquiterpenes

measured under laboratory conditions are produced by endophytes (Bäck et al., 2010), decomposers (Rösecke et al., 2000, Bäck et al., 2010, Weikl et al., 2016) and ectomycorrhizal fungi (Ditengou et al., 2015). The low volatility and high reactivity of sesquiterpenes can result in much higher concentrations near the sources than average concentrations in the soil horizon. Quantified sesquiterpene concentrations can be underestimated since organic soil is highly porous media and sesquiterpenes as highly reactive compounds can be converted into other compounds by chemical reactions with soil air oxidants. A lack of

pure standards also increased the uncertainty of sesquiterpene analyses.

Our results show moderate correlation between isoprenoid concentrations and soil temperature, which was expected as biological and physico-chemical processes such as diffusion and volatility are directly influenced by temperature (Peñuelas and Staudt, 2010). Moreover, enzyme activity of microbial metabolism that lead to the VOC production (Mancuso et al., 2015) are strongly affected by temperature. The results also showed negative correlations between soil water content

and the monoterpene concentrations, although the correlation was not significant. Diffusion of gases can be effectively prevented by water, which blocks the microspores in the soil in wet weather or poorly drained soils. In sandy soils, water movement downward from organic soil horizon is usually efficient and this can also transport water soluble OVOCs (verbenone, 1-butanol, isopropanol, 2-butanone, 1-hexanol and cis-3-hexenyl acetate, and slightly water-soluble methyl-12-furoate, 1-penten-3-ol, 1-pentanol, butyl acetate, trans-3-hexen-1-ol, trans-2-hexen-1-ol, and α-pinene oxide) into the mineral

soil and reduce the differences in VOC concentrations between the organic and mineral soil horizons.

The soil concentrations were mostly decoupled from forest floor VOC fluxes, which indicates that belowground sources are different from those that release VOCs from the soil surface. Most of the measured fluxes at the forest floor level probably originates from understorey vegetation and decomposing organic layer, humus (Hewitt and Street, 1992; Aaltonen et al., 2011; Faubert et al., 2012; Rinnan et al., 2014). The forest floor vegetation also absorbs VOCs on the



moist leaf surfaces, which creates a bidirectional flux especially under moist conditions (Aaltonen et al., 2013). However, the time lag between concentrations deeper in the soil and the flux measured above the humus layer make it difficult to compare the concentrations with the fluxes. Temperature and moisture conditions are also probably different between belowground and soil surface, which suggests that effect of physico-chemical processes is different. There was some correlation between

isoprenoid concentration and fluxes, when individual soil pits were compared. The total uncertainty at the 10 µg m$^{-2}$ h$^{-1}$ emission level with the used quantification method was found to be relatively low, 14–44% for monoterpenes and 14–20% for sesquiterpenes (Mäki et al., 2017).

Spatial variation in isoprenoid concentrations was substantial, even though the forest at the site was fairly homogenous. Similarly, very high spatial variation in forest floor VOC fluxes at the SMEAR II stand was reported by Aaltonen

et al. (2011, 2013). This reflects the spatial heterogeneity in soil structure and soil processes, and which occurred in many other measurements, and underlines the importance of having a sufficient number of parallel sampling points. Despite the high spatial variability between the pits no significant differences in individual isoprenoid concentrations were found between each soil pit which were compared among soil horizons.

## 4.2 Seasonal and inter-annual variation

The three-year measurements of the 2009‑2011 period indicated noteworthy concentrations of isoprenoids in the belowground horizons that were similar in magnitude to reported aboveground concentrations. The following aboveground concentration ranges were obtained: α-pinene (0.2–6.3 µg m$^{-3}$), Δ-3-carene (0.1–2.5 µg m$^{-3}$), β-pinene (0.04–0.3 µg m$^{-3}$), and camphene (0.02–0.3 µg m$^{-3}$) in the same boreal coniferous forest by Hakola et al., (2009). Soil VOC concentrations are not directly comparable with air concentrations as the soil air volume is concentrated only into soil pores. Belowground isoprenoid

concentrations varied seasonally, and the highest concentrations were measured during summer and early autumn in 2009 and 2011, whereas high belowground concentrations monoterpene concentrations were measured in late summer, in October, and in December in 2016. The annual variability was also mostly covered, since measurements were executed in spring, summer and autumn in 2009, 2011, and also in 2016. The seasonal variation in monoterpene concentrations was more distinctive than in sesquiterpenes. However, the variation in the belowground isoprenoid concentrations was less clear than what has been

observed in isoprenoid fluxes sampled from the forest floor (Aaltonen et al., 2011, 2013; Mäki et al., 2017). The timing of the high concentrations also differs from the peaks in forest floor VOC fluxes that were observed in earlier studies. Soil VOC concentrations were highest as early as in mid and late summer, whereas forest floor fluxes do not start to increase before the end of the summer (Aaltonen et al., 2011). High soil water content in spring hinders the diffusion process and could also cause isoprenoid accumulation in the soil. Isoprene was not consistently observed and its highest concentrations were always

obtained during autumn, not during the season of active shoot growth. VOC concentration measurements were conducted in the different gas collectors during the first measurement 2008‑2011 and the second (2016) measurement campaigns. The experimental variables that changed in this study were structure, installation and the length of the recovery time between installation and measurements. Changes in any of these variables can impact upon the VOC concentrations between the two





campaign periods. The recovery period between installation and first sampling was as short as six months in 2008, but by 2009 it was 11 months. The recovery period was as long as five years by the time the second campaign was implemented.

During the second campaign, the measurements indicated noteworthy concentrations of isoprenoids belowground throughout the year. Soil concentrations and forest floor fluxes of sesquiterpenes were relatively high during

spring, which was contrary to the findings of two earlier studies that reported the branch measurements of sesquiterpene emissions mainly occurred in midsummer (Tarvainen et al., 2005; Hakola et al., 2006). Sesquiterpenes in soil can originate from vegetation and decomposition processes with decaying substrates. Belowground monoterpene concentrations varied seasonally and had high concentrations in late summer, in October, and in December. High concentrations in October are in agreement with observations of the timing of isoprenoid fluxes from the forest floor reported in other studies (Hellén et al.,

2006; Aaltonen et al., 2011, 2013; Mäki et al., 2017). Our results indicate that isoprenoid production is not limited only to the maximum litter production period in autumn, but that the organic soil is a relatively active VOC production source during the whole snow free period. Decomposition processes slowly release isoprenoids from needle storages (Kainulainen and Holopainen, 2002) and vegetation drops small amounts of litter the year-round, thus litter is a continuous and renewable VOC source on the soil surface. The low seasonal variation in monoterpene concentrations in deeper mineral soils compared to

organic soil and top mineral soil may be related to differences in source abundance between vertical soil horizons. The VOCs can also be captured in soils through adsorption on clay minerals (Deng et al., 2017), which indicates that VOCs are not released into the atmosphere.

We also measured the soil isoprenoid concentrations on one occasion during the snow cover period in this study. It is likely that the relatively high concentrations of VOCs we found in that sample taken from organic soil in wintertime

are related to physical characteristics of the snowpack. The snowpack would expected to have icy layers and low temperatures; characteristics that would render it to be relatively impermeable and thus hinder the diffusion of VOCs. Such a reduction in diffusion would result in the accumulation of VOCs at the snow-soil interface and also within the surface layers of soil itself. Aaltonen et al. (2012) measured both monoterpene ($0.4$–$6.2\ \mu g\ m^{-3}$) and sesquiterpene ($0.08$–$1.0\ \mu g\ m^{-3}$) concentrations inside the snowpack, and showed that the concentrations were generally higher close to the soil surface and lower just next the

snowpack air interface. We found that the wintertime monoterpene concentrations in the organic layer were high, and were in the same order of magnitude as those reported inside snowpack during winters (Aaltonen et al., 2012). Microbial activity can lead VOC production in the snowpack close to the soil surface (Liptzin et al., 2015).

It could be argued that the period of snow cover is diminishing due climate warming, which could mean that VOC emissions would increase overall and also as a result of elevated temperatures. Alternatively, activity might be reduced

because of waterlogging in warmer wetter winters, which could make snow cover less permeable (Aaltonen et al., 2012). The water-soluble VOCs can also be sequestered by wet snow. Instead VOCs would probably be released into the atmosphere in spring, when the snow melts. Chamber flux measurements showed that some VOCs are released through the soil surface and snowpack into the atmosphere in December during continuous snow cover, which indicates that soil and snowpack is also a

VOC source during wintertime. So far, there is a very small number of other studies that have measured VOC fluxes during wintertime and more research is needed on how soil contributes to the atmospheric processes in winter time.

## 5 Conclusions

Our unique measurement setup demonstrates that boreal forest soil is a significant and diverse VOC source, in which
dominating monoterpenes concentrations are comparable to the air concentrations above a coniferous forest. Soil is a potential VOC source during winter time and this phenomenon merits further study and our measurement setup would be a potentially useful tool for investigating that. These measurements revealed high belowground isoprenoid concentrations, thus, the next step is to study the formation processes of VOCs by using a laboratory approach. It would also be important to determine in the laboratory, how strongly production processes of VOCs are regulated by temperature and water content, and also study
how the warming climate will impact upon VOC fluxes from the boreal forest floor.

*Data availability.* Mäki, Mari (2017), "Data for the manuscript", Mendeley Data, v1. DOI: 10.17632/dn2rj3yf9p.1

*Author contributions.* Manuscript preparation and analyzing results (M.M. and H.A.). All authors contributed to the
15 experimental planning, the discussion of the results and the writing on the manuscript.

*Acknowledgements.* We would like to thank the staff of the SMEAR II and Hyytiälä Forestry Field station for their help and for use of the facilities in making this study. We gratefully acknowledge the financial support by the Jenny and Antti Wihuri Foundation, the Academy of Finland Centre of Excellence programme (project no 272041) and the Academy of Finland project
130984 and 286685 are also gratefully acknowledged.

*Disclaimer.* The authors affirm that this study is an original contribution and has not been submitted elsewhere. The authors declare no competing financial interests.

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

## Figures and Tables

**Table 1:** Soil characteristics of the measurement site. The depth of lower horizon border (cm), volume weight (g cm$^{-3}$), particle size of clay (%), silt (%), and sand (%), N-content (mg g$^{-1}$), C-content (mg g$^{-1}$), and pH value (in CaCl$_2$) of the measurement pits for the different soil horizons at the SMEAR II station in 1995. Values are means of measurement pits 1, 4, and 5.

| Horizon | Depth of lower horizon border | Volume weight (g cm$^{-3}$) | Rocks (% of weight) | Particle size (% of weight) | | |
| --- | --- | --- | --- | --- | --- | --- |
| | | | | clay | silt | sand |
| O | 0.00 | | | | | |
| A | 6.54 | 0.75 | 28.61 | 5.65 | 13.72 | 52.02 |
| B | 26.83 | 0.86 | 27.68 | 6.72 | 13.01 | 52.60 |
| C | 71.14 | 1.27 | 36.25 | 6.91 | 12.56 | 44.28 |

| Horizon | N-content (mg g$^{-1}$) | C-content (mg g$^{-1}$) | pH in CaCl$_2$ |
| --- | --- | --- | --- |
| O | 13.46 | 355.68 | 3.39 |
| A | 1.02 | 32.23 | 3.53 |
| B | 1.07 | 23.51 | 4.36 |
| C | 0.13 | 4.15 | 4.49 |





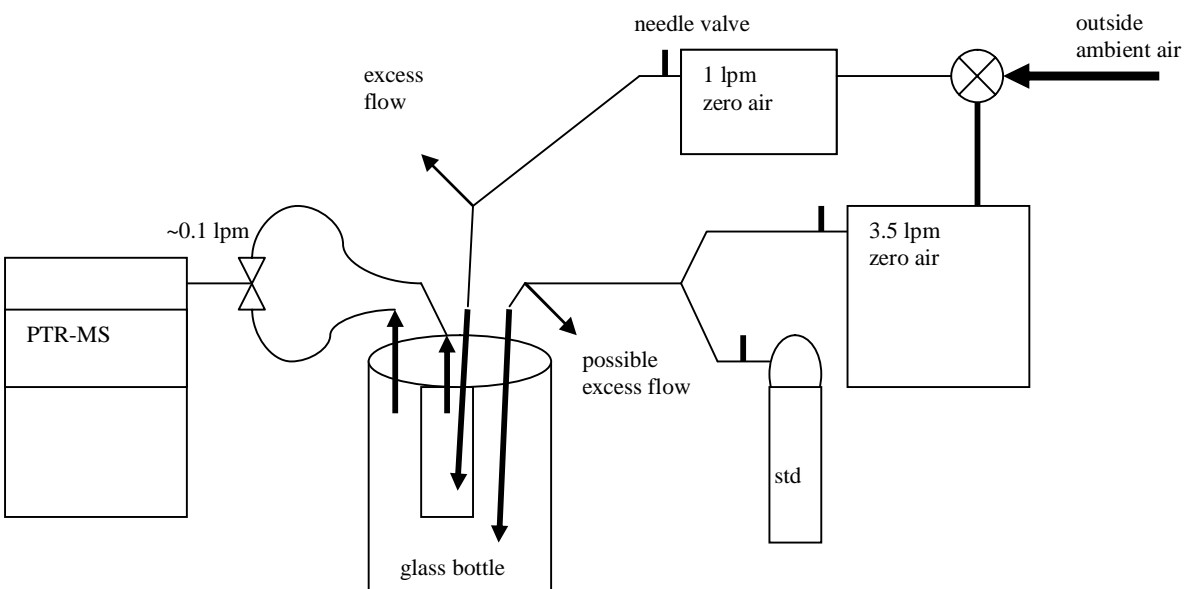

**Figure 1: Set-up for the permeability tests of the sintered gas collectors.**

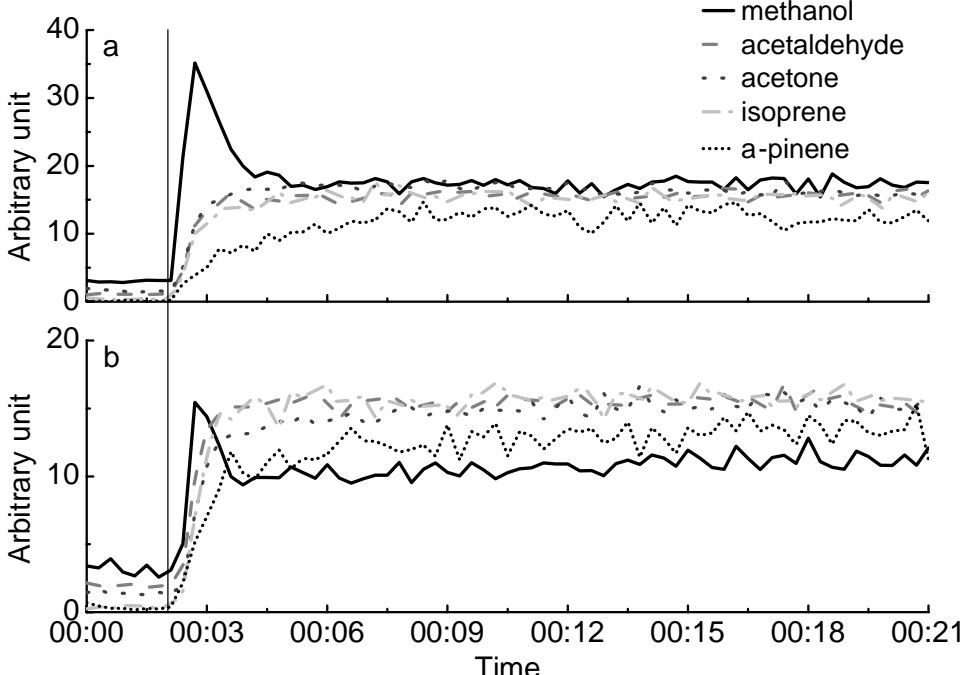

**Figure 2: Results of the permeability tests with the five VOCs. Panel a) shows the results with dry collector and panel b) with a**

5  **wetted collector. Vertical line shows the time point when the introduction of the VOC standard began.**





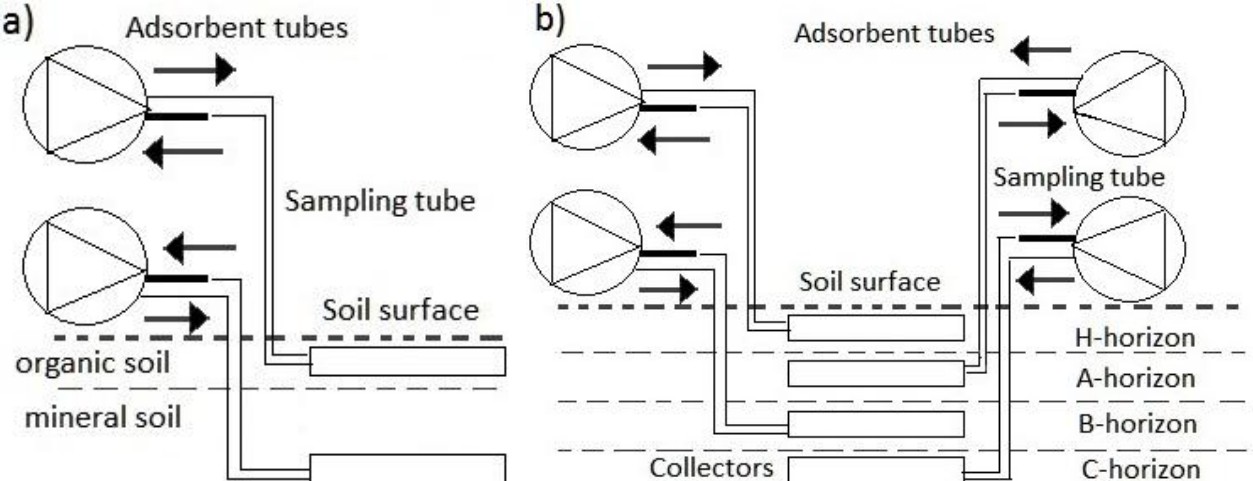

**Figure 3: Measurement set-up for soil VOC concentration profiles a) in 2008- 2011 and b) in 2016.**

**Table 2:** Soil depth (cm) and soil surface coverages (%) of ericoid shrubs, mosses, grasses, and non-vegetative surface on the different measurement pits in 2016.

| Pit | Soil depth (cm) | Ericoid shrubs (%) | Mosses (%) | Grasses (%) | Non-vegetative surface (%) |
|-----|-----------------|--------------------|------------|-------------|----------------------------|
| 1 | 50 | 25 | 10 | ¾ | 65 |
| 2 | 60 | 5 | ¾ | 5 | 90 |
| 3 | 80 | 25 | 20 | ¾ | 55 |
| 4 | 130 | 15 | 30 | 18 | 37 |
| 5 | 160 | 7 | 2 | 8 | 83 |





**Figure 4: Isoprene and individual monoterpene (a) and sesquiterpene (b) concentrations (µg m⁻³) from the different soil horizons (H (N=52), A (N=65), B (N=65), and C (N=65)) in 2016. Concentrations are means and error bars are standard error of the whole data for each soil horizon. SQT1 was not identified.**







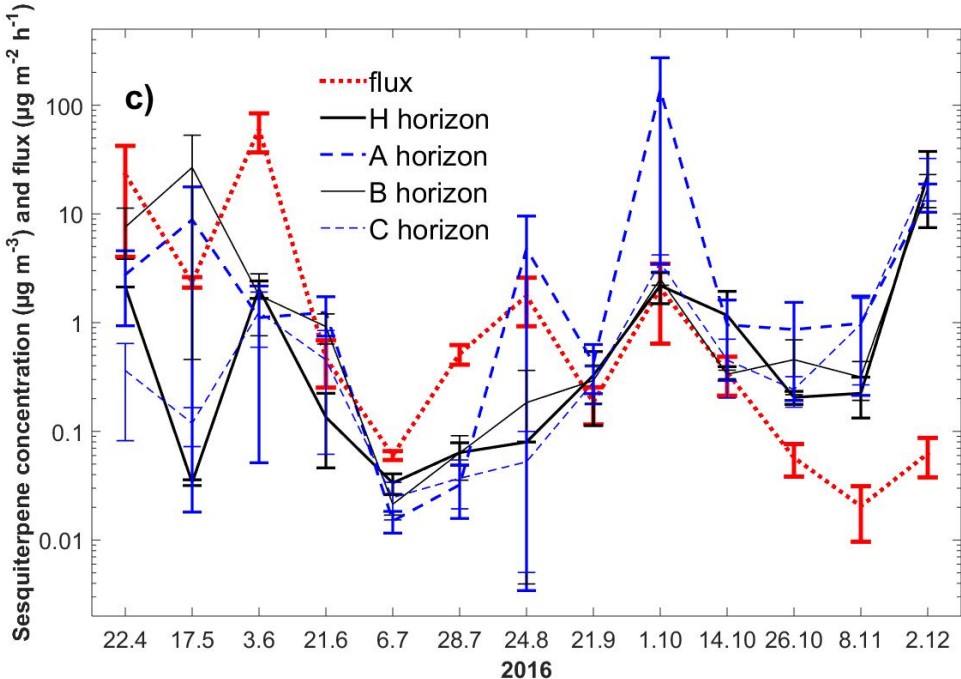

**Figure 5:** The mean a) isoprene, b) monoterpene, and c) sesquiterpene fluxes (µg m$^{-2}$ h$^{-1}$) from the forest floor and concentration (µg m$^{-3}$) from each soil horizon from April to December in 2016. Error bars are standard error of the four (H-horizon) or five (A-, B-, and C-horizon) gas collectors.

**Table 3:** The soil depth (cm), the mean monoterpene, sesquiterpene, oxygenated VOCs (C4–C15 alcohols, carbonyls and acetates, methyl-2/3-furoates and α-pinene oxide) and CO$_2$ fluxes above the soil surface (µg m$^{-2}$ h$^{-1}$), chamber temperature (°C), soil temperature (°C, A-horizon), and soil water content (m$^3$ m$^{-3}$, A-horizon) from the measurement pits. Values are means (S.E.) of the whole dataset in 2016 (N= 6–13). The effect of soil horizon on fluxes and environmental conditions was tested with the Kruskal-Wallis test (p<0.050). Significant differences between the pits are indicated with different letters (Kruskal-Wallis test; p<0.050).

| Pit | Soil depth cm | Monoterpenes µg m$^{-2}$ h$^{-1}$ | Sesquiterpenes µg m$^{-2}$ h$^{-1}$ | OVOC µg m$^{-2}$ h$^{-1}$ | CO$_2$ flux µg m$^{-2}$ h$^{-1}$ | Chamber temperature C° | Soil temperature C° | Soil water content m$^3$ m$^{-3}$ |
|---|---|---|---|---|---|---|---|---|
| 1 | 50 | 49.05[a] (30.30) | 4.56[a] (2.80) | 5.12[a] (3.72) | 0.15[a] (0.02) | 10.82[a] (2.28) | 8.29[a] (1.27) | 0.14[a] (0.01) |
| 2 | 60 | 19.71[a] (7.25) | 6.34[a] (5.50) | 1.84[a] (0.67) | 0.13[ac] (0.04) | 11.08[a] (1.99) | 8.67[a] (1.54) | 0.12[a] (0.09) |
| 3 | 80 | 20.73[a] (4.80) | 11.16[a] (10.91) | 0.59[a] (0.14) | 0.03[b] (0.02) | 9.33[a] (2.77) | 8.39[a] (1.32) | 0.34[b] (0.06) |
| 4 | 130 | 26.09[a] (8.72) | 8.08[a] (7.63) | 0.61[a] (0.26) | 0.08[c] (0.01) | 10.45[a] (3.04) | 7.72[a] (1.51) | 0.30[b] (0.02) |
| 5 | 160 | 61.90[a] (36.95) | 0.67[a] (0.25) | 0.96[a] (0.33) | 0.21[a] (0.07) | 12.24[a] (2.40) | 6.58[a] (1.20) | 0.78[c] (0.12) |





**Table 4:** Pearson correlation between the total monoterpene and sesquiterpene concentrations and soil temperature (°C) and water content (m$^3$ m$^{-3}$) from the organic and the B-horizon in 2008–2011 and from the different soil horizons in 2016. The significance level of p<0.100 (o), p<0.050 (*), p<0.010 (**), p<0.001 (***)) was used. VOC concentrations were measured from the different gas collectors in 2008–2011 and in 2016 (Fig. 1a and 1b).

| Year | Horizon | Correlation coefficient | N | P value | Correlation coefficient | N | P value |
|---|---|---|---|---|---|---|---|
| | Soil temperature C° | Monoterpene concentration µg m$^{-3}$ | | | Sesquiterpene concentration µg m$^{-3}$ | | |
| 2008–2011 | H | **0.35** | 49 | 0.007** | **0.29** | 34 | 0.049* |
| 2008–2011 | B | **0.46** | 51 | 0.0004*** | 0.16 | 36 | 0.17 |
| 2016 | H | -0.01 | 52 | 0.52 | -0.32 | 52 | 0.97 |
| 2016 | A | **0.33** | 65 | 0.005** | 0.02 | 65 | 0.44 |
| 2016 | B | -0.21 | 65 | 0.98 | -0.27 | 65 | 0.98 |
| 2016 | C | -0.17 | 65 | 0.89 | -0.32 | 65 | 0.99 |
| | Soil water content m$^3$ m$^{-3}$ | Monoterpene concentration µg m$^{-3}$ | | | Sesquiterpene concentration µg m$^{-3}$ | | |
| 2008–2011 | H | -0.13 | 49 | 0.81 | -0.30 | 34 | 0.96 |
| 2008–2011 | B | -0.35 | 51 | 0.99 | -0.43 | 36 | 0.99 |
| 2016 | H | -0.01 | 52 | 0.53 | -0.11 | 52 | 0.75 |
| 2016 | A | -0.09 | 65 | 0.76 | **0.23** | 65 | 0.04* |
| 2016 | B | -0.09 | 65 | 0.75 | 0.03 | 65 | 0.42 |
| 2016 | C | -0.09 | 65 | 0.74 | **0.19** | 65 | 0.098o |

**Table 5:** Annual mean isoprenoid concentrations (S.E., µg m$^{-3}$) in soil. Note that the year 2008 consists of only one sampling in November. BDL means below detection limit of the VOC quantification.

| µg m$^{-3}$ | 2008 1 sampling | | 2009 9 samplings | | 2010 3 samplings | | 2011 4 samplings | |
|---|---|---|---|---|---|---|---|---|
| | -5 cm | -17cm | -5 cm | - 17cm | -5 cm | - 17cm | -5 cm | - 17cm |
| isoprene | 0.01 | 4x10$^{-3}$ | 0.02 (0.01) | 0.01 (2x10$^{-3}$) | 0.06 (0.01) | 0.03 (3x10$^{-3}$) | 0.02 (0.02) | 0.01 (2x10$^{-3}$) |
| methyl butenol | 4x10$^{-3}$ | 3x10$^{-3}$ | 0.01 (0.01) | 3x10$^{-3}$ (1x10$^{-3}$) | 0.01 (3x10$^{-3}$) | 0.02 (0.01) | 0.01 (2x10$^{-3}$) | 4x10$^{-3}$(1x10$^{-3}$) |
| **Monoterpenes** | | | | | | | | |
| α-pinene | 0.63 | 0.60 | 1.88 (0.92) | 2.35 (0.99) | 1.37 (0.72) | 1.61 (1.13) | 2.88 (0.80) | 1.80 (0.54) |
| camphene | 0.02 | 0.02 | 0.07 (0.02) | 0.11 (0.06) | 0.06 (0.03) | 0.04 (0.02) | 0.04 (0.01) | 0.04 (0.02) |
| ß-pinene | 0.04 | 0.04 | 0.35 (0.24) | 0.26 (0.10) | 0.07 (0.03) | 0.43 (0.35) | 0.14 (0.04) | 0.11 (0.05) |
| Δ-3-carene | 0.84 | 0.67 | 2.56 (0.79) | 1.42 (0.35) | 1.34 (0.77) | 1.07 (0.70) | 2.92 (0.92) | 2.14 (0.61) |




| | | | | | | | |
|---|---|---|---|---|---|---|---|
| p-cymene | 0.06 | 0.04 | 0.07 (0.02) | 0.06 (0.01) | 0.09 (0.03) | 0.03 (0.01) | 0.06 (0.02) | 0.04 (0.01) |
| 1,8-cineol | BDL | BDL | BDL | BDL | 0.02 ($3\times10^{-3}$) | 0.01 ($1\times10^{-3}$) | $3\times10^{-3}$ ($1\times10^{-3}$) | $3\times10^{-3}$ ($1\times10^{-3}$) |
| limonene | 0.07 | 0.05 | 0.20 (0.05) | 0.16 (0.06) | 0.21 (0.05) | 0.11 (0.03) | 0.25 (0.08) | 0.18 (0.05) |
| terpinolene | BDL | $4\times10^{-3}$ | 0.12 (0.07) | 0.12 (0.03) | 0.19 (0.13) | 0.10 (0.07) | 0.04 (0.01) | 0.03 (0.01) |
| linalool | BDL | BDL | $3\times10^{-3}$ | 0.04 (0.02) | 0.11 | 0.01 | 0.01 (0.01) | $3\times10^{-3}$ |
| myrcene | BDL | BDL | BDL | BDL | BDL | BDL | 0.01 ($1\times10^{-3}$) | 0.01 ($2\times10^{-3}$) |
| nopinone | BDL | BDL | 0.01 ($5\times10^{-3}$) | 0.02 (0.01) | 0.02 ($3\times10^{-3}$) | 0.01 ($2\times10^{-3}$) | $4\times10^{-3}$ ($1\times10^{-3}$) | $4\times10^{-3}$ ($2\times10^{-3}$) |
| bornylacetate | BDL | 0.001 | 0.05 (0.02) | 0.21 (0.10) | 0.02 (0.01) | 0.01 ($2\times10^{-3}$) | 0.01 (0.01) | $3\times10^{-3}$ ($1\times10^{-4}$) |
| **Total monoterpenes** | 1.66 | 1.43 | 4.85 (1.83) | 4.30 (1.44) | 3.18 (1.60) | 3.18 (2.14) | 6.34 (1.84) | 4.35 (1.27) |
| **Sesquiterpenes** | | | | | | | | |
| longicyclene | BDL | BDL | 0.05 (0.02) | 0.16 (0.06) | 0.11 (0.05) | 0.01 ($2\times10^{-3}$) | 0.01($1\times10^{-3}$) | 0.01 ($2\times10^{-3}$) |
| iso-longifolene | $3\times10^{-3}$ | $4\times10^{-3}$ | 0.04 (0.01) | 0.17 (0.07) | 0.02 (0.01) | 0.01 ($2\times10^{-3}$) | $4\times10^{-3}$ ($1\times10^{-3}$) | $3\times10^{-3}$ ($1\times10^{-3}$) |
| ß-caryophyllene | BDL | BDL | 0.07 (0.02) | 0.21 (0.08) | 0.08 (0.05) | 0.05 (0.01) | 0.03 (0.01) | 0.02 (0.01) |
| aromadendrene | BDL | BDL | 0.06 (0.02) | 0.26 (0.11) | 0.04 (0.02) | 0.01 (0.01) | 0.06 (0.04) | 0.08 |
| α-humulene | BDL | BDL | 0.06 (0.02) | 0.21 (0.10) | 0.02 (0.002) | 0.01 (0.01) | $1\times10^{-3}$ ($3\times10^{-4}$) | 0.01 (0.01) |
| ß-farnesene | BDL | BDL | 0.09 (0.03) | 0.38 (0.11) | 0.03 (0.003) | 0.01 ($2\times10^{-3}$) | 0.01 ($2\times10^{-3}$) | 0.01 ($2\times10^{-3}$) |
| **Total sesquiterpenes** | 0.03 | $4\times10^{-3}$ | 0.28 (0.08) | 1.03 (0.36) | 0.22 (0.08) | 0.07 (0.02) | 0.06 (0.02) | 0.04 (0.01) |

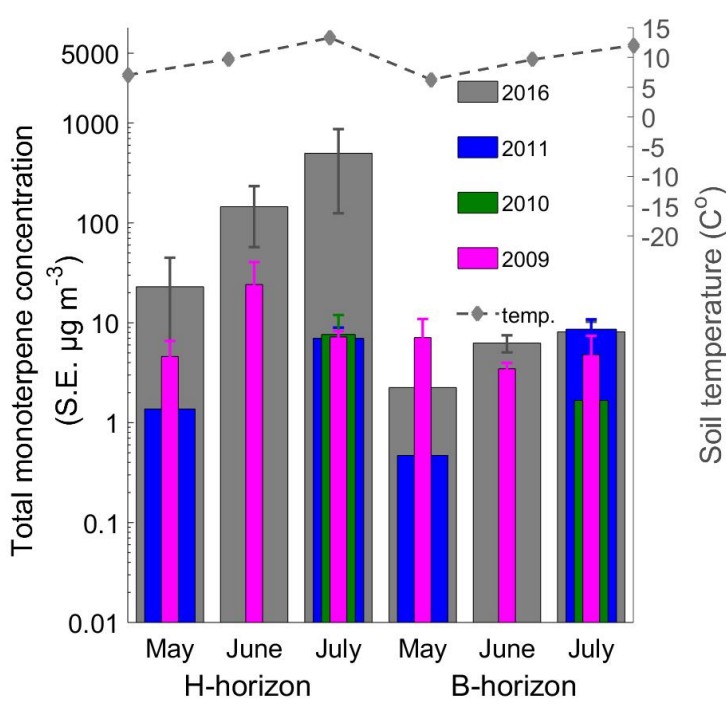



**Figure 6:** **The monthly mean monoterpene concentration (µg m$^{-3}$) and soil temperature (C°) for the H-horizon and the B-horizon during the summer months in 2009- 2011 and in 2016. Error bars are standard error of the three gas collectors in 2009, 2010, and 2011, and four (H-horizon) or five (B-horizon) gas collectors in 2016.**

**Appendix**

**Table A1:** Measurement depths (cm) of the different soil horizons (H, A, B, and C) from eight measurement pits. The campaign measurements were made from three soil pits in 2008- 2011 and from five pits in 2016.

| | | | | | | | | | |
|---|---|---|---|---|---|---|---|---|---|
| **2008–2011** | | | | | | | | | |
| **Pit 1** | | **Pit 2** | | **Pit 3** | | | | | |
| Horizon | Depth | Horizon | Depth | Horizon | Depth | | | | |
| H | -5 | H | -5 | H | -5 | | | | |
| B | -17 | B | -17 | B | -17 | | | | |
| | | | | **2016** | | | | | |
| **Pit 1** | | **Pit 2** | | **Pit 3** | | **Pit 4** | | **Pit 5** | |
| Horizon | Depth | Horizon | Depth | Horizon | Depth | Horizon | Depth | Horizon | Depth |
| H | -2 | | | H | -3 | H | -3 | H | -2 |
| A | 2 | A | 1 | A | 7 | A | 3 | A | 5 |
| B | 9 | B1 | 13 | B | 29 | B1 | 15 | B1 | 20 |
| | | B2 | 35 | | | B2 | 27 | B2 | 33 |
| C | 23 | C | 63 | C | 62 | C1 | 50 | C | 57 |
| | | | | | | C2 | 97 | | |

**Table A2:** Measurement months for VOC concentrations measurements in 2008- 2011 period and for VOC concentration and chamber flux measurements in 2016.

| Year | Measurement months | Sampling times |
|---|---|---|
| 2008 | November | 1 |
| 2009 | April, May, June, July, October, November, and December | 1 |
| | September | 2 |
| 2010 | July, August, and November | 1 |
| 2011 | May, June, August, and October | 1 |
| 2016 | April, May, June, November, and December | 1 |
| | July, August, September, and October | 2 |




**Table A3:** The detection limit of measured VOCs in concentration (µg m$^{-3}$) and flux measurements (µg m$^{-2}$ h$^{-1}$). Compounds marked with ($^{\dagger}$) are only tentatively identified and quantified.

| Compound | Concentration 2008–2011 | Concentration 2016 | Flux | Compound | Concentration 2016 | Flux |
|---|---|---|---|---|---|---|
| isoprene | 5x10$^{-3}$ | 5x10$^{-3}$ | 0.01 | **OVOCs** | | |
| 2-methyl-3-buten-2-ol | 2x10$^{-3}$ | | | geraniol | 4x10$^{-3}$ | 0.12 |
| **Monoterpenes** | | | | methyl-12-furoate | 1x10$^{-3}$ | 0.01 |
| α-pinene | 1x10$^{-3}$ | 0.03 | 0.18 | α-bisabolol | 0.01 | 0.04 |
| camphene | 1x10$^{-3}$ | 0.02 | 0.01 | verbenone | 0.01 | 0.02 |
| Δ-3-carene | 2x10$^{-3}$ | 0.06 | 0.06 | nuciferol | 2x10$^{-4}$ | 0.01 |
| ß-pinene | 2x10$^{-3}$ | 0.01 | 0.01 | methy-3-furoate | 0.01 | 0.02 |
| myrcene | 2x10$^{-3}$ | 0.01 | 0.01 | 1-butanol | 0.06 | 0.18 |
| 1,8-cineol | 3x10$^{-3}$ | 0.01 | 0.01 | isopropanol | 2x10$^{-3}$ | 0.01 |
| linalool | 0.01 | 0.01 | 0.32 | 2-butanone | 0.02 | 0.04 |
| limonene | 2x10$^{-3}$ | 5x10$^{-3}$ | 0.10 | 1-penten-3-ol | 2x10$^{-3}$ | 5x10$^{-3}$ |
| p-cymene | 1x10$^{-3}$ | 0.02 | 0.02 | 1-pentanol | 0.01 | 0.01 |
| terpinolene | 0.01 | 2x10$^{-3}$ | 2x10$^{-3}$ | 2-methyl-2-buten-1-ol | 1x10$^{-3}$ | 2x10$^{-3}$ |
| **Sesquiterpenes** | | | | butyl acetate | 1x10$^{-3}$ | 5x10$^{-3}$ |
| bornylacetate | 3x10$^{-3}$ | 2x10$^{-3}$ | 4x10$^{-3}$ | cis-3-hexen-1-ol | 0.01 | 0.02 |
| longicyclene | 1x10$^{-3}$ | 1x10$^{-3}$ | 2x10$^{-3}$ | trans-3-hexen-1-ol | 0.01 | 0.01 |
| isolongifolene | 1x10$^{-3}$ | 2x10$^{-3}$ | 5x10$^{-3}$ | trans-2-hexen-1-ol | 2x10$^{-3}$ | 4x10$^{-3}$ |
| ß-caryophyllene | 0.01 | 0.01 | 0.01 | 1-hexanol | 1x10$^{-3}$ | 0.01 |
| aromadendrene$^{\dagger}$ | 3x10$^{-3}$ | 0.01 | 5x10$^{-3}$ | cis-2-hexen-1-ol | 0.01 | 0.02 |
| α-gurjunene | | 0.01 | 1.48 | 1-octen-3-ol | 2x10$^{-3}$ | 0.03 |
| α-humulene | 4x10$^{-3}$ | 2x10$^{-3}$ | 0.09 | 6-methyl-5-heptene-2-one | 5x10$^{-3}$ | 0.03 |
| ß-farnesene | 2x10$^{-3}$ | 0.02 | 0.02 | cis-3-hexenyl acetate | 0.03 | 0.06 |
| isocaryophyllene$^{\dagger}$ | | 4x10$^{-3}$ | 0.01 | hexyl acetate | 0.01 | 0.012 |
| SQT1$^{\dagger}$ | | 0.01 | 0.02 | trans-2-Hexenyl acetate | 3x10$^{-3}$ | 0.005 |
| α-buinesene$^{\dagger}$ | | 1x10$^{-3}$ | 0.02 | α-pinenepxide | 0.01 | 0.01 |
| γ-muurolene$^{\dagger}$ | | 2x10$^{-3}$ | 1x10$^{-3}$ | | | |
| α-bisabolene$^{\dagger}$ | | 3x10$^{-4}$ | 1x10$^{-3}$ | | | |
| β-himachalene$^{\dagger}$ | | 1x10$^{-3}$ | 3x10$^{-3}$ | | | |
| α-muurolene$^{\dagger}$ | | 1x10$^{-3}$ | 0.01 | | | |
| Δ-cadinene | | | 0.15 | | | |




**Table A4:** Isoprene, monoterpenes, sesquiterpenes and oxygenated VOC concentrations, means (S.E. µg m⁻³) of the different soil horizons (H (N=52), A (N=65), B (N=65), and C (N=65)) in 2016. Concentrations are means (S.E.) and medians of the whole data (BDL = below the detection limit of the VOC quantification). The effect of soil horizon on concentrations was tested with the Kruskal-Wallis test (p<0.050). Significant differences in concentrations between the horizons are indicated with different letters (Kruskal-Wallis test; p<0.050). Compounds marked with (†) are only tentatively identified and quantified.

| Concentration | H | A | B | C |
|---|---|---|---|---|
| isoprene | 0.02$^a$ (2x10$^{-3}$) 0.01 | 0.04$^a$ (0.03) 0.01 | 0.02$^a$ (2x10$^{-3}$) 0.01 | 0.02$^a$ (3x10$^{-3}$) 0.01 |
| α-pinene | 320.4$^a$ (135.8) 28.0 | 71.0$^a$ (22.5) 5.0 | 108.6$^b$ (103.6) 1.6 | 6.7$^b$ (2.2) 1.5 |
| camphene | 8.8$^a$ (2.6) 0.7 | 1.4$^a$ (0.4) 0.1 | 0.6$^b$ (0.6) 0.03 | 0.3$^b$ (0.1) 0.01 |
| Δ-3-carene | 58.8$^{ac}$ (22.8) 4.3 | 16.9$^a$ (5.2) 2.6 | 30.2$^{bc}$ (29.5) 1.2 | 2.8$^b$ (0.8) 1.3 |
| ß-pinene | 14.7$^a$ (4.4) 1.2 | 2.4$^b$ (1.2) 0.2 | 0.4$^c$ (0.3) 0.1 | 0.3$^c$ (0.1) 0.1 |
| myrcene | 18.1$^a$ (8.0) 1.9 | 1.7$^b$ (0.5) 0.1 | 1.7$^c$ (1.0) 0.05 | 0.7$^c$ (0.5) 0.04 |
| 1,8-cineol | 0.08$^a$ (0.04) 0.02 | 0.3$^a$ (0.2) 0.04 | 0.2$^a$ (0.1) 0.02 | 0.1$^a$ (0.03) 0.01 |
| linalool | 1.4$^a$ (1.0) 0.1 | 1.5$^a$ (0.6) 0.1 | 1.1$^a$ (0.6) 0.1 | 0.6$^a$ (0.3) 0.1 |
| limonene | 16.4$^{ab}$ (7.4) 0.7 | 3.5$^a$ (1.0) 1.0 | 2.4$^{ab}$ (1.1) 0.6 | 1.9$^b$ (1.0) 0.4 |
| p-cymene | 1.8$^a$ (1.4) 0.1 | 0.4$^{ab}$ (0.1) 0.1 | 0.2$^{ab}$ (0.1) 0.1 | 0.2$^b$ (0.04) 0.1 |
| terpinolene | 3.1$^a$ (1.4) 0.2 | 0.4$^{ab}$ (0.2) 0.03 | 0.4$^{bc}$ (0.4) 0.02 | 0.05$^c$ (0.02) 0.01 |
| **Sum of monoterpenes** | **422.2$^a$ (157.9) 35.6** | **96.0$^a$ (28.8) 9.7** | **144.6$^b$ (133.7) 4.2** | **12.8$^b$ (3.7) 4.1** |
| bornylacetate | 0.01$^a$ (0.01) 0.01 | 0.1$^a$ (0.1) 0.01 | 0.01$^a$ (1x10$^{-3}$) 4x10$^{-3}$ | 0.01$^a$ (1x10$^{-3}$) 4x10$^{-3}$ |
| longicyclene | 0.02$^a$ (0.02) 0.01 | 0.01$^a$ (5x10$^{-3}$) 3x10$^{-3}$ | 5x10$^{-3a}$ (1x10$^{-3}$) 3x10$^{-3}$ | 0.01$^a$ (3x10$^{-3}$) 2x10$^{-3}$ |
| isolongifolene | 0.3$^a$ (0.3) 0.01 | 0.04$^a$ (0.03) 0.02 | 0.05$^a$ (0.02) 0.03 | 0.1$^a$ (0.03) 0.05 |
| ß-caryophyllene | 0.1$^a$ (0.1) 0.02 | 0.4$^a$ (0.2) 0.02 | 0.6$^a$ (0.4) 0.02 | 0.2$^a$ (0.2) 0.03 |
| aromadendrene$^†$ | 0.02$^a$ (5x10$^{-3}$) 0.02 | 0.01$^a$ 0.01 | 0.03$^a$ (0.01) 0.03 | 0.03$^a$ (0.01) 0.01 |
| α-gurjunene | 3.4$^a$ (2.4) 0.1 | 25.7$^a$ (22.9) 0.2 | 3.1$^a$ (1.3) 0.2 | 3.0$^a$ (1.5) 0.1 |
| α-humulene | 0.4$^a$ (0.2) 5x10$^{-3}$ | 2.9$^a$ (1.6) 0.3 | 5.3$^a$ (4.2) 0.3 | 0.6$^a$ (0.3) 0.01 |
| ß-farnesene | 0.4$^a$ (0.3) 0.1 | 0.03$^a$ (0.01) 0.03 | 1.9$^a$ (1.5) 0.03 | 0.4$^a$ (0.4) |
| isocaryophyllene$^†$ | 0.01$^a$ (3x10$^{-3}$) 0.01 | 0.01$^a$ (3x10$^{-3}$) 0.01 | 3x10$^{-3a}$ (2x10$^{-3}$) 0.01 | 0.01$^a$ (1x10$^{-3}$) |
| SQT1$^†$ | 0.03$^a$ (0.01) 0.03 | 0.02$^a$ (4x10$^{-3}$) 0.02 | 0.04$^a$ (3x10$^{-3}$) 0.04 | 0.02$^a$ 0.02 |
| α-buinesene$^†$ | 0.03$^a$ (0.02) 0.01 | 0.5$^a$ (0.5) 0.02 | 0.1$^a$ (0.05) 4x10$^{-3}$ | 0.3$^a$ (0.1) 0.2 |
| γ-muurolene$^†$ | 0.02$^{ab}$ (0.02) 0.02 | 0.01$^a$ (0.01) 0.01 | 4x10$^{-3b}$ (1x10$^{-3}$) 3x10$^{-3}$ | 0.01$^{ab}$ (4x10$^{-3}$) 5x10$^{-3}$ |
| α-bisabolene$^†$ | 0.01$^a$ (0.01) 0.01 | 0.5$^a$ (0.3) 0.02 | 0.3$^a$ (0.3) 0.01 | 0.3$^a$ (0.2) 0.2 |
| β-himachalene$^†$ | 0.2$^a$ (0.1) 0.02 | 1.3$^a$ (1.0) 0.3 | 0.3$^a$ (0.1) 0.2 | 0.4$^a$ (0.2) 0.2 |
| α-muurolene$^†$ | 0.07$^a$ (0.04) 0.01 | 0.1$^a$ (0.03) 0.05 | 0.1$^a$ (0.03) 0.1 | 0.03$^a$ (0.01) 0.02 |





| | | | | |
|---|---|---|---|---|
| Δ-cadinene[†] | 0.01$^a$ (4x10$^{-3}$) 2x10$^{-3}$ | 0.1$^a$ (0.1) 0.01 | 0.01$^a$ (2x10$^{-3}$) 3x10$^{-3}$ | 0.01$^a$ (0.01) 4x10$^{-3}$ |
| **Sum of sesquiterpenes** | **2.4$^a$ (1.4) 0.2** | **15.1$^a$ (12.1) 0.2** | **4.3$^a$ (2.0) 0.3** | **2.1$^a$ (0.9) 0.2** |
| geraniol | 0.1$^a$ (0.1) 0.02 | 2.3$^a$ (2.3) 0.01 | 0.1$^a$ (0.03) 0.01 | 0.2$^a$ (0.1) 0.01 |
| methyl-2-furoate | 5x10$^{-3a}$ (1x10$^{-3}$) 4x10$^{-3}$ | 0.1$^a$ (0.1) 3x10$^{-3}$ | 5x10$^{-3a}$ (1x10$^{-3}$) 3x10$^{-3}$ | 4x10$^{-3a}$ (1x10$^{-3}$) 4x10$^{-3}$ |
| α-bisabolol | 0.01$^a$ 0.01 | 0.1$^a$ (0.03) 0.04 | 0.03$^a$ (0.01) 0.02 | 0.03$^a$ (0.01) 0.02 |
| verbenone | 2.6$^a$ (1.7) 0.2 | 0.1$^{ab}$ (0.1) 0.1 | 0.03$^{ab}$ 0.03 | 0.02$^b$ (3x10$^{-3}$) 0.02 |
| nuciferol | 0.02$^a$ (0.01) 0.02 | 0.03$^a$ (0.01) 0.02 | 0.02$^a$ (4x10$^{-3}$) 0.01 | 0.02$^a$ (4x10$^{-3}$) 0.01 |
| methy-3-furoate | BDL | BDL | 0.02$^a$ 0.02 | BDL |
| 1-butanol | 1.9$^a$ (1.0) 0.3 | 9.0$^a$ (6.8) 0.4 | 2.3$^a$ (1.0) 0.4 | 1.7$^a$ (0.6) 0.5 |
| isopropanol | 0.01$^a$ (1x10$^{-3}$) 0.01 | 0.01$^a$ (2x10$^{-3}$) 0.01 | 0.02$^a$ (0.01) 0.01 | 0.02$^a$ (0.01) 0.01 |
| 1-butanone | 0.2$^a$ (0.1) 0.03 | 0.7$^a$ (0.5) 0.03 | 0.1$^a$ (0.1) 0.04 | 0.2$^a$ (0.1) 0.03 |
| penten-3-ol | 0.5$^a$ (0.2) 0.4 | 1.2$^a$ (0.8) 0.02 | 0.3$^a$ (0.1) 0.1 | 0.2$^a$ (0.1) 0.1 |
| 1-pentanol | 0.2$^a$ (0.07) 0.03 | 0.6$^b$ (0.3) 0.1 | 0.1$^{ab}$ (0.02) 0.04 | 0.1$^a$ (0.03) 0.03 |
| 2-methyl-2-buten-1-ol | 0.04$^a$ (0.03) 0.01 | 0.1$^a$ (0.02) 0.03 | 0.01$^a$ (3x10$^{-3}$) 0.01 | 0.02$^a$ (0.01) 0.01 |
| butyl acetate | 0.01$^a$ (2x10$^{-3}$) 0.01 | 0.1$^a$ (0.04) 0.01 | 0.01$^a$ (2x10$^{-3}$) 0.01 | 0.01$^a$ (2x10$^{-3}$) 0.01 |
| cis-3-hexen-1-ol | BDL | BDL | BDL | BDL |
| trans-3-hexen-1-ol | BDL | 0.02$^a$ 0.02 | 1.8$^a$ (1.8) 1.8 | 0.01$^a$ (0.01) 0.01 |
| trans-2-hexen-1-ol | 0.4$^a$ (0.3) 0.01 | 0.1$^a$ (0.02) 0.01 | 0.4$^a$ (0.3) 0.01 | 0.2$^a$ (0.1) 0.03 |
| 1-hexanol | 0.02$^a$ (5x10$^{-3}$) 0.01 | 0.04$^a$ (0.03) 0.01 | 0.1$^a$ (0.1) 0.01 | 0.02$^a$ (4x10$^{-3}$) 0.01 |
| cis-2-hexen-1-ol | 0.01$^a$ (1x10$^{-3}$) 0.01 | 0.03$^a$ (0.02) 0.01 | 0.02$^a$ (0.01) 0.01 | 0.1$^a$ (0.04) 0.02 |
| 1-octen-3-ol | 0.2$^{ab}$ (0.1) 0.03 | 1.4$^a$ (1.3) 0.03 | 1.3$^{ab}$ (1.3) 0.03 | 0.25$^b$ (0.1) 0.02 |
| 6-methyl-5-heptene-2-one | 2.5$^a$ (1.4) 0.1 | 0.5$^a$ (0.2) 0.1 | 0.2$^b$ (0.1) 0.04 | 0.2$^b$ (0.1) 0.03 |
| cis-3-hexenyl acetate | 1.2$^a$ (0.8) 0.1 | 1.2$^a$ (0.7) 0.1 | 1.0$^a$ (0.5) 0.1 | 1.9$^a$ (0.8) 0.1 |
| hexyl acetate | 0.03$^a$ (0.01) 0.04 | 1.1$^a$ (1.1) 0.03 | 0.05$^a$ (0.02) 0.03 | 0.02$^a$ (4x10$^{-3}$) 0.02 |
| trans-2-hexenyl acetate | 0.3$^a$ (0.3) 0.02 | 0.02$^a$ (0.01) 4x10$^{-3}$ | 0.4$^a$ (0.3) 5x10$^{-3}$ | 0.2$^a$ (0.2) 5x10$^{-3}$ |
| α-pineneoxide | 1.0$^a$ (0.7) 0.2 | 1.3$^a$ (0.6) 0.1 | 0.4$^a$ (0.2) 0.1 | 0.04$^a$ (0.03) 0.04 |
| **Sum of OVOCs** | **5.4$^a$ (2.5) 1.2** | **10.0$^a$ (6.7) 0.8** | **4.0$^a$ (1.9) 0.5** | **2.8$^a$ (1.1) 0.6** |



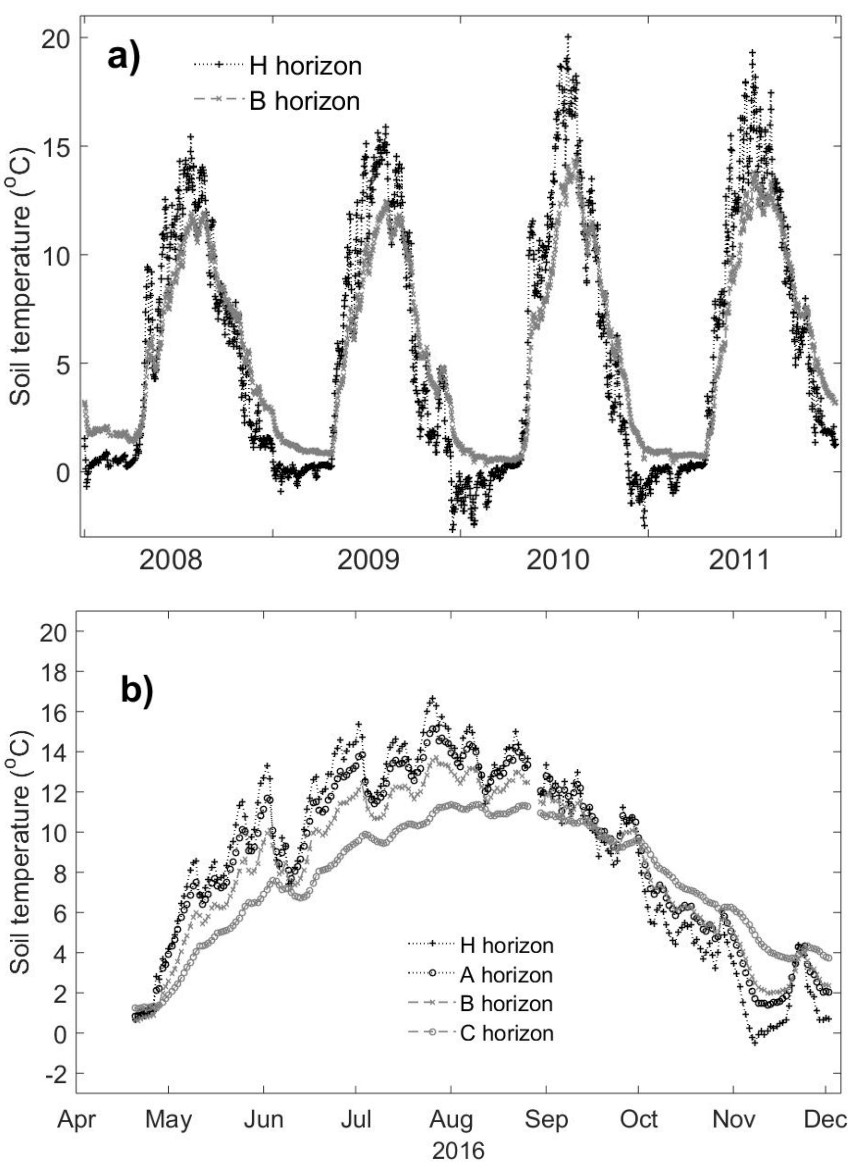

**Figure A1: a) Soil temperature (C°) measured at organic layer (0–5 cm) and B-horizon (10–28 cm) over the 2008-2011 period and b) soil temperature measured at the different soil horizons in 2016.**





**Figure A2: a) Soil water content (m³ m⁻³) measured at organic layer (0–5 cm) and B-horizon (10–28 cm) in years 2008- 2011 and b)
soil water content (m³ m⁻³) measured at the different soil horizons in 2016. High wintertime variation is explained the change from**
5   **liquid water to solid phase by freezing as TDR measurement method is highly sensitive for freezing.**




**Table A5**: Pearson correlations between the total monoterpene and sesquiterpene fluxes ($\mu g\ m^{-2}\ h^{-1}$) and concentrations ($\mu g\ m^{-3}$) in the organic and the A-horizons in the different measurement pits in 2016. The significance level of $p<0.100$ (o), $p<0.050$ (*), $p<0.010$ (**), $p<0.001$ (***)) was used.

| Concentration ($\mu g\ m^{-3}$) | | Monoterpene flux ($\mu g\ m^{-2}\ h^{-1}$) | | | Sesquiterpene flux ($\mu g\ m^{-2}\ h^{-1}$) | | |
|---|---|---|---|---|---|---|---|
| Pit | Horizon | Correlation | N | P value | Correlation | N | P value |
| 1 | H | -0.19 | 11 | 0.70 | 0.02 | 11 | 0.47 |
| 1 | A | -0.27 | 11 | 0.78 | -0.09 | 11 | 0.61 |
| 2 | A | 0.20 | 11 | 0.29 | **0.46** | 12 | 0.07° |
| 3 | H | -0.39 | 13 | 0.91 | **0.62** | 9 | 0.04* |
| 3 | A | -0.55 | 12 | 0.97 | **0.72** | 10 | 0.01** |
| 4 | H | **0.78** | 7 | 0.020* | -0.43 | 7 | 0.83 |
| 4 | A | -0.34 | 9 | 0.81 | -0.16 | 10 | 0.67 |
| 5 | H | -0.10 | 6 | 0.57 | 0.01 | 5 | 0.99 |
| 5 | A | **0.83** | 6 | 0.020* | 0.54 | 5 | 0.17 |