# Peer review of "Boreal forest soil is a significant and diverse source of volatile organic compounds"

_Biogeosciences, 2018_

## Referee Comment (RC1) · Anonymous Referee #1 · 22 Feb 2018

This manuscript is mainly focused on observations of belowground VOC concentrations at different soil depths and comparison with aboveground VOC exchange measurements by means of dynamic soil enclosures. The method of probing belowground soil gas concentrations has been applied for a bunch of other trace gases but not yet for VOC, which is innovative and thus the most appealing aspect of this manuscript.

The authors state that the observed belowground VOC concentrations in general were not directly coupled with forest floor VOC fluxes measured by means of dynamic enclosures applying VOC-free air as purging gas. In my opinion, this may have different reasons: (#1) for the enclosure VOC exchange measurements, zero air have been applied as purging gas. This way, an artificial concentration gradient is stablished that force trace gas emission and omits any deposition or bi-directional exchange to be

observed. Hence the above ground exchange measurements are not representative and cannot be transferred to real world conditions (or reflect observed belowground concentrations). On the other hand (#2), the authors did not simultaneously measure the aboveground ambient air VOC concentrations. Only based on the latter, one could infer any fluxes (or even directions: positive or negative) between soil and the atmosphere and estimate whether soil is a source or sink for VOC. The authors state that belowground and aboveground concentration (the latter from an earlier campaign) were "similar in magnitude". In this case, one would assume that both emission and uptake is possible (the authors may also refer to, e.g., Gut et al. (2002) for a theoretical background of calculations necessary for this kind of soil trace gas measurements). But most importantly (#3), I have concerns about the belowground measurement procedure or representativeness of respective VOC concentrations. The question is how the belowground VOC concentrations were derived/calculated? I understand that the authors were running their sampling system in a closed loop, having a "perfect" sink for VOC at the side of the Tenax adsorption tube, i.e., the air flow downstream of the adsorption tube will be depleted of VOC. On the other side of the sampling system (at the inlet of the collector) this VOC-depleted air creates an artificial gradient that forces VOC from soil-air to penetrate through the collector membrane. This happens very soon after the sample pump starts, as the VOC that have been accumulated during the 15 min sample breaks are transferred to and trapped by the adsorption tubes within a time span of about 1-2 minutes (sample flow of 100-150 ml, and collector volume of 150 ml in 2009-2011; no numbers given for the volume in 2016). In the residual time of the (4x) 15 min sampling period the authors applied an artificial VOC concentration gradient by flushing the collectors with VOC-free air. That means that the longer the sample interval the more VOC will be accumulated in the adsorption tube. If taking the total sample volume (6-9 l) into account to calculate the VOC concentration this will not represent the VOC concentrations prevailing in the soil air. Rather, this is a measure of how much VOC can penetrate through the collector membrane when an artificial VOC gradient (with zero VOC in the collector) is applied. Of course the VOC penetration rate

will, among others, depend on the soil VOC concentrations, but will by no means be representative of (or equilibrated with) the soil-air VOC absolute concentrations (compare Fig. 2). In fact, if the calculation is accounting for the total sampling volume, the derived concentrations will be much lower (with the concentrations inside the collector being lower as soon as the pump is on). Else: with very large sampling volumes of 6-9 liters one can assume breakthrough in the adsorption tubes to occur to some extend by some of the VOCs, which make things even more complicated.

What the authors measure is, to the most of their sampling procedure, the VOC transmission rate of their collector wall/membrane. This issue of the method/calculation of belowground concentration is especially critical for the sesquiterpenes, which are indeed expected to have relatively long equilibration times or lower Teflon diffusion rates, respectively. Else: even though a major fraction of this manuscript is reporting on sesquiterpenes, these compounds were not tested for collector permeability. As stated by the authors, the collector permeation rate can be assumed to be much lower than for the other compounds, as the penetration rate is, among others, dependent on size. Differences in transmission rates in Teflon can easily span several orders of magnitude, please see, e.g.: https://www.chemours.com/KIV/zh_CN/assets/downloads/Chemours_Teflon_FEP_Film_Tech_Bulletin_K26942.pdf

The authors either should have used an online VOC analytical device with high temporal resolution to detect the short high concentration peak directly after they had started the sampling pump. Only this small volume of air (if at all) can reflect the soil-air concentrations, assuming that the collector air volume has reached equilibrium during the 15 min sampling breaks. Or they should have used an online VOC analyzer within a close loop sampling system, which does not interfere with (adsorb) VOCs. Please correct me if I am wrong.

Due to the issues #1-3, an interpretation of the results of the manuscript is indeed difficult to achieve, and any conclusions can only be of speculative nature. Concerning issue #1, one could state that using zero-air for purging the enclosures reveals a

measure of the potential soil VOC emission capacity. And a direct comparison with the belowground exchange measurements doesn't make sense anyhow, due to multiple reasons.

Concerning issue #2, the authors give a range of VOC concentrations measured in ambient air above this forest in an earlier campaign; and state that those were "similar in magnitude" as the observed belowground concentrations. This could be a fair projection, but only in case issue #3 could be solved.

Concerning issue #3, one could state that the observed belowground concentrations (and vertical profiles) are a (very)lower-bound estimate, but this is very much dependent on the individual compounds (diffusion characteristics). Then the vertical VOC concentration differences between the different soil depth/horizons could at least be discussed, rather than any of the absolute VOC concentrations.

In general, the way of data presentation/structure in the manuscript is sometimes not easy to follow or not precise, and the interpretation of the correlation analysis is kind of vague or speculative. In view of the issues presented above, the decoupling of the belowground VOC concentrations from the forest floor fluxes and the scarceness of correlations/gradients (e.g. for sesquiterpenes and OVOC) is not surprising. Even though the authors try to pin down potential dependencies (soil temperature and water content) by displaying tabulated statistical data, the data evaluation did not give a conclusive picture. The correlation analysis sometimes gives encouraging numbers for some soil layer horizons, or individual pits of those, but not for some others or adjacent soil layers, which doesn't add confidence in respective interpretations. In a recent paper, the authors already concluded from dynamic (zero-air) enclosure measurements that belowground dynamics might not play a major role in isoprenoid exchange, but instead the litterfall is the most important factor triggering VOC emissions (Mäki et al., 2017); and with all the short-coming presented above this seems to be confirmed by the belowground vertical gradients of VOC concentration in this manuscript.

[Figure]

Concerning all issues above, I suggest that the authors consider reassessing their conclusions in respect with the critical points presented above (and below) and resubmit a new version.

The M&M section needs to describe more details.

- Before describing the collector permeability test (in section 2.2), the authors should first introduce/describe the innovative type of collectors used (as they did later in section 2.3.).

- What does "wet collector" mean (page 4, line 6)? How did you wet it? Did you apply humidified air?

- What is the meaning of "The beaks length . . ." (page 4, line 11)?

- Is the "sampling system" (page 4, line 22) the same as the "collector"? Otherwise I don't get it. What do the authors mean with ". . . with the pits" at the end of the same sentence? Connect the tubes with the pits? The term "within the pits" make more sense to me. Please clarify.

- "For aboveground sampling" (page 4, line 28:) can be misinterpreted, as you didn't do any aboveground measurements. I suggest to merge this and the follow-up sentence in a concise way.

- Permeability test: which concentrations did you use for the test (in the Fig. 2 it says "arbitrary units")? Were the concentrations inside the collector the same as outside ("at constant level"), as Fig. 2 lets assume?

- What is meant by "The possibility of creating a flux collectors that did not originate from the actual measured horizon" (page 5, line 4)?

- What is meant by "were closed between the samplings" (page 5, line 5). Closed in between the consecutive sampling intervals of one sample procedure (closed for 15 min during sampling brakes)? Or did the authors close the tubes when they finished

one complete sampling cycle?

- I got lost understanding the different pits versus investigated soil horizon designations (2 versus 4 in the different campaigns) versus soil depth (5 in table 2). From Table A1, I understand that the two soil depths (organic and mineral) investigated in 2008-2011 refer to horizons H & B in 2016. On page 8, line 22, the authors state "mineral soil (A- and B-horizons)". I suggest to shortly describe which layers of the 2018-2011 measurements refer to which in 2016 in the M&M section. What do negative numbers of soil depth mean in Table A1?

- The total sampling volume was 6-9 liters of air. Did the authors test any VOC break-through? 9 liters is much more than normally applied.

- Omit "0 cm being the surface of organic layer, not mineral soil" (page 5, line 17). What do the negative soil depth numbers in Table A1 mean? May be I didn't get the above.

- What is meant by ". . . installed in the vertical face interfacing with the undisturbed soil . . .." (page 5, line 27)?

- Page 6, line 4: in case that VOC-free air was used to flush the enclosures (as in Mäki et al. 2017) you should mention this here; as in this way, an artificial concentration gradient is produced that enhances trace gas emission and omits any deposition or bi-directional exchange rates to be observed.

- Please give a some more details of the enclosure system applied (instead of only citing your previous paper). Also state the basic calculation formulas here.

Other minor issues:

- Page 1, line 16: omit "the" in "during the two measurement campaigns"

- Page 8, line 3: the authors state that "Belowground VOC concentrations were dom-inated by monoterpenes and sesquiterpenes, but the monoterpene concentrations were mainly decoupled from forest floor monoterpene fluxes." Obviously also the SQT

and OVOC were "decoupled".

- Page 8, line 5: what is meant by "Belowground VOC concentrations in the vertical soil horizons". Suggest: "Belowground vertical gradients of VOC concentrations".

- Page 8, line 15: what is meant by ". . . when each soil horizon was tested separately"?

- Page 8, line 21: "Total monoterpene concentrations in organic soil were highest in late summer and in December": Comparing late summer (28.07., 24.8., 21.9.) with fall (1.10., 14.10., 26.10., 8.11., 2.12.), this is hard to tell. It is sometimes hard to follow what the authors exactly mean when discussing data in spring, early/late summer, autumn in the different chapters of the main text. Did they really plot means in Fig. 5, or media (error bars are not evenly distributed to the positive/negative direction)?

- Page 8, line 25: "Total sesquiterpene concentrations in mineral soil were clearly highest in spring, in early June, in late summer, and in October (Fig. 5)." Early June is still spring time. I can't see this general trend at all in Fig. 5c.

- Page 8, line 31: "There was no difference in VOC fluxes between measurement pits." I am not sure whether I got this right. It's hard to believe that all fluxes of all VOC (classes) were similar, due to the inhomogeneity of the forest floor mentioned.

- Page 9, line 3-4: I suggest: "In contrast to our hypothesis, the below ground vertical concentration profiles were not coupled to observed soil surface fluxes rates, ..."

- Page 9, line 5: "individual pits" is redundant.

- Page 9, line 12: "Confirming our third hypothesis that soil temperature and water content can be used to explain belowground VOC synthesis." Due to a lack of correlation with all other VOC classes versus soil horizons, I would not state that these results are confirming the third hypothesis. May be you can state that these individual correlations are in line with the hypothesis, but then you also have to mention that all other correlations fail to do so. Else: this sentence is missing its subject.

- Page 9, line 23: "The organic soil showed seasonal variation in 2011 and 2016 ... (Fig. 6)". As Fig. 6 only shows summer data: how can the authors claim that there are "seasonal variations?" Or did they mean inter-annual variations, or inter-campaign variations?

- Page 9, line 25: "Monoterpenes constituted almost 90% of the total VOC concentration, sesquiterpenes accounted for less than 10% between 2008 and 2011 (Table 5)." How does this VOC composition compare to (expected) ambient air data (in lack of own data, please give a general statement)?

- Section 3.4 ("Inter-annual variation"): the discussion on seasonal pattern is sometimes redundant (see section 3.1 and 3.2), but with different phrasings, e.g.: "Monoterpene concentrations in 2016 were highest in organic soil in summer, in October and in December, whereas seasonal variation was relatively small in mineral soil (in section 3.4)." versus "Total monoterpene concentrations in organic soil were highest in late summer and in December ... (section 3.2)".

- Page 12, line 19: "Belowground isoprenoid concentrations varied seasonally, and the highest concentrations were measured during summer and early autumn in 2009 and 2011, whereas high belowground concentrations monoterpene concentrations were measured in late summer, in October, and in December in 2016.". I think the authors should not compare the total isoprenoids (ISO, MT, SQT) in 2009-2011) with only the MT in 2006. What about the other isoprenoids (SQT, isoprene) and what about the OVOC in general?

- Page 13, line 27: "led to"

- Fig. 2: the x-axis has no units given

- Fig. 5: it seems that the x-axis has equi-distant steps for the different sample dates. I propose to use an absolute numeric time line (the sampling/breaks were not evenly distributed over time).

[Figure]

**BGD**

- Figure A2: any idea why the water content of A horizon (lying between the H and B horizon) is so much higher than all the others? Indeed, the soil water content can be quite inhomogeneous (e.g., by water channeling etc.). If only measured by one single sensor per soil depth, these measurements are not necessarily representative.

References:

Gut et al. (2002): NO emission from an Amazonian rain forest soil: Continuous measurements of NO flux and soil concentration. J. Geophys. Res., 107 (D20), 8057, doi:10.1029/2001JD000521.

Mäki et al. (2017): Contribution of understorey vegetation and soil processes to boreal forest isoprenoid exchange. Biogeosciences, 14 (5), 1055-1073, doi:10.5194/bg-14-1055-2017, 2017.

---

## Short Comment (SC1) · 28 Mar 2018

The manuscript bg-2018-22 describes 2 new setups to measure seasonal and depth dynamics of volatile organic compounds in a haplic podzol in a boreal forest (SMEAR II site, Finnland). The manuscript compares results measured with 2 methods and concludes about seasonality, which might be just caused by the differences in the methods. Additionally, the manuscript is written rather descriptive and general with a focus on atmospheric chemistry rather than biogeosciences. As it is, the manuscript might be better for publication in AMT or ACP. Instead of comparing the concentrations within

the soil profile to the flux from the surface into the atmosphere, I would like to read more about possible biogeochemical processes involved in the production of the individual compounds based on literature.

I will point out some additional references and ideas to change the focus more towards biogeosciences. In general the measurement of soil VOCs measured in depth profiles measured via TD-tubes and analysis by GC-MS is very challenging and unique, thus, I recommend the manuscript for publication. I just have problems to conclude about seasonality if 2 different methods have been applied and no pressure was measured. I recommend to focus rather in the dynamics within the soil depth profile rather than on the seasonality. More detailed comments for a revision are addressed bellow.

First of all, I have a problem with the term storages. It suggests that e.g. in plants isoprenoids are stored and released based on physico-chemical processes. While this might certainly be true for the top litter layer, there is strong evidence that microbes in soil can actively produce mono- and sesquiterpenes (e.g. Schulz and Dickschat 2007, Yamada et al., 2015) within their metabolism. Page 1,line 27: It is not really the high organic carbon content which results high VOC emissions from organic horizons, but rather the highest abundance and activity of autotrophic and heterotrophic microbes in that layer.

Page 2, line 1-16: This is a general intro for the relevance of VOCs in atmospheric chemistry. Given the focus of the manuscript on VOC dynamics within a soil profile, I would recommend to start here with the role of VOCs as infochemicals (Insam and Seewald, 2010 and Schulz and Dickschat, 2007). As stated already on page 11 line 31 ff., the VOCs produced in the soil profile differ from the VOCs released into the atmosphere. Thus they are not necessarily transported all the way up into the atmosphere and thus their role within the soil should be focused.

Page 2, line 12 ff.: Diffusion also is dependent on soil moisture, not only soil temperature (see e.g. Skopp et al., 1990).

Page 2, line 22 ff.: A major result of snow cover is that the soil is isolated from the cold air temperatures and is not freezing. Thus, I agree that microbial processes might still be ongoing. However, given the fact that microbial metabolism is strongly correlated to soil temperature, which should be quite soil in winter, I think an enrichment effect is more likely. The snow acts as a lid of a static chamber.

Page 4, line 21 ff.: In both setups polytetrafluoroethylene (PTFE) tubes, which were closed on one side by a sintering method, have been used. I have problems to understand how the first method, which was applied from 2008 until 2011 to suck air out of the sintered tube with a pump, reflect "diffusion of gases to occur"(line 21 ff.). According to my knowledge a pump creates a pressure difference from the inner tube to the surrounding soil. Thus, the soil air was sucked into the tube and does not reflect natural conditions where molecular diffusion occurs. This was improved in the second setup in 2016, where air was circulated through the same tubes and the assumption of molecular diffusion for that data are more likely to be valid. Without a pressure measurement as e.g. Gut et al. 1998, I have problems to follow the assumption of molecular diffusion for the first setup. PTFE tubes can be manufactured with different volume density and thus the mesh of stainless steel is also important to prevent that the soil is changing the inner volume of the PTFE tubing. Thus, the volume density should be included in the method description.

Page 8, line 9 ff.: Highest sesquiterpene and OVOC concentrations in the A horizon should be discussed with respect to the difference in particle density of O and A horizon material. This impacts the overall water filled pore space and thus might explain your result. In general, it is expected to observe highest concentrations in the O horizon. Another point which is missing in the discussion is the potential of utilizing sesquiterpenes and OVOCs as microbial signaling in the A horizon.

Page 8, line 18 ff.: low oxygen availability does not necessarily results low aerobic microbial activity. Anaerobic microbes will be still active.

Page 9, line 3 ff.: I don't understand why the hypothesis surface VOC fluxes and below-ground VOC concentrations are similar was formulated? Wouldn't exactly the opposite be true? The surface VOC flux is dependent on the turbulent eddy diffusion coefficient, whereas the belowground VOC concentrations are dependent on the molecular diffusion coefficient. Since they are several orders of magnitude different, I would not expect that surface VOC fluxes and belowground VOC concentrations should follow the same trend/pattern.

Page 9, line 18 ff.: I have problems to follow interannual variability if 2 different methods have been applied.

Page 10, line 11 ff.: It is a kind of recapitulation to summarize results and discussion in 2 sentences. I would remove both and rather move them into the conclusion.

Page 10, line 16 ff.: I agree, but in the discussion section I want to read also why the monoterpene concentrations are highest in the organic horizon (not soil)? The pores in the organic layer are much larger than in the mineral soil. Thus, fungi, which need to grow hyphae from one particle to another are rather slow. Thus it is not surprising that on the other hand bacteria were found in e.g. Timonen et al. 2017 to be high abundant in the humus. It is known that bacteria can easily colonize particles in the organic horizon since most are mobile. Thus, you could interpret the production of e.g. 3-carene and camphene from fungi as active inhibition of the swarming and swimming motility of bacteria. Such findings have been published already (Schmidt et al., 2015). These findings suggest that the production of terpenoids is rather connected to microbial activity than on microbial abundance. I am sure that you can find much more correlations of your data to microbial processes.

Page 11, line 20 ff.: Just for curiosity, can you comment on the speciation into $\alpha$-, $\beta$-, $\gamma$- sesquiterpenes?

Page 11, line 31 ff.: I did not yet found the commonly reported functions of belowground VOCs (e.g. defense communication and signaling).

Page 13, line 29 ff.: I don't the reference for climate change fits for your manuscript. I am not an expert for snow cover, but as far as I know snow isolates the soil surface. Thus, if the snow in the future will not be present anymore, I would assume that the surface temperature of the soil should be colder.

Page 14, line 1: The sentence ...more research is needed I find too general. I think your manuscript shows some nice trends about VOCs in the soil depth profile which could be combined with existing literature to microbial processes. I can agree to ...more research is needed to combine soil VOCs to microbial processes.

Page 14, line 4: I find the conclusions rather short and just analyzing the temperature and moisture dynamics not very informative. I recommend discussing and concluding about microbial processes within the soil profile. Also you measured $CO_2$, but did not really talk about the correlation.

Minor comments: It is confusing to read about the thicknesses (page 3 line 24 ff.) which are not reflected in the horizon borders in Table 1. Also I am missing the E horizon, which might explain the differences.

Fig. 1: The scheme is rather a fast draft. I got especially lost following arrows which are not connected to a tube. Maybe for the non-expert reader it would be worth to include in the figure caption that the sintered version of the PTFE tube means that it is closed on one side? Also a lid of the glas bottle would help.

Fig. 2: Maybe I did not get it, but the arbitrary signals for dry and wet indicate a moisture effect. In case you used cps, it might be worth to think about a different way to plot the data to correct for that effect or mention it in the method section?

Fig. 3: The collectors in a) were installed in 5 and 17 cm, which are not the interface of the horizons. I suggest to focus on 2016 only and include the surface tube for flux measurement in b) plus a depth y-axis in cm. It is confusing to use the term "organic soil" and "mineral soil" while you speak about H-, A-, B-, and C-horizons.

Fig. 4: There seems to be a problem either of the graphic or my printer for some error bars. I would like to read what processes cause the large differences of a-pinene, 3-carene, linalool and limonene in the soil profile. It is also worth to think about a classification into different relationships of VOC concentration with depth (e.g. exponential vs. linear, etc.). The carbon content and microbial biomass should decrease exponentially with depth. Thus, if VOC concentrations follow a different pattern, e.g. a-gurjunene, a-humulene and b-himachalene it could indicate that their production is not linked to the storage in plant litter, but rather likely to microbes which are most abundant/active in a specific layer of soil (A-horizon).

Fig. 5: a) It is hard to explain the elevated isoprene around 01.10 for the A horizon. Wouldnt it make sense to finally conclude that predominantly there is no difference in isoprene concentration and flux except this single event?

Tab. 3: I recommend to plot $CO_2$ versus Sesquiterpene concentration and $CO_2$ versus Monoterpene concentration and discuss the contribution of autotrophs ($CO_2$ consumers) and heterotrophs ($CO_2$ producers), respectively.

---

## Author Comment (AC1) · 2 May 2018

The comment was uploaded in the form of a supplement:
https://www.biogeosciences-discuss.net/bg-2018-22/bg-2018-22-AC1-supplement.pdf

---

## Author Comment (AC2) · 2 May 2018

Dear Reviewers and the Editor,

We would like to express our appreciation for taking your time to evaluate this manuscript. We have considered the suggestions with great care and rand for your good and constructive comments which have improved the manuscript in many places. We have revised the manuscript based on the suggestions. Below we will respond to the reviewer's feedback (reviewer's suggestions numbered in black font and our response below each comment in blue font).

Reviewer's suggestions:

Reviewer #1: This manuscript is mainly focused on observations of belowground VOC concentrations at different soil depths and comparison with aboveground VOC exchange measurements by means of dynamic soil enclosures. The method of probing belowground soil gas concentrations has been applied for a bunch of other trace gases but not yet for VOC, which is innovative and thus the most appealing aspect of this manuscript. The authors state that the observed belowground VOC concentrations in general were not directly coupled with forest floor VOC fluxes measured by means of dynamic enclosures applying VOC-free air as purging gas. In my opinion, this may have different reasons: (#1) for the enclosure VOC exchange measurements, zero air have been applied as purging gas. This way, an artificial concentration gradient is stablished that force trace gas emission and omits any deposition or bi-directional exchange to be observed. Hence the above ground exchange measurements are not representative and cannot be transferred to real world conditions (or reflect observed belowground concentrations).

We welcome the referee comments on the measurement methods as they improve opportunities to develop our measurement systems in the future. We have done changes in the text in order to clarify the method and believe it is now much clearer.

1) The enclosure measurements have originally been developed for inert gases and significantly larger fluxes (e.g. for $CO_2$ or $CH_4$). Also with reactive trace gases and small fluxes, the methods have been already applied in several papers before this manuscript. VOC flux measurements are commonly performed by using VOC-free air as a purging gas (e.g., Hakola et al 2006, Aalto et al., 2014, Aaltonen et al., 2011, 2013, Hellen et al., 2006, Mäki et al., 2017). Employment of dynamic chambers to measure trace gas fluxes is a reliable method, since flushing of the chamber headspace helps to avoid pressure and gas concentration changes inside the soil. Frequently, MnO2-scrubber and active carbon filters are being used between the supply air pump and the chamber to filter the air that is pumped into the chamber headspace. With concentrations that are often close to detection limits of the instruments, the most important requirements for the replacement air is a stable concentration with a steady flow rate. Employment of dynamic chambers to measure trace gas fluxes is considered as a reliable method, since flushing of the chamber headspace helps to avoid pressure and gas concentration changes inside the soil, and the artificial concentration gradient can be mainly avoided by sufficient flushing the chamber before the sampling period to achieve a steady-state VOC concentration inside the closed chamber. The fluxes are determined from concentration change during the chamber closure and calculated using mass balance equations, which eliminates the effect of an artificial concentration gradient. The dynamic enclosure method was previously tested in field conditions using standard gas with known VOC concentrations and quadrupole-PTR-MS (Kolari et al., 2012). The chamber system underestimates the artificially generated VOC emission rates at varying degree: for isoprene, monoterpene and many oxygenated VOCs the underestimation is 5-30%. The most uncertainties originate from adsorption of VOCs to moist or reactive surfaces, which are unavoidable when the enclosure contains living plant material.

Due to the reasons listed above, we believe that the dynamic enclosure measurements as implemented here are as accurate and representative as one can reasonably have, when the number of sampling points and the spatial placement is not unlimited and when the dynamics of field conditions are causing random noise to the measurements. We added description of the uncertainties and the reference Kolari et al., 2009 to the manuscript on Page 7, lines 8-12.

2) The authors did not simultaneously measure the aboveground ambient air VOC concentrations. Only based on the latter, one could infer any fluxes (or even directions: positive or negative) between soil and the atmosphere and estimate whether soil is a source or sink for VOC. The authors state that belowground and aboveground concentration (the latter from an earlier campaign) were "similar in magnitude". In this case, one would assume that both emission and uptake is possible (the authors may also refer to, e.g., Gut et al. (2002) for a theoretical background of calculations necessary for this kind of soil trace gas measurements).

We thank the referee for being thorough in reading the text. Here, the aboveground concentration has been used wrongly as a synonym for emissions, which is obviously not the case. The aboveground ambient air VOC concentrations can be obtained from the enclosure measurements before the closing of the chamber. We calculated that the ambient air concentrations were 0.3 to 17 per cent of the monoterpene concentrations in the O-horizon and the results are presented on Page 11, lines 11-14.
We acknowledge that for many VOCs both emission and uptake is possible, and indeed have also seen in our data that this is the case especially for water soluble compounds like methanol or acetone (see e.g. Aaltonen et al., 2013).
We referred to Gut et al., (2002) as a theoretical background on Page 5, line 26.

Aaltonen H., Aalto J., Kolari P., Pihlatie M., Pumpanen J., Kulmala M., Nikinmaa E., Vesala T., and Bäck J.: Continuous VOC flux measurements on boreal forest floor. Plant and Soil, 369, 241–256, doi:10.1007/s11104-012-1553-4, 2013.
Gut, A., van Dijk, S. M., Scheibe, M., Rummel, U., Welling, M., Ammann, C., Meixner, F. X., Kirkman, G. A., Andreae, M. O., and Lehmann, B. E.: NO emission from an Amazonian rain forest soil: Continuous measurements of NO flux and soil concentration, J. Geophys. Res.-Atmos., 107, 8057, doi:10.1029/2001JD000521, 2002.

3): I have concerns about the belowground measurement procedure or representativeness of respective VOC concentrations. The question is how the belowground VOC concentrations were derived/calculated? I understand that the authors were running their sampling system in a closed loop, having a "perfect" sink for VOC at the side of the Tenax adsorption tube, i.e., the air flow downstream of the adsorption tube will be depleted of VOC.
On the other side of the sampling system (at the inlet of the collector) this VOC-depleted air creates an artificial gradient that forces VOC from soil-air to penetrate through the collector membrane. This happens very soon after the sample pump starts, as the VOC that have been accumulated during the 15 min sample breaks are transferred to and trapped by the adsorption tubes within a time span of about 1-2 minutes (sample flow of 100-150 ml, and collector volume of 150 ml in 2009-2011; no numbers given for the volume in 2016).

The VOC concentrations (C, $\mu g\ m^{-3}$) for the different soil horizons were calculated with Eq. (2):

$$C = \frac{m}{\frac{V}{1000000}} \qquad\qquad (2)$$

where $m$ is the mass of sample (ng) and $V$ is sampled volume (ml), divided by 1000000 to calculate the unit conversion from ml to m$^3$. $V$ was calculated using Eq. (3):

$$V = F * t \qquad (3)$$

where t is the sampling time (min) and F is the flow rate of the sampling (F, ml min$^{-1}$).

Soil is by nature very heterogenous measurement environment, and the spatial and temporal representativeness can naturally be questioned as always in soil measurements. We designed the sampling schedule so that both the temporal and spatial variation should be sufficiently covered with the 5-8 soil pits: 18 sampling events between November 2008 and October of 2011 (3 pits); and 13 sampling events in 2016 (5 pits). The measurement and analysis is tedious and time consuming, and therefore more samples could not be taken, unfortunately.

The measurement procedure was designed (and tested) in particular to avoid creating artificial gradients of VOCs in the soil and to avoid the possibility for sucking VOCs from a larger 'footprint' to the sample, and thus the sampling duration was optimized to 15 min with a break in between the samplings. The 15-min break between individual samplings was used to stabilize VOC concentration between the gas collector and surrounding soil air. The permeability test of the collector (Fig. 2) indicates that the collector itself should not create VOC vacuum by restricting the VOC flow.

3) In the residual time of the (4x) 15 min sampling period the authors applied an artificial VOC concentration gradient by flushing the collectors with VOC-free air. That means that the longer the sample interval the more VOC will be accumulated in the adsorption tube. If taking the total sample volume (6-9 l) into account to calculate the VOC concentration this will not represent the VOC concentrations prevailing in the soil air. Rather, this is a measure of how much VOC can penetrate through the collector membrane when an artificial VOC gradient (with zero VOC in the collector) is applied. Of course the VOC penetration rate will, among others, depend on the soil VOC concentrations, but will by no means be representative of (or equilibrated with) the soil-air VOC absolute concentrations (compare Fig. 2). In fact, if the calculation is accounting for the total sampling volume, the derived concentrations will be much lower (with the concentrations inside the collector being lower as soon as the pump is on). Else: with very large sampling volumes of 6-9 liters one can assume breakthrough in the adsorption tubes to occur to some extend by some of the VOCs, which make things even more complicated. What the authors measure is, to the most of their sampling procedure, the VOC transmission rate of their collector wall/membrane. This issue of the method/calculation of belowground concentration is especially critical for the sesquiterpenes, which are indeed expected to have relatively long equilibration times or lower Teflon diffusion rates, respectively. Else: even though a major fraction of this manuscript is reporting on sesquiterpenes, these compounds were not tested for collector permeability. As stated by the authors, the collector permeation rate can be assumed to be much lower than for the other compounds, as the penetration rate is, among others, dependent on size. Differences in transmission rates in Teflon can easily span several orders of magnitude, please see, e.g.: https://www.chemours.com/KIV/zh_CN/assets/downloads/Chemours_Teflon_FEP_Film_Tech_Bulletin_K26942.pdf

The aim of this study was to measure the VOC concentration of certain soil volume within soil pores, not the VOC concentration of gas collector. We used the sample volume of 6–9 L to make sure that the amount sampled exceeded the detection limit of the TD-GC-MS for monoterpene and sesquiterpene quantification. On-line analysis techniques would be ideal for this kind of measurements, but PTR-quadrupole-MS is not capable to differentiate monoterpene species or measure sesquiterpenes even in single mass level reliably. Since sesquiterpenes occur in extremely

low concentrations and are highly reactive compounds, this means that they are very challenging to measure. Low flow rate was used to avoid/prevent the flow-through of VOCs in the adsorption tubes. 6-9 liters of sample air is required to achieve measurable concentrations of VOC from soil air. Usually 0.05-0.10 L min$^{-1}$ flow rates are used in measuring VOCs with flow through chamber techniques (Aaltonen et al., 2011, Hellen et al., 2006, Mäki et al., 2017).

The gas collectors were made of polytetrafluoroethylene (PTFE) by sintering, with pore sizes of 5–10 μm. The pores in the collectors allow the diffusion of gases to occur, while water is unable to percolate into the collector. This was clarified on Page 4, line 31.

Breakthrough volumes of tubes have been tested and no breakthrough have been observed for the studied compounds even with higher (12 L) sampling volumes. This sentence was added on Page 6, lines 4-5.

We wrote in the manuscript: "All the VOC standard compounds permeate the collector easily and concentrations reach a constant level in order of minutes (maximum 7 minutes) also with the wetted collector (Fig. 3). α-Pinene was the heaviest compound in the calibration VOC gas mixture, and as expected, its diffusion through the wall of the collector was the slowest of the VOCs measured, with stabilization time 7 min. In contrast methanol peaked immediately after introducing the gas mixture into the glass bottle, but after that it stabilized quickly in 4 min. It was assumed therefore that stabilization of sesquiterpenes would take longer, since they are heavier than monoterpenes, thus the 15-min break time was chosen." Molecular weight of methanol is 32.04 g/mol and molecular weight of α-pinene is 136.23 g/mol, while it is 204.357 g/mol for α-gurjunene.

Teflon tubing was not permeating but only used for conducting the air flow to the sintered Teflon collector and consequently from the collector to the Tenax adsorbent.

Results from permeability tests of the PTFE collector were shown in Figure 2 on Page 23.

[Figure]

Figure 3: Results of the permeability tests of the PTFE collector with the five VOCs. A permeability test was used to monitor how fast VOCs permeate into the gas collector and to determine how fast VOC concentrations stabilize between the air inside and outside the collector. Panel a) shows the results with dry collector and panel b) with a wetted collector. Vertical line shows the time point when the introduction of the VOC standard began.

Aaltonen, H., Pumpanen, J., Pihlatie, M., Hakola, H., Hellén, H., Kulmala, L., Vesala, T., and Bäck, J.: Boreal pine forest floor biogenic volatile organic compound fluxes peak in early summer and autumn, Agricultural and Forest Meteorology, 151, 682–691, doi:10.1016/j.agrformet.2010.12.010, 2011.

Hellén, H., Hakola, H., Pystynen, K.H., Rinne, J. and Haapanala, S.: C2-C10 hydrocarbon emissions from a boreal wetland and forest floor. Biogeosciences, 3: 167–174, doi:10.5194/bg-3-167-2006, 2006.

Mäki, M., Heinonsalo, J., Hellén, H., and Bäck, J.: Contribution of understorey vegetation and soil processes to boreal forest isoprenoid exchange. Biogeosciences, 14(5), 1055-1073, doi:10.5194/bg-14-1055-2017, 2017.

4) The authors either should have used an online VOC analytical device with high temporal resolution to detect the short high concentration peak directly after they had started the sampling pump. Only this small volume of air (if at all) can reflect the soil-air concentrations, assuming that the collector air volume has reached equilibrium during the 15 min sampling breaks. Or they should have used an online VOC analyzer within a close loop sampling system, which does not interfere with (adsorb) VOCs. Please correct me if I am wrong.

This would not be possible with online VOC analyzers (quadrupole-PTR-MS or PTR-TOF-MS), because they are unable to separate compounds with same molecular mass such as different monoterpenes or sesquiterpenes. Also detection limits especially for sesquiterpenes with quadrupole-PTR-MS are very high. We agree that the PTR-TOF could have been used for VOC concentration measurements, but the quantification accuracy of sesquiterpenes compared to real values is still unclear with this instrument. In order to cover the spatial variability in soil better, we wanted to install the collectors over a large area, and to measure them with PTR-MS would have required long tubes. The highly reactive sesquiterpenes can be transformed in long tubes through the chemical reactions before they will reach the detector. Further, online sampling of soil air is not as simple as sampling of ambient air, because soil water content can be high in the deeper soil horizons. We are anyway confident that the relative differences in the concentrations measured in different soil layers with this technique represent true differences in the actual concentrations between the layers.

One value of this experiment was that we were able to detect over 50 different VOCs from soil air. This was possible by using the TD-GC-MS, but would not have been possible with quadrupole-PTR-MS, because it is unable to separate compounds with same molecular mass such as different monoterpenes or sesquiterpenes.

5) Due to the issues #1-3, an interpretation of the results of the manuscript is indeed difficult to achieve, and any conclusions can only be of speculative nature. Concerning issue #1, one could state that using zero-air for purging the enclosures reveals a measure of the potential soil VOC emission capacity. And a direct comparison with the belowground exchange measurements doesn't make sense anyhow, due to multiple reasons. Concerning issue #2, the authors give a range of VOC concentrations measured in ambient air above this forest in an earlier campaign; and state that those were "similar in magnitude" as the observed belowground concentrations. This could be a fair projection, but only in case issue #3 could be solved. Concerning issue #3, one could state that the observed belowground concentrations (and vertical profiles) are a (very) lower-bound estimate, but this is very much dependent on the individual compounds (diffusion characteristics). Then the vertical VOC concentration differences between the different soil depth/horizons could at least be discussed, rather than any of the absolute VOC concentrations. In general, the way of data presentation/structure in the manuscript is sometimes not easy to follow or not precise, and the interpretation of the correlation analysis is kind of vague or speculative. In view of the issues presented above, the decoupling of the belowground VOC concentrations from the forest floor fluxes and the scarceness

of correlations/gradients (e.g. for sesquiterpenes and OVOC) is not surprising. Even though the authors try to pin down potential dependencies (soil temperature and water content) by displaying tabulated statistical data, the data evaluation did not give a conclusive picture. The correlation analysis sometimes gives encouraging numbers for some soil layer horizons, or individual pits of those, but not for some others or adjacent soil layers, which doesn't add confidence in respective interpretations.

We thank the referee for comprehensive criticism. In our responses to issues 1-3, we have tried to clarify and sharpen the methodological concerns the referee raises. It could indeed be said that all emission measurements are in fact emission 'potentials', however this terminology has already been reserved to the Guenther emission model concept where emissions are normalized to standard temperature and irradiance levels, so it may be even more confusing to use the word in a different context.

We agree with the Reviewer that the vertical VOC concentration differences between the different soil horizons could be discussed next to the absolute VOC concentrations. We have now phrased the comparisons between soil horizons in several places (Page 9, lines 27-30 and Page 10, lines 1-4) as the relative differences.

We agree that due to the multiple cross-correlating drivers, a conclusive and convincing statement of concentrations of all compounds and their relations to environment is difficult to obtain, especially in a field campaign as here. However, we still believe that our data is measured in a solid manner, analyzed with up-to-date methods and brings about novel understanding on the possible role of soil processes in VOC production. This is - as far as we know - the first quantitative analysis of this topic. We added this statement to the end of the conclusions on Page 16, line 32.

We also added following sentences to the manuscript on Page 6, lines 7-10: Especially for sesquiterpenes, which are expected to have low diffusion rates, concentrations are lower-end estimates. During the sampling sesquiterpenes are not expected to be diffusing fast enough through the walls of the tubes and most of the mass is actually collected during the 1-2 minutes of the sampling.

Limits of the methods have now been discussed in the manuscript and results have been discussed more by the differences in the concentrations and not by the absolute values.

6) In a recent paper, the authors already concluded from dynamic (zero-air) enclosure measurements that belowground dynamics might not play a major role in isoprenoid exchange, but instead the litterfall is the most important factor triggering VOC emissions (Mäki et al., 2017); and with all the short-coming presented above this seems to be confirmed by the belowground vertical gradients of VOC concentration in this manuscript. Concerning all issues above, I suggest that the authors consider reassessing their conclusions in respect with the critical points presented above (and below) and resubmit a new version.

We have rewritten the conclusions on Page 16, lines 28-32: Soil vertical layer VOC concentrations were analysed and compared with simultaneous chamber flux measurements in field conditions in a Boreal coniferous forest. We detected more than 50 different VOCs, mostly mono- and sesquiterpenes, and belowground concentrations of VOCs differed between soil layers during the second campaign. Sources of the forest soil VOCs probably differ depending on the compound and soil layer. Dominating monoterpenes concentrations are comparable to the air concentrations above a coniferous forest. This is - as far as we know - the first quantitative analysis of this topic.

7) The M&M section needs to describe more details. Before describing the collector permeability test (in section 2.2), the authors should first introduce/describe the innovative type of collectors used (as they did later in section 2.3.).

The collector permeability test was moved after the gas collector description on Page 5, lines 10-26.

8) What does "wet collector" mean (page 4, line 6)? How did you wet it? Did you apply humidified air?

Before permeability measurements with the wetted collector, it was wetted with ultrapure water. Also the gas flow was humidified. This was done in the test to mimic the moist conditions inside the soil where the collectors remained between measurements. This description was added on Page 5, lines 17-18.

9) What is the meaning of "The break length…" (page 4, line 11)?

The 15-min break between individual samplings was used to stabilize VOC concentration between the gas collector and surrounding soil air. This was clarified on Page 5, lines 33-34.

10) Is the "sampling system" (page 4, line 22) the same as the "collector"? Otherwise I don't get it. What do the authors mean with ": : : with the pits" at the end of the same sentence? Connect the tubes with the pits? The term "within the pits" make more sense to me. Please clarify.

'Sampling system' is the whole measurement system, where gas collectors (gas permeable PTFE-tubes) are connected to stainless steel tubes from which the VOC samples were taken. The misspelling on Page 5, line 1 was corrected.

11) "For aboveground sampling" (page 4, line 28:) can be misinterpreted, as you didn't do any aboveground measurements. I suggest to merge this and the follow-up sentence in a concise way.

These two sentences were rewritten on Page 5, lines 6-9.

12) Permeability test: which concentrations did you use for the test (in the Fig. 2 it says "arbitrary units")? Were the concentrations inside the collector the same as outside ("at constant level"), as Fig. 2 lets assume?

The units are arbitrary, since the instrument was not calibrated for the test. As only concentration differences were measured, the calibration was not required. At the end of the individual tests, the VOC concentrations outside the collector was also measured and observed to be equal with concentrations inside.

13) What is meant by "The possibility of creating a flux collectors that did not originate from the actual measured horizon" (page 5, line 4)?

We mean that if the VOC sampling flow would be very high, it would suck VOCs from the surrounding soil horizons. With small flow rate, we minimize this risk and sample VOCs from the gas collector, which is placed in the middle of certain soil horizon. The misspelling was corrected on Page 5, line 30.

14) What is meant by "were closed between the samplings" (page 5, line 5). Closed in between the consecutive sampling intervals of one sample procedure (closed for 15 min during sampling brakes)? Or did the authors close the tubes when they finished one complete sampling cycle?

*We agree that this part was very unclearly written and we made corrections on Page 5, lines 32-33.*

15) I got lost understanding the different pits versus investigated soil horizon designations (2 versus 4 in the different campaigns) versus soil depth (5 in table 2). From Table A1, I understand that the two soil depths (organic and mineral) investigated in 2008-2011 refer to horizons H & B in 2016. On page 8, line 22, the authors state "mineral soil (A- and B-horizons)". I suggest to shortly describe which layers of the 2018-2011 measurements refer to which in 2016 in the M&M section. What do negative numbers of soil depth mean in Table A1?

*Details were clarified on Page 6, lines 23-24, and in the Table A1 on Page 31.*

16) The total sampling volume was 6-9 liters of air. Did the authors test any VOC breakthrough? 9 liters is much more than normally applied.

*The total sample volume of 6–9 L was used to exceed the detection limit of the TD-GC-MS. One value of this experiment was that we were able to detect over 50 different VOCs from soil air. We think that smaller sample volume would have been sufficient for monoterpene quantification. We used higher sample volume, because we wanted to measure sesquiterpene concentrations as well. Breakthrough volumes of studied compounds were tested and no breakthrough was observed even with the sampling volume of 12 L (Page 6, lines 5-6).*

17) Omit "0 cm being the surface of organic layer, not mineral soil" (page 5, line 17). What do the negative soil depth numbers in Table A1 mean? May be I didn't get the above.

*Correction was made on Page 6, line 13-15. Details were clarified in the Table A1 on Page 31.*

18) What is meant by ": : : installed in the vertical face interfacing with the undisturbed soil: : :." (page 5, line 27)?

*The sentence was rewritten on Page 6, line 25.*

19) Page 6, line 4: in case that VOC-free air was used to flush the enclosures (as in Mäki et al. 2017) you should mention this here; as in this way, an artificial concentration gradient is produced that enhances trace gas emission and omits any deposition or bi-directional exchange rates to be observed.

*We added the following sentence on Page 7, line 5: "We flushed the chamber headspace for 30 minutes to equilibrate the measurement system." See also our response to Q1.*

20) Please give a some more details of the enclosure system applied (instead of only citing your previous paper). Also state the basic calculation formulas here.

*We added the following details on Page 7, lines 5-8, and Page 7, lines 12-18. "We flushed the chamber headspace for 30 minutes to equilibrate the measurement system. During the chamber enclosure, we continuously pushed ($1\,l\,min^{-1}$) filtered (active carbon trap and MnO2-coated copper net) ambient air into the chamber headspace and sampled the incoming and outgoing air for 1.5–2 hours through two Tenax TA-Carboback-B adsorbent tubes (flow rate $0.1$-$0.15\,l\,min^{-1}$)." We also added the equation for the flux calculations.*

21) Other minor issues: Page 1, line 16: omit "the" in "during the two measurement campaigns"

The word was removed on Page 1, line 16.

22) Page 8, line 3: the authors state that "Belowground VOC concentrations were dominated by monoterpenes and sesquiterpenes, but the monoterpene concentrations were mainly decoupled from forest floor monoterpene fluxes." Obviously also the SQT and OVOC were "decoupled".

We made a clarification on Page 11, lines 2-3.

23) Page 8, line 5: what is meant by "Belowground VOC concentrations in the vertical soil horizons". Suggest: "Belowground vertical gradients of VOC concentrations".

This title was corrected based on the suggestion (Page 9, lines 24-25).

24) Page 8, line 15: what is meant by "… when each soil horizon was tested separately"?

We rewrote the sentence on Page 10, lines 8-9. Our aim was to say that there were no differences in VOC concentrations between the soil pits for O-, A-, B-, or C-horizon.

25) Page 8, line 21: "Total monoterpene concentrations in organic soil were highest in late summer and in December": Comparing late summer (28.07., 24.8., 21.9.) with fall (1.10., 14.10., 26.10., 8.11., 2.12.), this is hard to tell. It is sometimes hard to follow what the authors exactly mean when discussing data in spring, early/late summer, autumn in the different chapters of the main text. Did they really plot means in Fig. 5, or media (error bars are not evenly distributed to the positive/negative direction)?

In the Fig. 5 on Pages 26 and 27, we plotted the mean isoprene, monoterpene, and sesquiterpene fluxes and concentrations for the O- and A-horizon. Error bars of monoterpenes and sesquiterpenes are not evenly distributed to the positive/negative direction, because the values are presented in log scale on the y-axis. Details were clarified on Page 10, lines 15-22.

26) Page 8, line 25: "Total sesquiterpene concentrations in mineral soil were clearly highest in spring, in early June, in late summer, and in October (Fig. 5)." Early June is still spring time. I can't see this general trend at all in Fig. 5c.

The sentence was rewritten on Page 10, lines 19-20: "Total sesquiterpene concentrations in the A-horizon were highest in spring (22.4. and 17.5.), in late summer (24.8.), and in October (1.10.)".

27) Page 8, line 31: "There was no difference in VOC fluxes between measurement pits." I am not sure whether I got this right. It's hard to believe that all fluxes of all VOC (classes) were similar, due to the inhomogeneity of the forest floor mentioned.

The sentence was rewritten on Page 10, lines 25-26. This sentence was meant to present that there were no statistically significant differences in VOC fluxes between measurement pits within the different VOC groups. Typical feature for VOC emissions is that the flux rate variation is very high.

28) Page 9, line 3-4: I suggest: "In contrast to our hypothesis, the below ground vertical concentration profiles were not coupled to observed soil surface fluxes rates, ..."

This sentence was rewritten based on the Reviewer's suggestion (Page 10, lines 29-30).

29) Page 9, line 5: "individual pits" is redundant.

The sentence was rewritten based on the Reviewer's comment (Page 10, line 32).

30) Page 9, line 12: "Confirming our third hypothesis that soil temperature and water content can be used to explain belowground VOC synthesis." Due to a lack of correlation with all other VOC classes versus soil horizons, I would not state that these results are confirming the third hypothesis. May be you can state that these individual correlations are in line with the hypothesis, but then you also have to mention that all other correlations fail to do so. Else: this sentence is missing its subject.

We think that this is a very good suggestion from the Reviewer. Our observation was rewritten on Page 11, lines 19-22.

31) Page 9, line 23: "The organic soil showed seasonal variation in 2011 and 2016 : : : (Fig. 6)". As Fig. 6 only shows summer data: how can the authors claim that there are "seasonal variations?" Or did they mean inter-annual variations, or inter-campaign variations?

Figure 6 showed seasonal variation from spring (May) and summer (July) in 2011 and 2016. The statement was clarified on the Page 12, lines 1-2. We agree with the Reviewer that you can also see inter-campaign variation between the campaigns one (2009-2011) and two (2016).

32) Page 9, line 25: "Monoterpenes constituted almost 90% of the total VOC concentration, sesquiterpenes accounted for less than 10% between 2008 and 2011 (Table 5)." How does this VOC composition compare to (expected) ambient air data (in lack of own data, please give a general statement)?

Most of the sesquiterpenes, especially ß-caryophyllene which is the main sesquiterpene emitted by Scots pine, are so reactive towards ozone that they cannot be measured in the ambient air. Lifetime of B-caryophyllene in the air at the site is less than 2 minutes, while for monoterpenes lifetimes are few hours. Therefore VOC composition is not directly comparable to the emission composition. High sesquiterpene emissions have been measured at the site both from Norway spruce and Scots Pine shoots in summer (Hakola et al., 2017, Hakola et al. 2006), but due to high reactivity ambient air concentrations of sesquiterpenes have been mainly below detection limits, while monoterpenes have been detected in the ambient air even during winter (Hakola et al. 2012).

Hakola H., Hellén H., Rinne J., Hemmilä M., and Kulmala M., 2012. In situ chromatographic measurements of volatile organic compounds in a Boreal Forest. Atmospheric Chemistry and Physics, 12, 11665-11678.
Hakola, H., Tarvainen, V., Praplan, A. P., Jaars, K., Hemmilä, M., Kulmala, M., Bäck, J., and Hellén, H. Terpenoid and carbonyl emissions from Norway spruce in Finland during the growing season. Atmospheric Chemistry and Physics, 17, 3357–3370, 2017, doi:10.5194/acp-17-3357-2017.

33) Section 3.4 ("Inter-annual variation"): the discussion on seasonal pattern is sometimes redundant (see section 3.1 and 3.2), but with different phrasings, e.g.: "Monoterpene concentrations in 2016 were highest in organic soil in summer, in October and in December, whereas seasonal variation was relatively small in mineral soil (in section 3.4)." versus "Total monoterpene concentrations in organic soil were highest in late summer and in December : : : (section 3.2)".

The sentence on Page 12, lines 11-12 was removed.

34) Page 12, line 19: "Belowground isoprenoid concentrations varied seasonally, and the highest concentrations were measured during summer and early autumn in 2009 and 2011, whereas high belowground concentrations monoterpene concentrations were measured in late summer, in October, and in December in 2016.". I think the authors should not compare the total isoprenoids (ISO, MT, SQT) in 2009-2011) with only the MT in 2006. What about the other isoprenoids (SQT, isoprene) and what about the OVOC in general?

Corrections were made on Page 15, lines 7-10. We did not quantify OVOC concentrations in 2008-2011.

35) Page 13, line 27: "led to"

We rewrote the sentence on Page 16, line 14.

36) Fig. 2: the x-axis has no units given

We added the units for the x-axis (Page 24).

37) Fig. 5: it seems that the x-axis has equi-distant steps for the different sample dates. I propose to use an absolute numeric time line (the sampling/breaks were not evenly distributed over time).

We agree with the Reviewer that this suggestion will make the Figure 5 more realistic. We have modified the Figure 5 (Pages 26 and 27). We have also removed the data from the B- and C-horizon to make the trends more visible.

38) Figure A2: any idea why the water content of A horizon (lying between the H and B horizon) is so much higher than all the others? Indeed, the soil water content can be quite inhomogeneous (e.g., by water channeling etc.). If only measured by one single sensor per soil depth, these measurements are not necessarily representative.
References:
Gut et al. (2002): NO emission from an Amazonian rain forest soil: Continuous measurements of NO flux and soil concentration. J. Geophys. Res., 107 (D20), 8057, doi:10.1029/2001JD000521.
Mäki et al. (2017): Contribution of understorey vegetation and soil processes to boreal forest isoprenoid exchange. Biogeosciences, 14 (5), 1055-1073, doi:10.5194/bg-14-1055-2017, 2017.

The Figure A2b shows that soil water content is not highest in the A-horizon (black line with circle markers), but instead in the C-horizon (gray line with circle markers). Soil water content is highest in the C-horizon, because rainfall will percolate through the soil profile into the C-horizon. The measurement sensors are close to groundwater in the C-horizon. Soil volumetric water content in the O-, A-, and B-horizon are means of five measurement pits and volumetric water content in the C-horizon is mean of four measurement pits at the SMEAR II station. This clarification was added into the M&M section (Page, 7, lines 26-28). We agree with the Reviewer that soil water content can vary strongly between sensors immediately after rain events, but we strongly believe that these continuous soil water content measurements are representative for actual soil water content in the long run.
        We have changed the coloring of the lines to make our point more clear (Fig A1 and A2, Pages 35-36).

Reviewer #2: The manuscript bg-2018-22 describes 2 new setups to measure seasonal and depth dynamics of volatile organic compounds in a haplic podzol in a boreal forest (SMEAR II site, Finland). The manuscript compares results measured with 2 methods and concludes about

seasonality, which might be just caused by the differences in the methods. Additionally, the manuscript is written rather descriptive and general with a focus on atmospheric chemistry rather than biogeosciences. As it is, the manuscript might be better for publication in AMT or ACP. Instead of comparing the concentrations within the soil profile to the flux from the surface into the atmosphere, I would like to read more about possible biogeochemical processes involved in the production of the individual compounds based on literature.

I will point out some additional references and ideas to change the focus more towards biogeosciences. In general the measurement of soil VOCs measured in depth profiles measured via TD-tubes and analysis by GC-MS is very challenging and unique, thus, I recommend the manuscript for publication. I just have problems to conclude about seasonality if 2 different methods have been applied and no pressure was measured. I recommend to focus rather in the dynamics within the soil depth profile rather than on the seasonality. More detailed comments for a revision are addressed bellow. First of all, I have a problem with the term storages. It suggests that e.g. in plants isoprenoids are stored and released based on physico-chemical processes. While this might certainly be true for the top litter layer, there is strong evidence that microbes in soil can actively produce mono- and sesquiterpenes (e.g. Schulz and Dickschat 2007, Yamada et al., 2015) within their metabolism. Page 1, line 27: It is not really the high organic carbon content which results high VOC emissions from organic horizons, but rather the highest abundance and activity of autotrophic and heterotrophic microbes in that layer.

We have tried to emphasize this aspect throughout the manuscript. We agree on the scientific evidence that VOCs are produced by microbial metabolism and we wrote corrections into the text by using the suggested references on Page 1, lines 27-28. On Page 1, line 27, our aim was to point out that easily available carbon enhances microbial metabolism, which can lead to higher VOC production, but we clarified this on Page 1, lines 28-29 and Page 2, lines 1-2 based on the feedback: Organic soil layers can be a substantial source of VOCs due to the high abundance and activity of autotrophic and heterotrophic microbes. They drive decomposition processes where easily available carbon is utilised for microbial metabolism and VOCs may be produced either actively as secondary metabolites or as by-products in the decomposition process.

The measurement method itself was actually the same in both campaigns. Samples were collected by circulating air in the gas collectors and through Tenax TA–Carbopack-B adsorbent tubes at flow rates that ranged 100–150 ml min$^{-1}$ using portable pumps and impermeable PTFE tubing. Gas collectors were placed into the different measurement pits in the campaigns one (2008-2011) and two (2016) and gas collector type was also different between campaigns. This probably will have some effect on VOC concentrations. For this reason, the main focus is in the second campaign (2016).

Schulz, S, and Dickschat, J.S. Bacterial volatiles: the smell of small organisms. Natural product reports, 24, p.814–842, 2007.
Yamanaka, K., Reynolds, K. A., Kersten, R. D., Ryan, K. S., Gonzalez, D. J., Nizet, V., Dorrestein, P. C., and Moore, B. S. Terpene synthases are widely distributed in bacteria. Proceedings of the National Academy of Sciences of the United States of America, 111, 1957–1962, doi:10.1073/pnas.1422108112, 2015.

2) Page 2, line 1-16: This is a general intro for the relevance of VOCs in atmospheric chemistry. Given the focus of the manuscript on VOC dynamics within a soil profile, I would recommend to start here with the role of As stated already on page 11 line 31 ff., the VOCs produced in the soil profile differ from the VOCs released into the atmosphere. Thus they are not necessarily transported all the way up into the atmosphere and thus their role within the soil should be focused.

Our aim in this chapter was to connect soil processes to atmospheric chemistry and to the climate change, but we agree that this part could be shorter. We change the focus of this chapter to soil VOCs dynamics on Page 2, lines 3-24.

Insam, H., and Seewald, M.: Volatile organic compounds (VOCs) in soils. Biology and Fertility of Soils, 46:199–213, doi:10.1007/s00374-010-0442-3, 2010.
Schulz, S, and Dickschat, J. S. Bacterial volatiles: the smell of small organisms. Natural product reports, 24, p.814–842, 2007.

3) Page 2, line 12 ff.: Diffusion also is dependent on soil moisture, not only soil temperature (see e.g. Skopp et al., 1990).

We completely agree with the Reviewer and the sentence was rewritten on Page 2, lines 15-18.

4) Page 2, line 22 ff.: A major result of snow cover is that the soil is isolated from the cold air temperatures and is not freezing. Thus, I agree that microbial processes might still be ongoing. However, given the fact that microbial metabolism is strongly correlated to soil temperature, which should be quite soil in winter, I think an enrichment effect is more likely. The snow acts as a lid of a static chamber.

We agree with the Reviewer that one the main reasons for VOC concentrations to be so high inside snow bed is the lid effect of snow. We rewrote the sentence on Page 2, lines 32-33.

5) Page 4, line 21 ff.: In both setups polytetrafluoroethylene (PTFE) tubes, which were closed on one side by a sintering method, have been used. I have problems to understand how the first method, which was applied from 2008 until 2011 to suck air out of the sintered tube with a pump, reflect "diffusion of gases to occur" (line 21 ff.). According to my knowledge a pump creates a pressure difference from the inner tube to the surrounding soil. Thus, the soil air was sucked into the tube and does not reflect natural conditions where molecular diffusion occurs. This was improved in the second setup in 2016, where air was circulated through the same tubes and the assumption of molecular diffusion for that data are more likely to be valid. Without a pressure measurement as e.g. Gut et al. 1998, I have problems to follow the assumption of molecular diffusion for the first setup. PTFE tubes can be manufactured with different volume density and thus the mesh of stainless steel is also important to prevent that the soil is changing the inner volume of the PTFE tubing. Thus, the volume density should be included in the method description.

The measurement method was actually the same in both setups. Samples were collected by circulating air in the gas collectors and through Tenax TA–Carbopack-B adsorbent tubes at flow rates that ranged 100–150 ml min$^{-1}$ using portable pumps and impermeable PTFE tubing. Volume density of the PTFE tubes is not critical, because tube surface is impermeable. The PTFE tubes are used to transport gas sample from the gas collector into the Tenax TA–Carbopack-B adsorbent tube.

6) Page 8, line 9 ff.: Highest sesquiterpene and OVOC concentrations in the A horizon should be discussed with respect to the difference in particle density of O and A horizon material. This impacts the overall water filled pore space and thus might explain your result. In general, it is expected to observe highest concentrations in the O horizon. Another point which is missing in the discussion is the potential of utilizing sesquiterpenes and OVOCs as microbial signaling in the A horizon.

We added microbial signaling on Page 13, lines 20-25.

Soil properties also explain high monoterpene concentrations in the O-horizon. Soil porosity is higher in the O-horizon compared to the A-horizon, which means that gas diffusion is faster in the O-horizon compared to the A-horizon. The effect of rain filling soil pores and transporting VOCs towards deeper soil layers is likely stronger in the O-horizon. This was also added on Page 13, lines 1-4.

7) Page 8, line 18 ff.: low oxygen availability does not necessarily results low aerobic microbial activity. Anaerobic microbes will be still active.

We agree with the Reviewer and we rewrote the sentence on Page 10, lines 12-13.

8) Page 9, line 3 ff.: I don't understand why the hypothesis surface VOC fluxes and belowground VOC concentrations are similar was formulated? Wouldn't exactly the opposite be true? The surface VOC flux is dependent on the turbulent eddy diffusion coefficient, whereas the belowground VOC concentrations are dependent on the molecular diffusion coefficient. Since they are several orders of magnitude different, I would not expect that surface VOC fluxes and belowground VOC concentrations should follow the same trend/pattern.

The Reviewer makes an excellent point that it is not straightforward to compare belowground concentrations and surface fluxes, because the production processes, transport mechanisms and temperature and moisture conditions are different. We thought that we could find similar seasonal pattern in belowground VOC concentrations and surface fluxes, but after the Reviewer's feedback, we decided to rewrite the hypothesis on Page 3, lines 18-20.
In the chamber headspace, we have a fan to homogenize the chamber air volume, but we don't have natural turbulent mixing, which regulates diffusion gradient between soil air and the below-canopy atmosphere. This can cause small error to the measurement system, although we continuously push VOC free air into the chamber headspace.

9) Page 9, line 18 ff.: I have problems to follow inter-annual variability if 2 different methods have been applied.

We agree that it is difficult to make comparisons between the campaigns one and two, when VOC concentrations measurements were done using two different measurement set-ups and we added this statement on Page 15, lines 11-13.

10) Page 10, line 11 ff.: It is a kind of recapitulation to summarize results and discussion in 2 sentences. I would remove both and rather move them into the conclusion.

We rewrote the conclusions based on the suggestions on Page 16, lines 28-32.

11) Page 10, line 16 ff.: I agree, but in the discussion section I want to read also why the monoterpene concentrations are highest in the organic horizon (not soil)? The pores in the organic layer are much larger than in the mineral soil. Thus, fungi, which need to grow hyphae from one particle to another are rather slow. Thus it is not surprising that on the other hand bacteria were found in e.g. Timonen et al. 2017 to be high abundant in the humus. It is known that bacteria can easily colonize particles in the organic horizon since most are mobile. Thus, you could interpret the production of e.g. 3-carene and camphene from fungi as active inhibition of the swarming and swimming motility of bacteria. Such findings have been published already (Schmidt et al., 2015). These findings suggest that the production of terpenoids is rather connected to microbial activity than on microbial abundance. I am sure that you can find much more correlations of your data to microbial processes.

This was an excellent advice. However, we did not determine microbial populations from the soil horizons. For this reason, it is difficult to make conclusions about interactions between bacteria and fungi, which would be supported by scientific evidence. We followed the Reviewer's suggestions and found that both fluxes and concentrations of the monoterpenes and sesquiterpenes correlate with the $CO_2$ flux in autumn, which supports our conclusion that VOC production was driven by microbial activity (heterotrophic production). We compared correlation between the total monoterpene and sesquiterpene fluxes ($\mu g\ m^{-2}\ h^{-1}$) and the chamber temperature (°C), and the $CO_2$ fluxes ($\mu g\ m^{-2}\ h^{-1}$) from the soil surface in spring, summer, and autumn in 2016 (Appendix Table A6, Page 37). We added a short description to the Material and Methods on Page 9, lines 7-9 and results on Page 11, lines 3-8. We also rewrote the discussion on Page 12, lines 26-28, based on these results.

We also compared correlation between the total monoterpene and sesquiterpene concentrations from the O- and the A-horizons and the $CO_2$ fluxes ($\mu g\ m^{-2}\ h^{-1}$) from the soil surface in spring, summer, and autumn in 2016 (Appendix Table A7, Page 38). We added this to the Material and Methods on Page 9, lines 9-11 and to the Results on Page 11, lines 8-10.

12) Page 11, line 20 ff.: Just for curiosity, can you comment on the speciation into α-, ß-, γ-sesquiterpenes?

We specified the sesquiterpenes without pure standards on Page 14, lines 1-2. Quantification of these sesquiterpenes is described on Page 8, lines 10-13: Calibration solutions for the sesquiterpenes, contained only longicyclene, isolongifolene, β-caryophyllene, α-humulene, α-gurjunene and β-farnesene. In 2016, other sesquiterpenes found in the samples were tentatively identified by their mass spectra and retention indices and quantified as β-caryophyllene, isolongifolene, or longicyclene.

13) Page 11, line 31 ff.: I did not yet found the commonly reported functions of belowground VOCs (e.g. defense communication and signaling).

We added a new sentence on Page 13, line 22-24, based on the suggestion.

14) Page 13, line 29 ff.: I don't the reference for climate change fits for your manuscript. I am not an expert for snow cover, but as far as I know snow isolates the soil surface. Thus, if the snow in the future will not be present anymore, I would assume that the surface temperature of the soil should be colder.

We agree with the Reviewer that soil surface temperature would probably be colder without an isolating snow cover. Our aim with this sentence was to point out that VOC emissions from boreal soils could be increased, if there is no snow cover that hinders VOC diffusion in the atmosphere, because microbial activity also occurs in low temperatures. If air temperatures will increase and snow melts earlier, it could also increase VOC emissions from organic soil though microbial decomposition and metabolism in spring, when coming radiation warms a dark soil surface. We see that this part of the manuscript was written unclearly and we rewrote the sentence on Page 16, lines 15-19.

15) Page 14, line 1: The sentence … more research is needed I find too general. I think your manuscript shows some nice trends about VOCs in the soil depth profile which could be combined with existing literature to microbial processes. I can agree to … more research is needed to combine soil VOCs to microbial processes.

We rewrote the sentence on Page 16, lines 25-26.

16) Page 14, line 4: I find the conclusions rather short and just analyzing the temperature and moisture dynamics not very informative. I recommend discussing and concluding about microbial processes within the soil profile. Also you measured CO2, but did not really talk about the correlation.

We modified the conclusions on Page 16, lines 28-32.

17) Minor comments: It is confusing to read about the thicknesses (page 3 line 24 ff.) which are not reflected in the horizon borders in Table 1. Also I am missing the E horizon, which might explain the differences.

This sentence was removed from the manuscript (Page 4, lines 3-4). The E-horizon is part of the A-horizon. Physical or chemical properties of the measurement pits have not been determined for the E-horizon. We agree, that it would have been useful information, when comparing VOC concentrations from the different measurement pits.

18) Fig. 1: The scheme is rather a fast draft. I got especially lost following arrows which are not connected to a tube. Maybe for the non-expert reader it would be worth to include in the figure caption that the sintered version of the PTFE tube means that it is closed on one side? Also a lid of the glas bottle would help.

We modified the Figure 2 on Page 24.

19) Fig. 2: Maybe I did not get it, but the arbitrary signals for dry and wet indicate a moisture effect. In case you used cps, it might be worth to think about a different way to plot the data to correct for that effect or mention it in the method section?

Figure 3 on Page 24 shows results of the PTFE collector tests with the five VOCs. A permeability test was performed to monitor how fast VOCs permeate from the soil into the collector and to determine how fast VOC concentrations stabilize between the air inside and outside the collector. This was clarified on Page 24, lines 6-8. All the VOC standard compounds permeate the collector easily and concentrations reach a constant level in order of minutes (maximum 7 minutes) also with the wetted collector.

20) Fig. 3: The collectors in a) were installed in 5 and 17 cm, which are not the interface of the horizons. I suggest to focus on 2016 only and include the surface tube for flux measurement in b) plus a depth y-axis in cm. It is confusing to use the term "organic soil" and "mineral soil" while you speak about H-, A-, B-, and C-horizons.

We rewrote the sentences by deleting confusing parts on Page 6, lines 14-15. We also edited the Figure 1b on Page 23.

21) Fig. 4: There seems to be a problem either of the graphic or my printer for some error bars. I would like to read what processes cause the large differences of a-pinene, 3-carene, linalool and limonene in the soil profile. It is also worth to think about a classification into different relationships of VOC concentration with depth (e.g. exponential vs. linear, etc.). The carbon content and microbial biomass should decrease exponentially with depth. Thus, if VOC concentrations follow a different pattern, e.g. a-gurjunene, a-humulene and b-himachalene it could indicate that their production is not linked to the storage in plant litter, but rather likely to microbes which are most abundant/active in a specific layer of soil (A-horizon).

We checked the error bars and they are correct in the Figure 4 (Pages 25-26). Please note that the error bars don't present standard deviation, but standard error of the individual measurements. The values are presented in log scale on the y-axis.

We agree that there is hardly any discussion on which processes were behind the individual monoterpenes. We added some discussion on Page 12, lines 23-24, and Page 13, lines 20-22. We did not find clear distinction between linear and exponential relationships between soil conditions and VOC concentrations for the different soil horizons.

22) Fig. 5: a) It is hard to explain the elevated isoprene around 01.10 for the A horizon. Wouldn't it make sense to finally conclude that predominantly there is no difference in isoprene concentration and flux except this single event?

We completely agree and this clarification was added on Page 10, line 22.

23) Tab. 3: I recommend to plot $CO_2$ versus Sesquiterpene concentration and CO2 versus Monoterpene concentration and discuss the contribution of autotrophs (CO2 consumers) and heterotrophs ($CO_2$ producers), respectively.

This was an excellent advice. We followed the Reviewer's suggestions and found that both fluxes and concentrations of the monoterpenes and sesquiterpenes correlate with the $CO_2$ flux in autumn, which supports our conclusion that VOC production was driven by microbial activity (heterotrophic consumption). We compared correlation between the total monoterpene and sesquiterpene fluxes ($\mu$g $m^{-2}$ $h^{-1}$) and the chamber temperature (°C), and the $CO_2$ fluxes ($\mu$g $m^{-2}$ $h^{-1}$) from the soil surface in spring, summer, and autumn in 2016 (Appendix Table A6, Page 37). We added a short description to the Material and Methods on Page 9, lines 7-9 and results on Page 11, lines 3-8. We also rewrote the discussion on Page 12, lines 26-28, based on these results.

We also compared correlation between the total monoterpene and sesquiterpene concentrations from the O- and the A-horizons and the $CO_2$ fluxes ($\mu$g $m^{-2}$ $h^{-1}$) from the soil surface in spring, summer, and autumn in 2016 (Appendix Table A7, Page 38). We added this to the Material and Methods on Page 9, lines 9-11 and to the Results on Page 11, lines 8-10.

**Boreal forest soil is a significant and diverse source of volatile organic compounds**

Mari Mäki[1,2], Hermanni Aaltonen[3], Jussi Heinonsalo[3], Heidi Hellén[3], Jukka Pumpanen[4], and Jaana Bäck[1,2]

[1]Institute for Atmospheric and Earth System Research / Forest Sciences, Helsinki, 00560, Finland.
[2]Faculty of Agriculture and Forestry, University of Helsinki, Helsinki, 00790, Finland.
[3]Finnish Meteorological Institute, Helsinki, Helsinki, 00560, Finland.
[4]Department of Environmental and Biological Sciences, University of Eastern Finland, Kuopio, 70600, Finland.

Keywords: volatile organic compounds, boreal forest, organic soil horizon

*Correspondence to*: Mari Mäki (mari.maki@helsinki.fi)

**Abstract.** Vegetation emissions of volatile organic compounds (VOCs) are intensively studied world-wide because oxidation products of VOCs contribute to atmospheric processes, but the quantities by which different species of VOCs are produced by soil, or how effectively belowground VOCs are released into the atmosphere from soil remains largely unknown. This is the first published study that measures belowground VOC concentrations at different depths in a podzol combined with simultaneous soil surface flux measurements in a boreal coniferous forest. More than 50 VOCs, dominated by monoterpenes and sesquiterpenes, were detected in the air space in the soil during  two measurement campaigns. Organic forest soil was a significant monoterpene source as it contained fresh isoprenoid-rich litter, and the concentrations of monoterpenes were comparable to the VOC concentrations in the air above the coniferous forest. Belowground monoterpene concentrations were largely decoupled from forest floor monoterpene fluxes; thus, it seems that production processes and storages of VOCs partly differ from those VOCs that are simultaneously emitted from the soil surface. Relatively high isoprenoid concentrations were measured under snow cover, which indicates that snow and ice cover hinders gas diffusion and causes belowground accumulation of VOCs when the activity of vegetation is very low.

**1 Introduction**

Soil and understorey vegetation emit VOCs and these emissions are released from the diverse storages and processes (Hayward et al., 2001; Smolander et al., 2006; Leff and Fierer, 2008; Bäck et al., 2010; Aaltonen et al., 2011; Faubert et al, 2012, and Mäki et al., 2017). These studies reported that VOCs are produced by understorey vegetation, roots, decomposition processes, soil microbes, and vegetative litter concentrated in the organic soil layer. Microbes produce mono- and sesquiterpenes actively in their metabolism (Schulz and Dickschat 2007, Yamada et al., 2015). Organic soil layers can be a substantial source of VOCs due to the high abundance and activity of autotrophic and heterotrophic microbes. They drive decomposition processes where

easily available carbon is utilised for microbial metabolism and VOCs may be produced either actively as secondary metabolites or as by-products in the decomposition process.

VOCs have a crucial role in soils as infochemicals (Insam and Seewald, 2010, Schulz and Dickschat, 2007) by transmitting messages between soil organisms. Soil temperature and humidity influence many physical and biological

5    processes related to VOC formation in soils (Asensio et al., 2007, Aaltonen et al., 2013). In soils, warming climate can affect VOC synthesis by mediating decomposition processes, whereby microbial enzyme activity is regulated by soil water content and temperature (Davidson and Janssens, 2006).

10   ~~SOA is formed in the atmosphere from condensed oxidation products of VOCs, and SOA particles contribute to cloud formation and affect the Earth's radiation budget by scattering and absorbing solar radiation (Arneth et al., 2010, Virtanen et al., 2010, Mahowald, 2011). This outcome is opposite to the effect of greenhouse gases, which is warming the climate. Warming can change vegetation cover and almost double VOC emissions from subarctic and arctic plants (Faubert et al., 2010, Kramshøj et al., 2016). Warming can also affect VOC synthesis in soils by mediating decomposition processes, whereby~~

15    Soil water content impacts upon the transport and diffusion of organic compounds (Skopp et al., 1990, Zhong et al., 2014) and VOC emissions from vegetation (Svendsen et al., 2016), whereas temperature affects gas volatilization and diffusion . Soil water content also affects the decomposition of soil organic matter i.e. biological processes (Davidson and Janssens, 2006). Soil temperature and water content should be measured in parallel with belowground VOC

20   concentrations and soil surface flux measurements to study how effectively VOCs from soils are released into the atmosphere from very complex structured podzol soils. Warming can also change vegetation cover and affect belowground VOC production of plants (Faubert et al., 2010, Kramshøj et al., 2016), which can almost double VOC emissions from subarctic and arctic plants. Soil VOC production contributes atmospheric chemistry, because isoprene, monoterpenes, and especially sesquiterpenes have a precursor potential for secondary organic aerosol (SOA) formation.

25         The VOCs in soils have also been suggested to have an effect on biological interactions, although the quantities and functions of compounds in soils are largely unknown (Tholl et al., 2006). VOCs can promote plant growth, control the nitrogen cycle, affect microbial metabolism and transmit long-distance communication between different decomposers (Insam and Seewald, 2010, Asensio et al., 2012, Peñuelas et al., 2014, Tahir et al., 2017). A deeper understanding on the dynamics of soil processes and the roles of different soil components to VOC formation is needed (Asensio et al., 2007, Leff and Fierer,

30   2008; Gray et al., 2010). The wintertime dynamics of soil VOC production is especially interesting, as activity of the vegetation during the snow cover period is low, but the concentrations in soil and inside the snowpack can be quite high (Aaltonen et al., 2012). This is probably due to snow and ice cover that hinders diffusion of VOCs produced by microbial metabolism in snow bed, especially close to soil surface.

[revised manuscript text omitted]
. 2). Before permeability measurements with the wetted collector, it was wetted with ultrapure water. Also the gas flow was humidified. The PTR-MS enabled fast response monitoring of the diffusion of VOC mixture. The field conditions did not allow the implementation of the PTR-MS for continuous VOC measurements, thus only an adsorbent tube collection method was used. The 15-min break between individual samplings was used to stabilize VOC concentration between the gas collector and surrounding soil air. All the VOC standard compounds permeate the collector easily and concentrations reach a constant level in order of minutes (maximum 7 minutes) also with the wetted collector (Fig. 3). α-Pinene was the heaviest compound in the VOC gas mixture, and consequently its diffusion through the wall of the collector was the slowest of the VOCs measured. In contrast methanol peaked immediately after introducing the gas mixture into the glass bottle, but after that it stabilised quickly. It can be assumed therefore that stabilization of sesquiterpenes would take longer, since they are heavier than monoterpenes, thus the 15-min break time was chosen.

[revised manuscript text omitted]

**2.43 VOC and $CO_2$ flux measurements and supporting data**

Soil collars for VOC and $CO_2$ flux measurements were placed next (20–50 cm) in the five VOC measurement pits (Table 2). Soil collars were placed in March, 2016, and the measurements were started in April, 2016. Isoprenoid and oxygenated VOC fluxes were measured using a dynamic enclosure chamber technique as described by Mäki et al. (2017). The headspace (height 40 cm, chamber volume 10 L) was a glass chamber placed for measurements on permanently installed soil collars (height 7 cm, diameter 21.7 cm). We flushed the chamber headspace for 30 minutes to equilibrate the measurement system. During the chamber enclosure, we continuously pushed (1 l min$^{-1}$) filtered (active carbon trap and MnO2-coated copper net) ambient air into the chamber headspace and sampled the incoming and outgoing air for 1.5–2 hours through two Tenax TA-Carboback-B adsorbent tubes (flow rate 0.1-0.15 l min$^{-1}$). The dynamic enclosure method was previously tested in field conditions using standard gas with known VOC concentrations and quadrupole-PTR-MS (Kolari et al., 2012). The chamber system underestimates the artificially generated VOC emission rates at varying degree: for isoprene, monoterpene and many oxygenated VOCs the underestimation is 5-30%. The most uncertainties originate from adsorption of VOCs to moist or reactive surfaces, which are unavoidable when the enclosure contains living plant material. We estimated the flux rate ($E$, µg m$^{-2}$ h$^{-1}$) of each VOC for soil area (area inside to collar, m$^2$) and time ($h$) using Eq. (1):

$$E = \left(C_{out} - C_{in}\right)\frac{F_{chamber}}{1000}\frac{60}{A}, \qquad\qquad\qquad (1)$$

where $C_{in}$ is the ingoing air concentration (µg m$^{-3}$) and $C_{out}$ is the outgoing air concentration (µg m$^{-3}$), $F_{chamber}$ (m$^3$ min$^{-1}$) is the filtered air that was pushed into the chamber headspace, and $A$ (m$^2$) is the soil surface area covered by the soil collar.

[revised manuscript text omitted]

concentrations of the organic soil ($R^2$=0.62, $p<0.050$) and of the A-horizon ($R^2$=0.72, $p<0.010$) in pit three (Appendix Table A5).  Ssesquiterpene  and OVOCs concentrations were decoupled from soil surface fluxes (Fig. 5). The monoterpene flux correlated with the chamber temperature from summer ($R^2$=0.43, $p<0.05$) to autumn ($R^2$=0.62, $p<0.01$) and with the $CO_2$ flux in autumn ($R^2$=0.79, $p<0.001$) (Appendix Table A6). The sesquiterpene flux also correlated with the chamber temperature in spring ($R^2$=0.52, $p<0.05$) and autumn ($R^2$=0.47, $p<0.05$) and with the $CO_2$ flux from summer ($R^2$=0.57, $p<0.05$) to autumn ($R^2$=0.63, $p<0.01$) (Appendix Table A6). There was no correlation between the monoterpene and sesquiterpene fluxes and the soil water content (data not shown). The monoterpene concentration in the A-horizon correlated with the $CO_2$ flux in autumn ($R^2$=0.76, $p<0.01$) (Appendix Table A7). There was also a correlation between the sesquiterpene concentration in the O-horizon and the $CO_2$ flux in autumn ($R^2$=0.68, $p<0.05$) (Appendix Table A7).

The belowground vertical monoterpene concentrations were also uncoupled to the ambient air concentrations measured about 30cm from the soil surface using the proton-transfer reaction mass-spectrometer (quadrupole-PTR-MS) from August to November in 2016 (Fig. 6). The ambient air concentrations were 0.3 to 17 per cent of the monoterpene concentrations in the O-horizon.

[revised manuscript text omitted]

Schulz, S, and Dickschat, J.S. Bacterial volatiles: the smell of small organisms. Natural product reports, 24, p.814–842, 2007.

Smolander, A., Ketola, R.A., Kotiaho, T., Kanerva, S., Suominen, K. and Kitunen, V., 2006. Volatile monoterpenes in soil atmosphere under birch and conifers: Effects on soil N transformations. Soil Biology & Biochemistry, 38: 3436–3442, doi:10.1016/j.soilbio.2006.05.019, 2006.

Svendsen, S.H., Lindwall, F., Michelsen, A. and Rinnan, R.: Biogenic volatile organic compound emissions along a high arctic soil moisture gradient. Science of The Total Environment, 573, pp.131-138, doi:10.1016/j.scitotenv.2016.08.100, 2016.

Tahir, H.A., Gu, Q., Wu, H., Raza, W., Hanif, A., Wu, L., Colman, M.V. and Gao, X.: Plant growth promotion by volatile organic compounds produced by Bacillus subtilis SYST2. Frontiers in Microbiology, 8, p. 171, doi:10.3389/fmicb.2017.00171, 2017.

Tarvainen, V., Hakola, H., Hellén, H., Bäck, J., Hari, P. and Kulmala, M.: Temperature and light dependence of the VOC emissions of Scots pine. Atmospheric Chemistry and Physics, 5: 989–998, doi:10.5194/acp-5-989-2005, 2005.

Tholl, D., Boland, W., Hansel, A., Loreto, F., Röse, U.S.R. and Schnitzler, J.-P.: Practical approaches to plant volatile analysis. The Plant Journal, 45: 540–560, doi:10.1111/j.1365-313X.2005.02612.x, 2006.

Timonen, S, Sinkko, H, Sun, H, Sietiö, O.-M., Rinta-Kanto, J.M., Kiheri, H., Heinonsalo, J.: Ericoid roots and mycospheres govern plant-specific bacterial communities in boreal forest humus. Microbial Ecology, 1–15, doi:10.1007/s00248-016-0922-6, 2017.

Virtanen, A., Joutsensaari, J., Koop, T., Kannosto, J., Yli-Pirilä, P., Leskinen, J., Mäkelä, J.M., Holopainen, J. K., Pöschl, U., Kulmala, M. and Worsnop, D.R.: An amorphous solid state of biogenic secondary organic aerosol particles. Nature, 467(7317), pp.824-827, doi:10.1038/nature09455, 2010.

Weikl, F., Ghirardo, A., Schnitzler, J.P. and Pritsch, K.: Sesquiterpene emissions from Alternaria alternata and Fusarium oxysporum: Effects of age, nutrient availability, and co-cultivation. Nature, Scientific reports, 6, doi:10.1038/srep22152, 2016.

Yamanaka, K., Reynolds, K. A., Kersten, R. D., Ryan, K. S., Gonzalez, D. J., Nizet, V., Dorrestein, P. C., and Moore, B. S. Terpene synthases are widely distributed in bacteria. Proceedings of the National Academy of Sciences of the United States of America, 111, 1957–1962, doi:10.1073/pnas.1422108112, 2015.

Zhong, L., Cantrell, K., Mitroshkov, A. and Shewell, J.: Mobilization and transport of organic compounds from reservoir rock and caprock in geological carbon sequestration sites. Environmental Earth Sciences, 71(9), pp.4261–4272. doi:10.1007/s12665-013-2823-z, 2014.

**Figures and Tables**

**Table 1:** Soil characteristics of the measurement site. The depth of lower horizon border (cm), volume weight (g cm$^{-3}$), particle size of clay (%), silt (%), and sand (%), N-content (mg g$^{-1}$), C-content (mg g$^{-1}$), and pH value (in CaCl$_2$) of the measurement pits for the different soil horizons at the SMEAR II station in 1995. Values are means of measurement pits 1, 4, and 5.

| Horizon | Depth of lower horizon border | Volume weight (g cm$^{-3}$) | Rocks (% of weight) | Particle size (% of weight) | | |
|---|---|---|---|---|---|---|
| | | | | clay | silt | sand |
| **O** | 0.00 | | | | | |
| **A** | 6.54 | 0.75 | 28.61 | 5.65 | 13.72 | 52.02 |
| **B** | 26.83 | 0.86 | 27.68 | 6.72 | 13.01 | 52.60 |

| Horizon | N-content (mg g$^{-1}$) | C-content (mg g$^{-1}$) | pH in CaCl$_2$ | | | |
|---|---|---|---|---|---|---|
| C | 71.14 | 1.27 | 36.25 | 6.91 | 12.56 | 44.28 |

| Horizon | N-content (mg g$^{-1}$) | C-content (mg g$^{-1}$) | pH in CaCl$_2$ |
|---|---|---|---|
| O | 13.46 | 355.68 | 3.39 |
| A | 1.02 | 32.23 | 3.53 |
| B | 1.07 | 23.51 | 4.36 |
| C | 0.13 | 4.15 | 4.49 |

[Figure]

**Figure 1: Measurement set-up for soil VOC concentration profiles a) in 2008- 2011 and b) in 2016.**

[Figure]

**Figure 2: Set-up for the permeability tests of the sintered gas collectors. The sintered gas collector means that PTFE tube is closed on one side.**

[Figure]

**Figure 3: Results of the permeability tests of the PTFE collector with the five VOCs. A permeability test was used to monitor how fast VOCs permeate into the gas collector and to determine how fast VOC concentrations stabilize between the air inside and outside the collector. Panel a) shows the results with dry collector and panel b) with a wetted collector. Vertical line shows the time point when the introduction of the VOC standard began.**

**Table 2:** Soil depth (cm) and soil surface coverages (%) of ericoid shrubs, mosses, grasses, and non-vegetative surface on the different measurement pits in 2016.

| Pit | Soil depth (cm) | Ericoid shrubs (%) | Mosses (%) | Grasses (%) | Non-vegetative surface (%) |
|-----|-----------------|--------------------|------------|-------------|----------------------------|
| 1 | 50 | 25 | 10 | ¾ | 65 |
| 2 | 60 | 5 | ¾ | 5 | 90 |
| 3 | 80 | 25 | 20 | ¾ | 55 |
| 4 | 130 | 15 | 30 | 18 | 37 |
| 5 | 160 | 7 | 2 | 8 | 83 |

[Figure]

[Figure]

**Figure 4: Isoprene and individual monoterpene (a) and sesquiterpene (b) concentrations (µg m⁻³) from the different soil horizons (O (N=52), A (N=65), B (N=65), and C (N=65)) in 2016. Concentrations are means and error bars are standard error of the whole data for each soil horizon. SQT1 was not identified.**

[Figure]

[Figure]

**Figure 5: The mean a) isoprene, b) monoterpene, and c) sesquiterpene fluxes (µg m$^{-2}$ h$^{-1}$) from the forest floor and concentration (µg m$^{-3}$) from  each soil horizon from April to December in 2016. Error bars are standard error of the four (O-horizon) or five (A--horizon) gas collectors.**

**Table 3:** The soil depth (cm), the mean monoterpene, sesquiterpene, oxygenated VOCs (C4–C15 alcohols, carbonyls and acetates, methyl-2/3-furoates and α-pinene oxide) and $CO_2$ fluxes above the soil surface (µg m$^{-2}$ h$^{-1}$), chamber temperature (°C), soil temperature (°C, A-horizon), and soil water content (m$^3$ m$^{-3}$, A-horizon) from the measurement pits. Values are means (S.E.) of the whole dataset in 2016 (N= 6–13). The effect of soil horizon on fluxes and environmental conditions was tested with the Kruskal-Wallis test (p<0.050). Significant differences between the pits are indicated with different letters (Kruskal-Wallis test; p<0.050).

| Pit | Soil depth cm | Monoterpenes µg m$^{-2}$ h$^{-1}$ | Sesquiterpenes µg m$^{-2}$ h$^{-1}$ | OVOC µg m$^{-2}$ h$^{-1}$ | $CO_2$ flux µg m$^{-2}$ h$^{-1}$ | Chamber temperature C° | Soil temperature C° | Soil water content m$^3$ m$^{-3}$ |
|-----|-----|-----|-----|-----|-----|-----|-----|-----|
| 1 | 50 | 49.05[a] (30.30) | 4.56[a] (2.80) | 5.12[a] (3.72) | 0.15[a] (0.02) | 10.82[a] (2.28) | 8.29[a] (1.27) | 0.14[a] (0.01) |
| 2 | 60 | 19.71[a] (7.25) | 6.34[a] (5.50) | 1.84[a] (0.67) | 0.13[ac] (0.04) | 11.08[a] (1.99) | 8.67[a] (1.54) | 0.12[a] (0.09) |
| 3 | 80 | 20.73[a] (4.80) | 11.16[a] (10.91) | 0.59[a] (0.14) | 0.03[b] (0.02) | 9.33[a] (2.77) | 8.39[a] (1.32) | 0.34[b] (0.06) |
| 4 | 130 | 26.09[a] (8.72) | 8.08[a] (7.63) | 0.61[a] (0.26) | 0.08[c] (0.01) | 10.45[a] (3.04) | 7.72[a] (1.51) | 0.30[b] (0.02) |
| 5 | 160 | 61.90[a] (36.95) | 0.67[a] (0.25) | 0.96[a] (0.33) | 0.21[a] (0.07) | 12.24[a] (2.40) | 6.58[a] (1.20) | 0.78[c] (0.12) |

[Figure]

**Figure 6: The mean monoterpene concentration (µg m$^{-3}$) from the O- and A-horizon and from the ambient air from August to December in 2016. Error bars are standard error of the four (O-horizon) or five (A-horizon) gas collectors. Error bars of ambient air measurements are based on two measurement locations at the SMEAR II station.**

**Table 4:** Pearson correlation between the total monoterpene and sesquiterpene concentrations and soil temperature (°C) and water content (m$^3$ m$^{-3}$) from the O- and the B-horizon in 2008–2011 and from the different soil horizons in 2016. The significance level of p<0.100 (o), p<0.050 (*), p<0.010 (**), p<0.001 (***)) was used. VOC concentrations were measured from the different gas collectors in 2008–2011 and in 2016 (Fig. 1a and 1b).

| Year | Horizon | Correlation coefficient | N | P value | Correlation coefficient | N | P value |
|------|---------|-------------------------|---|---------|-------------------------|---|---------|
|      |         |                         |   |         |                         |   |         |

| | Soil temperature C° | Monoterpene concentration µg m$^{-3}$ | | | Sesquiterpene concentration µg m$^{-3}$ | | |
|---|---|---|---|---|---|---|---|
| 2008–2011 | O | **0.35** | 49 | 0.007** | **0.29** | 34 | 0.049* |
| 2008–2011 | B | **0.46** | 51 | 0.0004*** | 0.16 | 36 | 0.17 |
| 2016 | O | -0.01 | 52 | 0.52 | -0.32 | 52 | 0.97 |
| 2016 | A | **0.33** | 65 | 0.005** | 0.02 | 65 | 0.44 |
| 2016 | B | -0.21 | 65 | 0.98 | -0.27 | 65 | 0.98 |
| 2016 | C | -0.17 | 65 | 0.89 | -0.32 | 65 | 0.99 |

| | Soil water content m$^3$ m$^{-3}$ | Monoterpene concentration µg m$^{-3}$ | | | Sesquiterpene concentration µg m$^{-3}$ | | |
|---|---|---|---|---|---|---|---|
| 2008–2011 | O | -0.13 | 49 | 0.81 | -0.30 | 34 | 0.96 |
| 2008–2011 | B | -0.35 | 51 | 0.99 | -0.43 | 36 | 0.99 |
| 2016 | O | -0.01 | 52 | 0.53 | -0.11 | 52 | 0.75 |
| 2016 | A | -0.09 | 65 | 0.76 | **0.23** | 65 | 0.04* |
| 2016 | B | -0.09 | 65 | 0.75 | 0.03 | 65 | 0.42 |
| 2016 | C | -0.09 | 65 | 0.74 | **0.19** | 65 | 0.098º |

**Table 5:** Annual mean isoprenoid concentrations (S.E., µg m$^{-3}$) in soil. Note that the year 2008 consists of only one sampling in November. BDL means below detection limit of the VOC quantification.

| µg m$^{-3}$ | 2008 1 sampling | | 2009 9 samplings | | 2010 3 samplings | | 2011 4 samplings | |
|---|---|---|---|---|---|---|---|---|
| | 5 cm | 17cm | 5 cm | 17cm | 5 cm | 17cm | 5 cm | 17cm |
| isoprene | 0.01 | 4x10$^{-3}$ | 0.02 (0.01) | 0.01 (2x10$^{-3}$) | 0.06 (0.01) | 0.03 (3x10$^{-3}$) | 0.02 (0.02) | 0.01 (2x10$^{-3}$) |
| methyl butenol | 4x10$^{-3}$ | 3x10$^{-3}$ | 0.01 (0.01) | 3x10$^{-3}$ (1x10$^{-3}$) | 0.01 (3x10$^{-3}$) | 0.02 (0.01) | 0.01 (2x10$^{-3}$) | 4x10$^{-3}$(1x10$^{-3}$) |
| **Monoterpenes** | | | | | | | | |
| α-pinene | 0.63 | 0.60 | 1.88 (0.92) | 2.35 (0.99) | 1.37 (0.72) | 1.61 (1.13) | 2.88 (0.80) | 1.80 (0.54) |
| camphene | 0.02 | 0.02 | 0.07 (0.02) | 0.11 (0.06) | 0.06 (0.03) | 0.04 (0.02) | 0.04 (0.01) | 0.04 (0.02) |
| ß-pinene | 0.04 | 0.04 | 0.35 (0.24) | 0.26 (0.10) | 0.07 (0.03) | 0.43 (0.35) | 0.14 (0.04) | 0.11 (0.05) |
| Δ-3-carene | 0.84 | 0.67 | 2.56 (0.79) | 1.42 (0.35) | 1.34 (0.77) | 1.07 (0.70) | 2.92 (0.92) | 2.14 (0.61) |
| p-cymene | 0.06 | 0.04 | 0.07 (0.02) | 0.06 (0.01) | 0.09 (0.03) | 0.03 (0.01) | 0.06 (0.02) | 0.04 (0.01) |
| 1,8-cineol | BDL | BDL | BDL | BDL | 0.02 (3x10$^{-3}$) | 0.01 (1x10$^{-3}$) | 3x10$^{-3}$ (1x10$^{-3}$) | 3x10$^{-3}$ (1x10$^{-3}$) |
| limonene | 0.07 | 0.05 | 0.20 (0.05) | 0.16 (0.06) | 0.21 (0.05) | 0.11 (0.03) | 0.25 (0.08) | 0.18 (0.05) |
| terpinolene | BDL | 4x10$^{-3}$ | 0.12 (0.07) | 0.12 (0.03) | 0.19 (0.13) | 0.10 (0.07) | 0.04 (0.01) | 0.03 (0.01) |
| linalool | BDL | BDL | 3x10$^{-3}$ | 0.04 (0.02) | 0.11 | 0.01 | 0.01 (0.01) | 3x10$^{-3}$ |
| myrcene | BDL | BDL | BDL | BDL | BDL | BDL | 0.01 (1x10$^{-3}$) | 0.01 (2x10$^{-3}$) |
| nopinone | BDL | BDL | 0.01 (5x10$^{-3}$) | 0.02 (0.01) | 0.02 (3x10$^{-3}$) | 0.01 (2x10$^{-3}$) | 4x10$^{-3}$ (1x10$^{-3}$) | 4x10$^{-3}$ (2x10$^{-3}$) |
| bornylacetate | BDL | 0.001 | 0.05 (0.02) | 0.21 (0.10) | 0.02 (0.01) | 0.01 (2x10$^{-3}$) | 0.01 (0.01) | 3x10$^{-3}$ (1x10$^{-4}$) |

| | | | | | | | | |
|---|---|---|---|---|---|---|---|---|
| **Total monoterpenes** | 1.66 | 1.43 | 4.85 (1.83) | 4.30 (1.44) | 3.18 (1.60) | 3.18 (2.14) | 6.34 (1.84) | 4.35 (1.27) |
| **Sesquiterpenes** | | | | | | | | |
| longicyclene | BDL | BDL | 0.05 (0.02) | 0.16 (0.06) | 0.11 (0.05) | 0.01 ($2\times10^{-3}$) | 0.01($1\times10^{-3}$) | 0.01 ($2\times10^{-3}$) |
| iso-longifolene | $3\times10^{-3}$ | $4\times10^{-3}$ | 0.04 (0.01) | 0.17 (0.07) | 0.02 (0.01) | 0.01 ($2\times10^{-3}$) | $4\times10^{-3}$ ($1\times10^{-3}$) | $3\times10^{-3}$ ($1\times10^{-3}$) |
| ß-caryophyllene | BDL | BDL | 0.07 (0.02) | 0.21 (0.08) | 0.08 (0.05) | 0.05 (0.01) | 0.03 (0.01) | 0.02 (0.01) |
| aromadendrene | BDL | BDL | 0.06 (0.02) | 0.26 (0.11) | 0.04 (0.02) | 0.01 (0.01) | 0.06 (0.04) | 0.08 |
| α-humulene | BDL | BDL | 0.06 (0.02) | 0.21 (0.10) | 0.02 (0.002) | 0.01 (0.01) | $1\times10^{-3}$ ($3\times10^{-4}$) | 0.01 (0.01) |
| ß-farnesene | BDL | BDL | 0.09 (0.03) | 0.38 (0.11) | 0.03 (0.003) | 0.01 ($2\times10^{-3}$) | 0.01 ($2\times10^{-3}$) | 0.01 ($2\times10^{-3}$) |
| **Total sesquiterpenes** | 0.03 | $4\times10^{-3}$ | 0.28 (0.08) | 1.03 (0.36) | 0.22 (0.08) | 0.07 (0.02) | 0.06 (0.02) | 0.04 (0.01) |

[Figure]

Figure 7: **The monthly mean monoterpene concentration (μg m$^{-3}$) and soil temperature (C°) for the O-horizon and the B-horizon during the summer months in 2009- 2011 and in 2016. Error bars are standard error of the three gas collectors in 2009, 2010, and 2011, and four (O-horizon) or five (B-horizon) gas collectors in 2016.**

10 **Appendix**

**Table A1:** Measurement depths (cm) of the different soil horizons (O, A, B, and C) from eight measurement pits. The campaign measurements were made from three soil pits in 2008- 2011 (0 cm being the surface of O-horizon) and from five pits in 2016 (0 cm being the surface of mineral soil).

2008–2011

| Pit 1 | | Pit 2 | | Pit 3 | |
|---|---|---|---|---|---|
| Horizon | Depth | Horizon | Depth | Horizon | Depth |
| O | -5 | H | -5 | O | -5 |
| B | -17 | B | -17 | B | -17 |

2016

| Pit 1 | | Pit 2 | | Pit 3 | | Pit 4 | | Pit 5 | |
|---|---|---|---|---|---|---|---|---|---|
| Horizon | Depth | Horizon | Depth | Horizon | Depth | Horizon | Depth | Horizon | Depth |
| O | -2 | | | O | -3 | O | -3 | O | -2 |
| A | 2 | A | 1 | A | 7 | A | 3 | A | 5 |
| B | 9 | B1 | 13 | B | 29 | B1 | 15 | B1 | 20 |
| | | B2 | 35 | | | B2 | 27 | B2 | 33 |
| C | 23 | C | 63 | C | 62 | C1 | 50 | C | 57 |
| | | | | | | C2 | 97 | | |

**Table A2:** Measurement months for VOC concentrations measurements in 2008- 2011 period and for VOC concentration and chamber flux measurements in 2016.

| Year | Measurement months | Sampling times |
|---|---|---|
| 2008 | November | 1 |
| 2009 | April, May, June, July, October, November, and December | 1 |
| | September | 2 |
| 2010 | July, August, and November | 1 |
| 2011 | May, June, August, and October | 1 |
| 2016 | April, May, June, November, and December | 1 |
| | July, August, September, and October | 2 |

**Table A3:** The detection limit of measured VOCs in concentration ($\mu g\ m^{-3}$) and flux measurements ($\mu g\ m^{-2}\ h^{-1}$). Compounds marked with ($^{\dagger}$) are only tentatively identified and quantified.

| Compound | Concentration 2008–2011 | Concentration 2016 | Flux | Compound | Concentration 2016 | Flux |
|---|---|---|---|---|---|---|

| | | | | | | |
|---|---|---|---|---|---|---|
| isoprene | $5\times10^{-3}$ | $5\times10^{-3}$ | 0.01 | **OVOCs** | | |
| 2-methyl-3-buten-2-ol | $2\times10^{-3}$ | | | geraniol | $4\times10^{-3}$ | 0.12 |
| **Monoterpenes** | | | | methyl-12-furoate | $1\times10^{-3}$ | 0.01 |
| α-pinene | $1\times10^{-3}$ | 0.03 | 0.18 | α-bisabolol | 0.01 | 0.04 |
| camphene | $1\times10^{-3}$ | 0.02 | 0.01 | verbenone | 0.01 | 0.02 |
| Δ-3-carene | $2\times10^{-3}$ | 0.06 | 0.06 | nuciferol | $2\times10^{-4}$ | 0.01 |
| ß-pinene | $2\times10^{-3}$ | 0.01 | 0.01 | methy-3-furoate | 0.01 | 0.02 |
| myrcene | $2\times10^{-3}$ | 0.01 | 0.01 | 1-butanol | 0.06 | 0.18 |
| 1,8-cineol | $3\times10^{-3}$ | 0.01 | 0.01 | isopropanol | $2\times10^{-3}$ | 0.01 |
| linalool | 0.01 | 0.01 | 0.32 | 2-butanone | 0.02 | 0.04 |
| limonene | $2\times10^{-3}$ | $5\times10^{-3}$ | 0.10 | 1-penten-3-ol | $2\times10^{-3}$ | $5\times10^{-3}$ |
| p-cymene | $1\times10^{-3}$ | 0.02 | 0.02 | 1-pentanol | 0.01 | 0.01 |
| terpinolene | 0.01 | $2\times10^{-3}$ | $2\times10^{-3}$ | 2-methyl-2-buten-1-ol | $1\times10^{-3}$ | $2\times10^{-3}$ |
| **Sesquiterpenes** | | | | butyl acetate | $1\times10^{-3}$ | $5\times10^{-3}$ |
| bornylacetate | $3\times10^{-3}$ | $2\times10^{-3}$ | $4\times10^{-3}$ | cis-3-hexen-1-ol | 0.01 | 0.02 |
| longicyclene | $1\times10^{-3}$ | $1\times10^{-3}$ | $2\times10^{-3}$ | trans-3-hexen-1-ol | 0.01 | 0.01 |
| isolongifolene | $1\times10^{-3}$ | $2\times10^{-3}$ | $5\times10^{-3}$ | trans-2-hexen-1-ol | $2\times10^{-3}$ | $4\times10^{-3}$ |
| ß-caryophyllene | 0.01 | 0.01 | 0.01 | 1-hexanol | $1\times10^{-3}$ | 0.01 |
| aromadendrene[†] | $3\times10^{-3}$ | 0.01 | $5\times10^{-3}$ | cis-2-hexen-1-ol | 0.01 | 0.02 |
| α-gurjunene | | 0.01 | 1.48 | 1-octen-3-ol | $2\times10^{-3}$ | 0.03 |
| α-humulene | $4\times10^{-3}$ | $2\times10^{-3}$ | 0.09 | 6-methyl-5-heptene-2-one | $5\times10^{-3}$ | 0.03 |
| ß-farnesene | $2\times10^{-3}$ | 0.02 | 0.02 | cis-3-hexenyl acetate | 0.03 | 0.06 |
| isocaryophyllene[†] | | $4\times10^{-3}$ | 0.01 | hexyl acetate | 0.01 | 0.012 |
| SQT1[†] | | 0.01 | 0.02 | trans-2-Hexenyl acetate | $3\times10^{-3}$ | 0.005 |
| α-buinesene[†] | | $1\times10^{-3}$ | 0.02 | α-pinenepxide | 0.01 | 0.01 |
| γ-muurolene[†] | | $2\times10^{-3}$ | $1\times10^{-3}$ | | | |
| α-bisabolene[†] | | $3\times10^{-4}$ | $1\times10^{-3}$ | | | |
| β-himachalene[†] | | $1\times10^{-3}$ | $3\times10^{-3}$ | | | |
| α-muurolene[†] | | $1\times10^{-3}$ | 0.01 | | | |
| Δ-cadinene | | | 0.15 | | | |

**Table A4:** Isoprene, monoterpenes, sesquiterpenes and oxygenated VOC concentrations, means (S.E. µg m$^{-3}$) of the different soil horizons (O (N=52), A (N=65), B (N=65), and C (N=65)) in 2016. Concentrations are means (S.E.) and medians of the whole data (BDL = below the detection limit of the VOC quantification). The effect of soil horizon on concentrations was tested with the Kruskal-Wallis test ($p<0.050$). Significant differences in concentrations between the horizons are indicated with different letters (Kruskal-Wallis test; $p<0.050$). Compounds marked with ([†]) are only tentatively identified and quantified.

| Concentration | O | A | B | C |
|---|---|---|---|---|
| isoprene | 0.02[a] (2x10[-3]) 0.01 | 0.04[a] (0.03) 0.01 | 0.02[a] (2x10[-3]) 0.01 | 0.02[a] (3x10[-3]) 0.01 |
| α-pinene | 320.4[a] (135.8) 28.0 | 71.0[a] (22.5) 5.0 | 108.6[b] (103.6) 1.6 | 6.7[b] (2.2) 1.5 |
| camphene | 8.8[a] (2.6) 0.7 | 1.4[a] (0.4) 0.1 | 0.6[b] (0.6) 0.03 | 0.3[b] (0.1) 0.01 |
| Δ-3-carene | 58.8[ac] (22.8) 4.3 | 16.9[a] (5.2) 2.6 | 30.2[bc] (29.5) 1.2 | 2.8[b] (0.8) 1.3 |
| ß-pinene | 14.7[a] (4.4) 1.2 | 2.4[b] (1.2) 0.2 | 0.4[c] (0.3) 0.1 | 0.3[c] (0.1) 0.1 |
| myrcene | 18.1[a] (8.0) 1.9 | 1.7[b] (0.5) 0.1 | 1.7[c] (1.0) 0.05 | 0.7[c] (0.5) 0.04 |
| 1,8-cineol | 0.08[a] (0.04) 0.02 | 0.3[a] (0.2) 0.04 | 0.2[a] (0.1) 0.02 | 0.1[a] (0.03) 0.01 |
| linalool | 1.4[a] (1.0) 0.1 | 1.5[a] (0.6) 0.1 | 1.1[a] (0.6) 0.1 | 0.6[a] (0.3) 0.1 |
| limonene | 16.4[ab] (7.4) 0.7 | 3.5[a] (1.0) 1.0 | 2.4[ab] (1.1) 0.6 | 1.9[b] (1.0) 0.4 |
| p-cymene | 1.8[a] (1.4) 0.1 | 0.4[ab] (0.1) 0.1 | 0.2[ab] (0.1) 0.1 | 0.2[b] (0.04) 0.1 |
| terpinolene | 3.1[a] (1.4) 0.2 | 0.4[ab] (0.2) 0.03 | 0.4[bc] (0.4) 0.02 | 0.05[c] (0.02) 0.01 |
| **Sum of monoterpenes** | **422.2[a] (157.9) 35.6** | **96.0[a] (28.8) 9.7** | **144.6[b] (133.7) 4.2** | **12.8[b] (3.7) 4.1** |
| bornylacetate | 0.01[a] (0.01) 0.01 | 0.1[a] (0.1) 0.01 | 0.01[a] (1x10[-3]) 4x10[-3] | 0.01[a] (1x10[-3]) 4x10[-3] |
| longicyclene | 0.02[a] (0.02) 0.01 | 0.01[a] (5x10[-3]) 3x10[-3] | 5x10[-3a] (1x10[-3]) 3x10[-3] | 0.01[a] (3x10[-3]) 2x10[-3] |
| isolongifolene | 0.3[a] (0.3) 0.01 | 0.04[a] (0.03) 0.02 | 0.05[a] (0.02) 0.03 | 0.1[a] (0.03) 0.05 |
| ß-caryophyllene | 0.1[a] (0.1) 0.02 | 0.4[a] (0.2) 0.02 | 0.6[a] (0.4) 0.02 | 0.2[a] (0.2) 0.03 |
| aromadendrene[†] | 0.02[a] (5x10[-3]) 0.02 | 0.01[a] 0.01 | 0.03[a] (0.01) 0.03 | 0.03[a] (0.01) 0.01 |
| α-gurjunene | 3.4[a] (2.4) 0.1 | 25.7[a] (22.9) 0.2 | 3.1[a] (1.3) 0.2 | 3.0[a] (1.5) 0.1 |
| α-humulene | 0.4[a] (0.2) 5x10[-3] | 2.9[a] (1.6) 0.3 | 5.3[a] (4.2) 0.3 | 0.6[a] (0.3) 0.01 |
| ß-farnesene | 0.4[a] (0.3) 0.1 | 0.03[a] (0.01) 0.03 | 1.9[a] (1.5) 0.03 | 0.4[a] (0.4) |
| isocaryophyllene[†] | 0.01[a] (3x10[-3]) 0.01 | 0.01[a] (3x10[-3]) 0.01 | 3x10[-3a] (2x10[-3]) 0.01 | 0.01[a] (1x10[-3]) |
| SQT1[†] | 0.03[a] (0.01) 0.03 | 0.02[a] (4x10[-3]) 0.02 | 0.04[a] (3x10[-3]) 0.04 | 0.02[a] 0.02 |
| α-buinesene[†] | 0.03[a] (0.02) 0.01 | 0.5[a] (0.5) 0.02 | 0.1[a] (0.05) 4x10[-3] | 0.3[a] (0.1) 0.2 |
| γ-muurolene[†] | 0.02[ab] (0.02) 0.02 | 0.01[a] (0.01) 0.01 | 4x10[-3b] (1x10[-3]) 3x10[-3] | 0.01[ab] (4x10[-3]) 5x10[-3] |
| α-bisabolene[†] | 0.01[a] (0.01) 0.01 | 0.5[a] (0.3) 0.02 | 0.3[a] (0.3) 0.01 | 0.3[a] (0.2) 0.2 |
| β-himachalene[†] | 0.2[a] (0.1) 0.02 | 1.3[a] (1.0) 0.3 | 0.3[a] (0.1) 0.2 | 0.4[a] (0.2) 0.2 |
| α-muurolene[†] | 0.07[a] (0.04) 0.01 | 0.1[a] (0.03) 0.05 | 0.1[a] (0.03) 0.1 | 0.03[a] (0.01) 0.02 |
| Δ-cadinene[†] | 0.01[a] (4x10[-3]) 2x10[-3] | 0.1[a] (0.1) 0.01 | 0.01[a] (2x10[-3]) 3x10[-3] | 0.01[a] (0.01) 4x10[-3] |
| **Sum of sesquiterpenes** | **2.4[a] (1.4) 0.2** | **15.1[a] (12.1) 0.2** | **4.3[a] (2.0) 0.3** | **2.1[a] (0.9) 0.2** |
| geraniol | 0.1[a] (0.1) 0.02 | 2.3[a] (2.3) 0.01 | 0.1[a] (0.03) 0.01 | 0.2[a] (0.1) 0.01 |

| | | | | |
|---|---|---|---|---|
| methyl-2-furoate | $5\times10^{-3a}$ ($1\times10^{-3}$) $\underline{4\times10^{-3}}$ | $0.1^a$ (0.1) $\underline{3\times10^{-3}}$ | $5\times10^{-3a}$ ($1\times10^{-3}$) $3\times10^{-3}$ | $4\times10^{-3a}$ ($1\times10^{-3}$) $4\times10^{-3}$ |
| α-bisabolol | $0.01^a$ $\underline{0.01}$ | $0.1^a$ (0.03) $\underline{0.04}$ | $0.03^a$ (0.01) $\underline{0.02}$ | $0.03^a$ (0.01) $\underline{0.02}$ |
| verbenone | $2.6^a$ (1.7) $\underline{0.2}$ | $0.1^{ab}$ (0.1) $\underline{0.1}$ | $0.03^{ab}$ $\underline{0.03}$ | $0.02^b$ ($3\times10^{-3}$) $\underline{0.02}$ |
| nuciferol | $0.02^a$ (0.01) $\underline{0.02}$ | $0.03^a$ (0.01) $\underline{0.02}$ | $0.02^a$ ($4\times10^{-3}$) $\underline{0.01}$ | $0.02^a$ ($4\times10^{-3}$) $\underline{0.01}$ |
| methy-3-furoate | BDL | BDL | $0.02^a$ $\underline{0.02}$ | BDL |
| 1-butanol | $1.9^a$ (1.0) $\underline{0.3}$ | $9.0^a$ (6.8) $\underline{0.4}$ | $2.3^a$ (1.0) $\underline{0.4}$ | $1.7^a$ (0.6) $\underline{0.5}$ |
| isopropanol | $0.01^a$ ($1\times10^{-3}$) $\underline{0.01}$ | $0.01^a$ ($2\times10^{-3}$) $\underline{0.01}$ | $0.02^a$ (0.01) $\underline{0.01}$ | $0.02^a$ (0.01) $\underline{0.01}$ |
| 1-butanone | $0.2^a$ (0.1) $\underline{0.03}$ | $0.7^a$ (0.5) $\underline{0.03}$ | $0.1^a$ (0.1) $\underline{0.04}$ | $0.2^a$ (0.1) $\underline{0.03}$ |
| penten-3-ol | $0.5^a$ (0.2) $\underline{0.4}$ | $1.2^a$ (0.8) $\underline{0.02}$ | $0.3^a$ (0.1) $\underline{0.1}$ | $0.2^a$ (0.1) $\underline{0.1}$ |
| 1-pentanol | $0.2^a$ (0.07) $\underline{0.03}$ | $0.6^b$ (0.3) $\underline{0.1}$ | $0.1^{ab}$ (0.02) $\underline{0.04}$ | $0.1^a$ (0.03) $\underline{0.03}$ |
| 2-methyl-2-buten-1-ol | $0.04^a$ (0.03) $\underline{0.01}$ | $0.1^a$ (0.02) $\underline{0.03}$ | $0.01^a$ ($3\times10^{-3}$) $\underline{0.01}$ | $0.02^a$ (0.01) $\underline{0.01}$ |
| butyl acetate | $0.01^a$ ($2\times10^{-3}$) $\underline{0.01}$ | $0.1^a$ (0.04) $\underline{0.01}$ | $0.01^a$ ($2\times10^{-3}$) $\underline{0.01}$ | $0.01^a$ ($2\times10^{-3}$) $\underline{0.01}$ |
| cis-3-hexen-1-ol | BDL | BDL | BDL | BDL |
| trans-3-hexen-1-ol | BDL | $0.02^a$ $\underline{0.02}$ | $1.8^a$ (1.8) $\underline{1.8}$ | $0.01^a$ (0.01) $\underline{0.01}$ |
| trans-2-hexen-1-ol | $0.4^a$ (0.3) $\underline{0.01}$ | $0.1^a$ (0.02) $\underline{0.01}$ | $0.4^a$ (0.3) $\underline{0.01}$ | $0.2^a$ (0.1) $\underline{0.03}$ |
| 1-hexanol | $0.02^a$ ($5\times10^{-3}$) $\underline{0.01}$ | $0.04^a$ (0.03) $\underline{0.01}$ | $0.1^a$ (0.1) $\underline{0.01}$ | $0.02^a$ ($4\times10^{-3}$) $\underline{0.01}$ |
| cis-2-hexen-1-ol | $0.01^a$ ($1\times10^{-3}$) $\underline{0.01}$ | $0.03^a$ (0.02) $\underline{0.01}$ | $0.02^a$ (0.01) $\underline{0.01}$ | $0.1^a$ (0.04) $\underline{0.02}$ |
| 1-octen-3-ol | $0.2^{ab}$ (0.1) $\underline{0.03}$ | $1.4^a$ (1.3) $\underline{0.03}$ | $1.3^{ab}$ (1.3) $\underline{0.03}$ | $0.25^b$ (0.1) $\underline{0.02}$ |
| 6-methyl-5-heptene-2-one | $2.5^a$ (1.4) $\underline{0.1}$ | $0.5^a$ (0.2) $\underline{0.1}$ | $0.2^b$ (0.1) $\underline{0.04}$ | $0.2^b$ (0.1) $\underline{0.03}$ |
| cis-3-hexenyl acetate | $1.2^a$ (0.8) $\underline{0.1}$ | $1.2^a$ (0.7) $\underline{0.1}$ | $1.0^a$ (0.5) $\underline{0.1}$ | $1.9^a$ (0.8) $\underline{0.1}$ |
| hexyl acetate | $0.03^a$ (0.01) $\underline{0.04}$ | $1.1^a$ (1.1) $\underline{0.03}$ | $0.05^a$ (0.02) $\underline{0.03}$ | $0.02^a$ ($4\times10^{-3}$) $\underline{0.02}$ |
| trans-2-hexenyl acetate | $0.3^a$ (0.3) $\underline{0.02}$ | $0.02^a$ (0.01) $\underline{4\times10^{-3}}$ | $0.4^a$ (0.3) $\underline{5\times10^{-3}}$ | $0.2^a$ (0.2) $\underline{5\times10^{-3}}$ |
| α-pineneoxide | $1.0^a$ (0.7) 0.2 | $1.3^a$ (0.6) $\underline{0.1}$ | $0.4^a$ (0.2) $\underline{0.1}$ | $0.04^a$ (0.03) $\underline{0.04}$ |
| **Sum of OVOCs** | **$5.4^a$ (2.5) $\underline{1.2}$** | **$10.0^a$ (6.7) $\underline{0.8}$** | **$4.0^a$ (1.9) $\underline{0.5}$** | **$2.8^a$ (1.1) $\underline{0.6}$** |

[Figure]

**Figure A1: a) Soil temperature (C°) measured at the O-horizon (0–5 cm) and B-horizon (10–28 cm) over the 2008-2011 period and b) soil temperature measured at the different soil horizons in 2016.**

[Figure]

**Figure A2: a) Soil water content (m³ m⁻³) measured at the O-horizon (0–5 cm) and B-horizon (10–28 cm) in years 2008- 2011 and b)**
**soil water content (m³ m⁻³) measured at the different soil horizons in 2016. High wintertime variation is explained the change from**
5   **liquid water to solid phase by freezing as TDR measurement method is highly sensitive for freezing.**

**Table A5**: Pearson correlations between the total monoterpene and sesquiterpene fluxes ($\mu g\ m^{-2}\ h^{-1}$) and concentrations ($\mu g\ m^{-3}$) in the O- and the A-horizons in the different measurement pits in 2016. The significance level of p<0.100 (o), p<0.050 (*), p<0.010 (**), p<0.001 (***)) was used.

| Concentration ($\mu g\ m^{-3}$) | | Monoterpene flux ($\mu g\ m^{-2}\ h^{-1}$) | | | Sesquiterpene flux ($\mu g\ m^{-2}\ h^{-1}$) | | |
|---|---|---|---|---|---|---|---|
| Pit | Horizon | Correlation | N | P value | Correlation | N | P value |
| 1 | O | -0.19 | 11 | 0.70 | 0.02 | 11 | 0.47 |
| 1 | A | -0.27 | 11 | 0.78 | -0.09 | 11 | 0.61 |
| 2 | A | 0.20 | 11 | 0.29 | **0.46** | 12 | 0.07º |
| 3 | O | -0.39 | 13 | 0.91 | **0.62** | 9 | 0.04* |
| 3 | A | -0.55 | 12 | 0.97 | **0.72** | 10 | 0.01** |
| 4 | O | **0.78** | 7 | 0.020* | -0.43 | 7 | 0.83 |
| 4 | A | -0.34 | 9 | 0.81 | -0.16 | 10 | 0.67 |
| 5 | O | -0.10 | 6 | 0.57 | 0.01 | 5 | 0.99 |
| 5 | A | **0.83** | 6 | 0.020* | 0.54 | 5 | 0.17 |

**Table A6.** Pearson correlation between the total monoterpene and sesquiterpene fluxes ($\mu g\ m^{-2}\ h^{-1}$) and chamber temperature (°C), and $CO_2$ fluxes ($\mu g\ m^{-2}\ h^{-1}$) from the soil surface in spring, summer, and autumn in 2016. The significance level of p<0.100 (o), p<0.050 (*), p<0.010 (**), p<0.001 (***)) was used. VOC fluxes were measured from the different measurement pits in 2016.

| Period | Correlation coefficient | N | P value | Correlation coefficient | N | P value |
|---|---|---|---|---|---|---|
| **Chamber temperature (C°)** | Monoterpene flux ($\mu g\ m^{-2}\ h^{-1}$) | | | Sesquiterpene flux ($\mu g\ m^{-2}\ h^{-1}$) | | |
| April-June | -0.12 | 16 | 0.67 | **0.52** | **14** | **0.02*** |
| July-September | **0.43** | **19** | **0.03*** | 0.24 | 19 | 0.16 |
| October-December | **0.62** | **18** | **0.003**** | **0.47** | **17** | **0.03*** |
| **$CO_2$ flux ($\mu g\ m^{-2}\ h^{-1}$)** | Monoterpene flux ($\mu g\ m^{-2}\ h^{-1}$) | | | Sesquiterpene flux ($\mu g\ m^{-2}\ h^{-1}$) | | |
| April-June | 0.30 | 13 | 0.16 | -0.42 | 12 | 0.91 |
| July-September | -0.42 | 15 | 0.94 | **0.57** | **15** | **0.01*** |
| October-December | **0.79** | **13** | **0.0007***** | **0.63** | **13** | **0.01*** |

**Table A7**. Pearson correlation between the $CO_2$ flux and the total monoterpene and sesquiterpene concentrations from the O- and the A-horizon in spring, summer, and autumn in 2016. The significance level of p<0.1 (o), p<0.05 (*), p<0.01 (**), p<0.001 (***)) was used.

| CO$_2$ flux | | Monoterpene concentration | | | Sesquiterpene concentration | | |
|---|---|---|---|---|---|---|---|
| Period | Horizon | Correlation | N | P value | Correlation | N | P value |
| April-June | O | 0.36 | 8 | 0.19 | -0.16 | 8 | 0.65 |
| July-September | O | -0.03 | 12 | 0.54 | -0.15 | 10 | 0.66 |
| October-December | O | 0.14 | 9 | 0.36 | **0.68** | **9** | **0.02*** |
| April-June | A | 0.35 | 13 | 0.12 | -0.25 | 13 | 0.80 |
| July-September | A | 0.20 | 17 | 0.22 | **0.50** | **14** | **0.04*** |
| October-December | A | **0.76** | **12** | **0.002**** | -0.12 | 10 | 0.63 |